# The geometry and dimensionality of brain-wide activity

Zezhen Wang[1†], Weihao Mai[2†], Yuming Chai[3,4], Kexin Qi[3,4], Hongtai Ren[5], Chen Shen[3,4], Shiwu Zhang[5], Guodong Tan[4], Yu Hu[2,6*], Quan Wen[1,3,4*]

[1]School of Data Science, University of Science and Technology of China, Hefei, China; [2]Division of Life Science, The Hong Kong University of Science and Technology, Hong Kong, China; [3]Hefei National Laboratory for Physical Sciences at the Microscale, Center for Integrative Imaging, University of Science and Technology of China, Hefei, China; [4]Division of Life Sciences and Medicine, University of Science and Technology of China, Hefei, China; [5]Department of Precision Machinery and Precision Instrumentation, University of Science and Technology of China, Hefei, China; [6]Department of Mathematics, The Hong Kong University of Science and Technology, Hong Kong, China

**\*For correspondence:**
mahy@ust.hk (YH);
qwen@ustc.edu.cn (QW)

[†]These authors contributed equally to this work

## eLife Assessment

This **important** study shows a surprising scale-invariance of the covariance spectrum of large-scale recordings in the zebrafish brain in vivo. A **convincing** analysis demonstrates that a Euclidean random matrix model of the covariance matrix recapitulates these properties. The results provide several new and insightful approaches for probing large-scale neural recordings.

**Abstract** Understanding neural activity organization is vital for deciphering brain function. By recording whole-brain calcium activity in larval zebrafish during hunting and spontaneous behaviors, we find that the shape of the neural activity space, described by the neural covariance spectrum, is scale-invariant: a smaller, *randomly sampled* cell assembly resembles the entire brain. This phenomenon can be explained by Euclidean Random Matrix theory, where neurons are reorganized from anatomical to functional positions based on their correlations. Three factors contribute to the observed scale invariance: slow neural correlation decay, higher functional space dimension, and neural activity heterogeneity. In addition to matching data from zebrafish and mice, our theory and analysis demonstrate how the geometry of neural activity space evolves with population sizes and sampling methods, thus revealing an organizing principle of brain-wide activity.

## Introduction

Geometric analysis of neuronal population activity has revealed the fundamental structures of neural representations and brain dynamics (*Churchland et al., 2012*; *Zhang et al., 2023*; *Kriegeskorte and Wei, 2021*; *Chung and Abbott, 2021*). Dimensionality reduction methods, which identify collective or latent variables in neural populations, simplify our view of high-dimensional neural data (*Cunningham and Yu, 2014*).Their applications to optical and multi-electrode recordings have begun to reveal important mechanisms by which neural cell assemblies process sensory information (*Stringer et al., 2019a*; *Si et al., 2019*), make decisions (*Mante et al., 2013*; *Yang et al., 2022*), maintain working memory (*Xie et al., 2022*) and generate motor behaviors (*Churchland et al., 2012*; *Nguyen et al., 2016*; *Lindén et al., 2022*; *Urai et al., 2022*).

In the past decade, the number of neurons that can be simultaneously recorded in vivo has grown exponentially (*Buzsáki, 2004*; *Ahrens et al., 2012*; *Jun et al., 2017*; *Stevenson and Kording, 2011*; *Nguyen et al., 2016*; *Sofroniew et al., 2016*; *Lin et al., 2022*; *Meshulam et al., 2019*; *Demas et al., 2021*). This increase spans various brain regions (*Musall et al., 2019*; *Stringer et al., 2019a*; *Jun et al., 2017*) and the entire mammalian brain (*Stringer et al., 2019b*; *Kleinfeld et al., 2019*). As more neurons are recorded, the multidimensional neural activity space, with each axis representing a neuron's activity level (*Figure 1A*), becomes more complex. The changing size of observed cell assemblies raises a number of basic questions. How does this space's geometry evolve and what structures remain invariant with increasing number of neurons recorded?

A key measure, the *effective dimension* or *participation ratio* (denoted as $D_{PR}$, *Figure 1B*), captures a major part of variability in neural activity (*Recanatesi et al., 2019*; *Litwin-Kumar et al., 2017*; *Gao et al., 2017* ; *Clark et al., 2023*; *Dahmen et al., 2020*). How does $D_{PR}$ vary with the number of sampled neurons (*Figure 1A*)? Two scenarios are possible: $D_{PR}$ grows continuously with more sampled neurons; $D_{PR}$ saturates as the sample size increases. Which scenario fits the brain? Furthermore, even if two cell assemblies have the same $D_{PR}$, they can have different shapes (the geometric configuration of the neural activity space, as dictated by the eigenspectrum of the covariance matrix, *Figure 1C*). How does the shape vary with the number of neurons sampled? Lastly, are we going to observe the same picture of neural activity space when using different recording methods such as two-photon microscopy, which records all neurons in a brain region, versus Neuropixels (*Jun et al., 2017*), which conducts a broad random sampling of neurons?

Here, we aim to address these questions by analyzing brain-wide Ca$^{2+}$ activity in larval zebrafish during hunting or spontaneous behavior (*Figure 2A*) recorded by Fourier light-field microscopy (*Cong et al., 2017*). The small size of this vertebrate brain, together with the volumetric imaging method, enables us to capture a significant amount of neural activity across the entire brain simultaneously. To characterize the geometry of neural activity beyond its dimensionality $D_{PR}$, we examine the eigenvalues or spectrum of neural covariance (*Hu and Sompolinsky, 2022*; *Figure 1C*). The covariance spectrum has been instrumental in offering mechanistic insights into neural circuit structure and function, such as the effective strength of local recurrent interactions and the depiction of network motifs (*Hu and Sompolinsky, 2022*; *Morales et al., 2023*; *Dahmen et al., 2020*). Intriguingly, we find that both the dimensionality and covariance spectrum remain invariant for cell assemblies that are randomly selected from various regions of the zebrafish brain. We also verify this observation in datasets recorded by different experimental methods, including light-sheet imaging of larval zebrafish (*Chen et al., 2018*), two-photon imaging of mouse visual cortex (*Stringer et al., 2019b*), and multi-area Neuropixels recording in the mouse (*Stringer et al., 2019b*). To explain the observed phenomenon, we model the covariance matrix of brain-wide activity by generalizing the Euclidean Random Matrix (ERM) (*Mézard et al., 1999*) such that neurons correspond to points distributed in a $d$-dimensional functional or feature space, with pairwise correlation decaying with distance. The ERM theory, studied in theoretical physics (*Mézard et al., 1999Goetschy and Skipetrov, 2013*), provides extensive analytical tools for a deep understanding of the neural covariance matrix model, allowing us to unequivocally identify three crucial factors for the observed scale invariance.

Building upon our theoretical results, we further explore the connection between the spatial arrangement of neurons and their locations in functional space, which allows us to distinguish among three sampling approaches: random sampling, anatomical sampling (akin to optical recording of all neurons within a specific region of the brain) and functional sampling (*Meshulam et al., 2019*). Our ERM theory makes distinct predictions regarding the scaling relationship between dimensionality and the size of cell assembly, as well as the shape of covariance eigenspectrum under various sampling methods. Taken together, our results offer a new perspective for interpreting brain-wide activity and unambiguously show its organizing principles, with unexplored consequences for neural computation.

## Results

### Geometry of neural activity across random cell assemblies in zebrafish brain

We recorded brain-wide Ca$^{2+}$activity at a volume rate of 10 Hz in head-fixed larval zebrafish (*Figure 2A*) during hunting attempts (Methods) and spontaneous behavior using a Fourier light-field microscopy

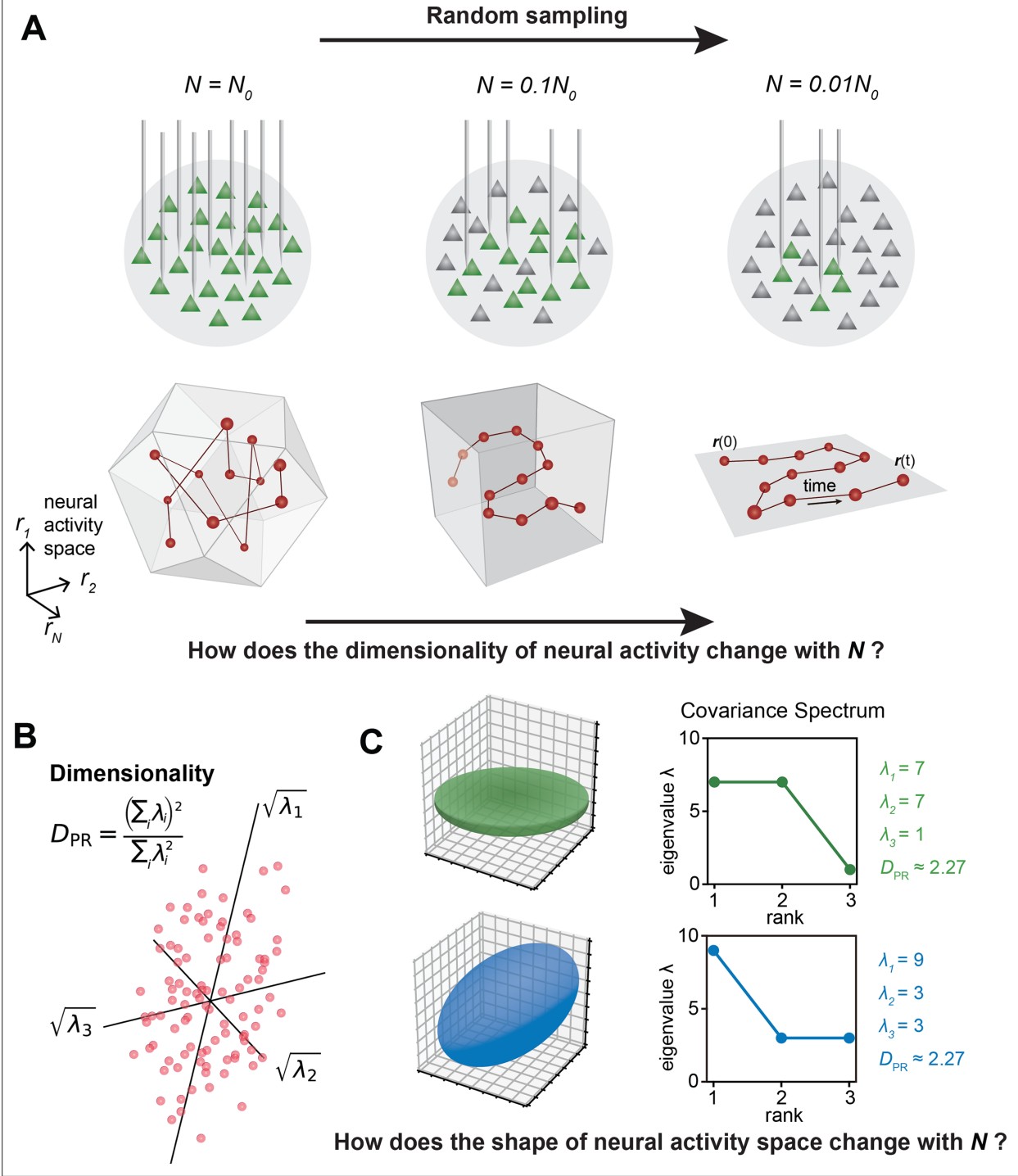

**Figure 1.** The relationship between the geometric properties of the neural activity space and the size of neural assemblies. (**A**) Illustration of how dimensionality of neural activity ($D_{PR}$) changes with the number of recorded neurons. (**B**) The eigenvalues of the neural covariance matrix dictate the geometrical configuration of the neural activity space with $\sqrt{\lambda_i}$ being the distribution width along a principal axis. (**C**) Examples of two neural populations with identical dimensionality ($D_{PR} = 25/11 \approx 2.27$) but different spatial configurations, as revealed by the eigenvalue spectrum (green: $\{\lambda_i\} = \{7, 7, 1\}$, blue: $\{\lambda_i\} = \{9, 3, 3\}$).

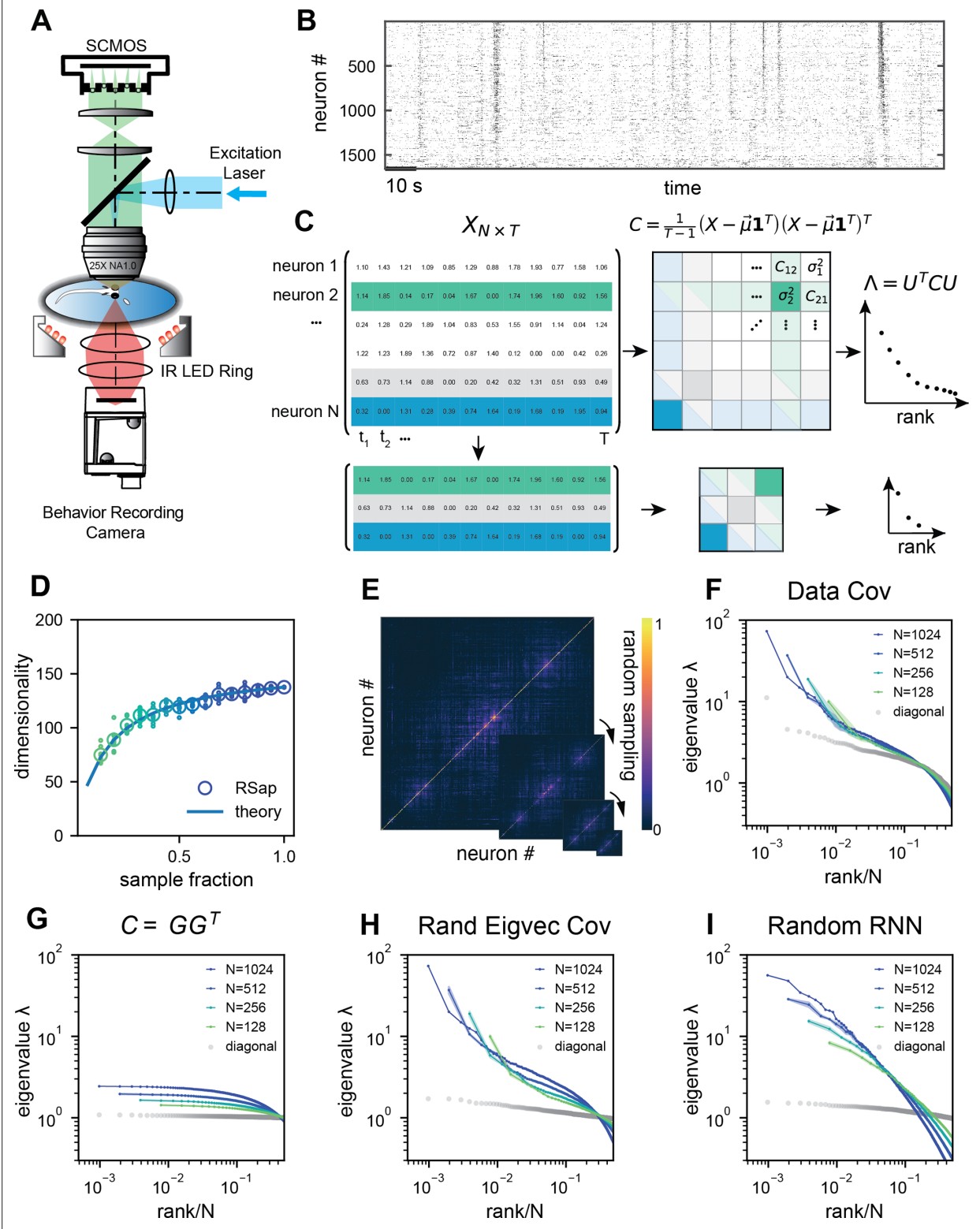

**Figure 2.** Whole-brain calcium imaging of zebrafish neural activity and the phenomenon of its scale-invariant covariance eigenspectrum. (**A**) Rapid light-field $Ca^{2+}$ imaging system for whole-brain neural activity in larval zebrafish. (**B**) Inferred firing rate activity from the brain-wide calcium imaging. The ROIs are sorted by their weights in the first principal component (***Stringer et al., 2019b***). (**C**) Procedure of calculating the covariance spectrum on the full and sampled neural activity matrices. (**D**) Dimensionality (circles, average across eight samplings (dots)), as a function of the sampling fraction. The curve is the predicted dimensionality using ***Equation 5***. (**E**) Iteratively sampled covariance matrices. Neurons are sorted in each matrix to maximize values

*Figure 2 continued*

near the diagonal. (**F**) The covariance spectra, that is, eigenvalue versus rank/*N*, for randomly sampled neurons of different sizes (colors). The gray dots represent the sorted variances $C_{ii}$ of all neurons. (**G–I**) Same as **F** but from three models of covariance (see details in Methods): (**G**) a Wishart random matrix calculated from a random activity matrix of the same size as the experimental data; (**H**) replacing the eigenvectors by a random orthogonal set; (**I**) covariance generated from a randomly connected recurrent network. The collapse index (CI), which quantifies the level of scale invariance in the eigenspectrum (see Methods), is: (**G**) CI = 0.214; (**H**) CI = 0.222; (**I**) CI = 0.139.

The online version of this article includes the following figure supplement(s) for figure 2:

**Figure supplement 1.** Experimental data description.

**Figure supplement 2.** The phenomenon of scale-invariant eigenspectra across different datasets.

**Figure supplement 3.** Negative covariances do not affect the eigenspectrum of the zebrafish data.

**Figure supplement 4.** Scale-invariant properties persist across different temporal sampling rates in neural recordings.

(*Cong et al., 2017*). Approximately 2000 ROIs (1977.3 ± 677.1, mean ± SD) with a diameter of 16.84 ± 8.51 µm were analyzed per fish based on voxel activity (Methods, *Figure 2—figure supplement 1*). These ROIs likely correspond to multiple nearby neurons with correlated activity. Henceforth, we refer to the ROIs as 'neurons' for simplicity.

We first investigate the dimensionality of neural activity $D_{\mathrm{PR}}$ (*Figure 1B*) in a randomly chosen cell assembly in zebrafish, similar to multi-area Neuropixels recording in a mammalian brain. We focus on how $D_{\mathrm{PR}}$ changes with a large sample size $N$. We find that if the mean squared covariance remains finite instead of vanishing with $N$, the dimensionality $D_{\mathrm{PR}}$ (*Figure 1B*) becomes sample size independent and depends only on the variance $\sigma_i^2$ and the covariance $C_{ij}$ between neurons $i$ and $j$:

$$\lim_{N\to\infty} D_{\mathrm{PR}} = \frac{\mathrm{E}(\sigma_i^2)^2}{\mathrm{E}_{i\neq j}(C_{ij}^2)}, \tag{1}$$

where $\mathrm{E}(\ldots)$ denotes average across neurons (Methods and *Dahmen et al., 2020*). The finite mean squared covariance condition is supported by the observation that the neural activity covariance $C_{ij}$ is positively biased and widely distributed with a long tail (*Figure 2—figure supplement 2A*). As predicted, the data dimensionality grows with sample size and reaches the maximum value specified by *Equation 1* (*Figure 2D*).

Next, we investigate the shape of the neural activity space described by the eigenspectrum of the covariance matrix derived from the activity of $N$ randomly selected neurons (*Figure 2C*). When the eigenvalues are arranged in descending order and plotted against the normalized rank $r/N$, where $r = 1, \ldots, N$ (we refer to it as the *rank plot*), this curve shows an approximate power law that spans 10 folds. Interestingly, as the size of the covariance matrices decreases ($N$ decreases), the eigenspectrum curves nearly collapse over a wide range of eigenvalues. This pattern holds across diverse datasets and experimental techniques (*Figure 2F*, *Figure 2—figure supplement 2E–L*). The similarity of the covariance matrices of randomly sampled neural populations can be intuitively visualized (*Figure 2E*), after properly sorting the neurons (Methods).

The scale invariance in the neural covariance matrix – the collapse of the covariance eigenspectrum under random sampling – is non-trivial. The spectrum is not scale invariant in a common covariance matrix model based on independent noise (*Figure 2G*). It is absent when replacing the neural covariance matrix eigenvectors with random ones, keeping the eigenvalues identical (*Figure 2H*). A recurrent neural network with random connectivity (*Hu and Sompolinsky, 2022*) does not yield a scale-invariant covariance spectrum (*Figure 2I*). A recently developed latent variable model (*Morrell et al., 2024*; *Appendix 1—figure 6*), which is able to reproduce avalanche criticality, also fails to generate the scale-invariant covariance spectrum. Thus, a new model is needed for the covariance matrix of neural activity.

## Modeling covariance by organizing neurons in functional space

Dimension reduction methods simplify and visualize complex neuron interactions by embedding them into a low-dimensional map, within which nearby neurons have similar activities. Inspired by these ideas, we use the ERM (*Mézard et al., 1999*) to model neural covariance. Imagine sprinkling neurons uniformly distributed on a $d$-dimensional *functional space* of size $L$ (*Figure 3A*), where the

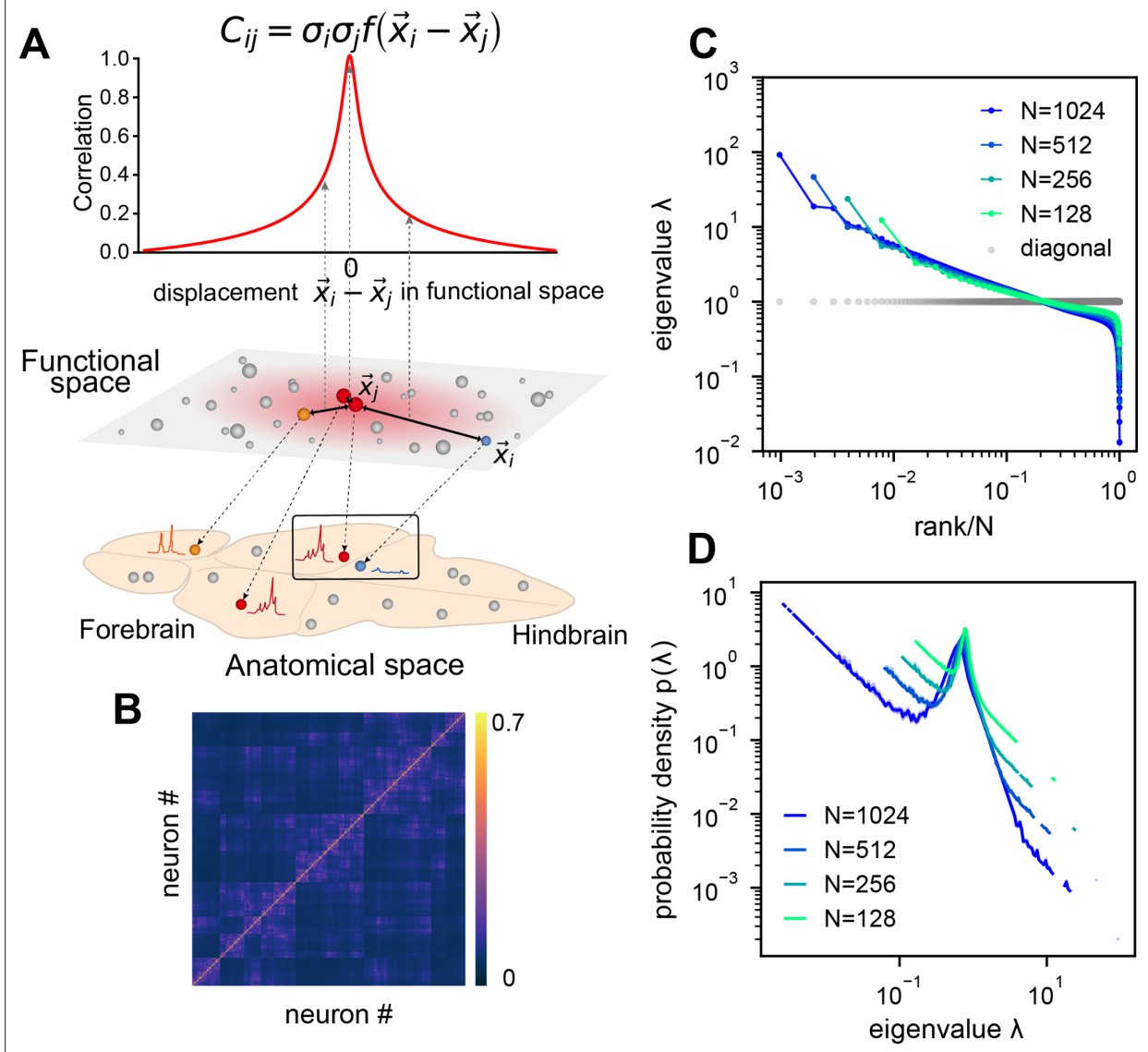

**Figure 3.** Euclidean Random Matrix (ERM) model of covariance and its eigenspectrum. (**A**) Schematic of the ERM model, which reorganizes neurons (circles) from the anatomical space to the functional space (here $d = 2$ is a two-dimensional box). The correlation between a pair of neurons decreases with their distance in the functional space according to a kernel function $f(\vec{x})$. This correlation is then scaled by neurons' variance $\sigma_i^2$ (circle size) to obtain the covariance $C_{ij}$. (**B**) An example ERM correlation matrix (i.e., when $\sigma_i^2 \equiv 1$). (**C**) Spectrum (same as *Figure 2F*) for the ERM correlation matrix in (**B**). The gray dots represent the sorted variances $C_{ii}$ of all neurons (same as in *Figure 2F*). (**D**) Visualizing the distribution of the same ERM eigenvalues in **C** by plotting the probability density function (pdf).

The online version of this article includes the following figure supplement(s) for figure 3:

**Figure supplement 1.** Covariance spectra under different kernel functions $f(\vec{x})$.

**Figure supplement 2.** Impact of $\eta$ and $d$ on the scale invariance of covariance eigenspectra in the Euclidean Random Matrix (ERM) with $f(\vec{x}) = e^{-\|\vec{x}\|^{\eta}}$.

distance between neurons $i$ and $j$ affects their correlation. Let $\vec{x}_i$ represent the functional coordinate of the neuron $i$. The distance-correlation dependency is described by *kernel function* $f(\vec{x}_i - \vec{x}_j) > 0$ with $f(0) = 1$, indicating closer neurons have stronger correlations, and decreases as distance $\|\vec{x}_i - \vec{x}_j\|$ increases (*Figure 3A* and Methods). To model the covariance, we extend the ERM by incorporating heterogeneity of neuron activity levels (shown as the size of the neuron in the functional space in *Figure 3A*).

$$C_{ij} = \sigma_i \sigma_j f(\vec{x}_i - \vec{x}_j), \quad i,j = 1, 2, \ldots, N. \tag{2}$$

The variance of neural activity $\sigma_i^2$ is drawn i.i.d. from a given distribution and is independent of neurons' position.

This multidimensional functional space may represent attributes to which neurons are tuned, such as sensory features (e.g., visual orientation *Hubel and Wiesel, 1959*, auditory frequency) and movement characteristics (e.g., direction, speed *Stefanini et al., 2020*; *Kropff et al., 2015*). In sensory systems, it represents stimuli as neural activity patterns, with proximity indicating similarity in features. For motor control, it encodes movement parameters and trajectories. In the hippocampus, it represents the place field of a place cell, acting as a cognitive map of physical space (*O'Keefe, 1976*; *Moser et al., 2008*; *Tingley and Buzsáki, 2018*).

We first explore the ERM with various forms of $f(\vec{x})$ and find that fast-decaying functions like Gaussian and exponential functions do not produce eigenspectra similar to the data and no scale invariance over random sampling (*Figure 3—figure supplement 1A–H* and Appendix 2). Thus, we turn to slow-decaying functions including the power law, which produce spectra similar to the data (*Figure 3C, D*; see also *Figure 3—figure supplement 2*). We adopt a particular kernel function because of its closed-form and analytical properties: $f(\vec{x}) = \epsilon^\mu(\epsilon^2 + \|\vec{x}\|^2)^{-\mu/2}$ (Methods). For large distance $\|\vec{x}\| \gg \epsilon$, it approximates a power law $f(\vec{x}) \approx \epsilon^\mu\|\vec{x}\|^{-\mu}$ and smoothly transitions at small distance to satisfy the correlation requirement $f(0) = 1$ (*Appendix 1—figure 3I, J*).

## Analytical theory on the conditions of scale invariance in ERM

To determine the conditions for scale invariance in ERM, we analytically calculate the eigenspectrum of covariance matrix $C$ (*Equation 2*) for large $N, L$ using the replica method (*Mézard et al., 1999*). A key order parameter emerging from this calculation is the neuron density $\rho := N/L^d$. In the high-density regime $\rho\epsilon^d \approx 1$, the covariance spectrum can be approximated in a closed form (Methods). For the slow-decaying kernel function $f(\vec{x})$ defined above, the spectrum for large eigenvalues follows a power law (Appendix 2):

$$\lambda \sim (r/N)^{-1+\frac{\mu}{d}}\rho^{\frac{\mu}{d}},$$

$$\text{and equivalently } p(\lambda) \sim \rho^{\frac{\mu}{d-\mu}}\lambda^{-\frac{2d-\mu}{d-\mu}},$$

(3)

where $r$ is the rank of the eigenvalues in descending order and $p(\lambda)$ is their probability density function. *Equation 3* intuitively explains the scale invariance over random sampling. Sampling in the ERM reduces the neuron density $\rho$. The eigenspectrum is $\rho$-independent whenever $\mu/d \approx 0$. This indicates two factors contributing to the scale invariance of the eigenspectrum. First, a small exponent $\mu$ in the kernel function $f(\vec{x})$ means that pairwise correlations slowly decay with functional distance and can be significantly positive across various functional modules and throughout the brain. For a given $\mu$, an increase in dimension $d$ improves the scale invariance. The dimension $d$ could represent the number of independent features or latent variables describing neural activity or cognitive states.

We verify our theoretical predictions by comparing sampled eigenspectra in finite-size simulated ERMs across different $\mu$ and $d$ (*Figure 4A*). We first consider the case of homogeneous neurons ($\sigma_i^2 \equiv 1$ in *Equation 2*, revisited later) in these simulations (*Figures 3C, D and 4A*), making $C$'s entries correlation coefficients. To quantitatively assess the level of scale invariance, we introduce a *collapse index* (CI, see Methods for a detailed definition). Motivated by *Equation 3*, the CI measures the shift of the eigenspectrum when the number of sampled neurons changes. The smaller CI values indicate higher scale invariance. Intuitively, it is defined as the area between spectrum curves from different sample sizes (*Figure 4A*, upper right). In the log–log scale rank plot, *Equation 3* shows the spectrum shifts vertically with $\rho$.

Thus, we define CI as this average displacement (*Figure 4A*, upper right, Methods), and a smaller CI means more scale invariant. Using CI, *Figure 4A* shows that scale invariance improves with slower correlation decay as $\mu$ decreases and the functional dimension $d$ increases. Conversely, with large $\mu$ and small $d$, the covariance eigenspectrum varies significantly with scale (*Figure 4A*).

Next, we consider the general case of unequal neural activity levels $\sigma_i^2$ and check for differences between the correlation (equivalent to $\sigma_i^2 \equiv 1$) and covariance matrix spectra. Using the collapsed index (CI), we compare the scale invariance of the two spectra in the experimental data. Intriguingly, the CI of the covariance matrix is consistently smaller (more scale-invariant) across all datasets

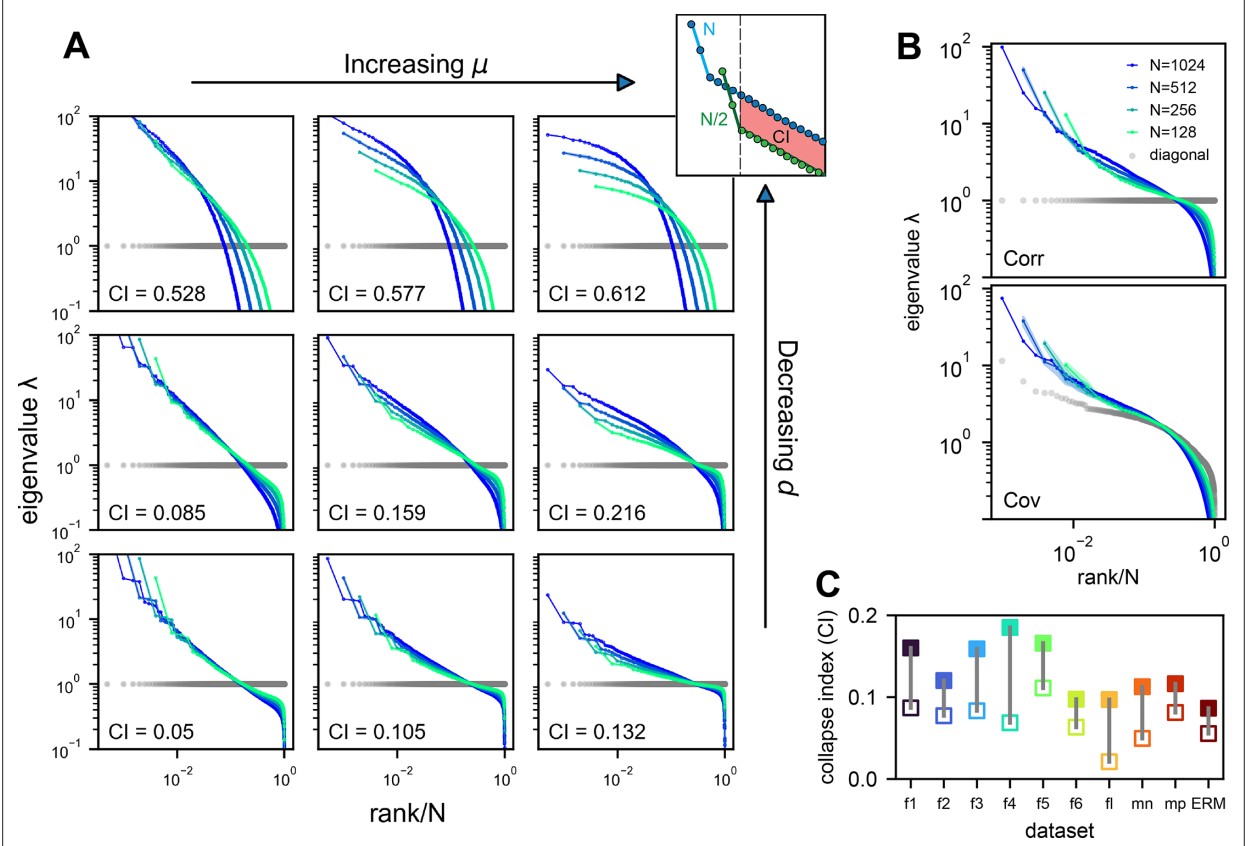

**Figure 4.** Three factors contributing to scale invariance. (**A**) Impact of $\mu$ and $d$ (see text) on the scale invariance of Euclidean Random Matrix (ERM) spectrum (same plots as *Figure 3C*) with $f(\vec{x}) = \epsilon^{\mu}(\epsilon^2 + \|\vec{x}\|^2)^{-\mu/2}$. The degree of scale invariance is quantified by the collapse index (CI), which essentially measures the area between different spectrum curves (upper right inset). For comparison, we fix the same coordinate range across panels hence some plots are cropped. The gray dots represent the sorted variances $C_{ii}$ of all neurons (same as in *Figure 2F*). (**B**) Top: sampled correlation matrix spectrum in an example animal (fish 1). Bottom: same as top but for the covariance matrix that incorporates heterogeneous variances. The gray dots represent the sorted variances $C_{ii}$ of all neurons (same as in *Figure 2F*). (**C**) The CI of the correlation matrix (filled squares) is found to be larger than that for the covariance matrix (opened squares) across different datasets: f1 to f6: six light-field zebrafish data (10 Hz per volume, this paper); fl: light-sheet zebrafish data (2 Hz per volume, *Chen et al., 2018*); mn: mouse Neuropixels data (downsampled to 10 Hz per volume); mp: mouse two-photon data (3 Hz per volume, *Stringer et al., 2019b*).

The online version of this article includes the following figure supplement(s) for figure 4:

**Figure supplement 1.** Comparison between Euclidean Random Matrix (ERM) simulation and theory.

**Figure supplement 2.** Impact of heterogeneous activity levels on the scale invariance.

(*Figure 4C*, *Figure 4—figure supplement 2C*, open vs. closed squares), indicating that the heterogeneity of neuronal activity variances significantly affects the eigenspectrum and the geometry of neural activity space (*Tian et al., 2024*). By extending our spectrum calculation to the intermediate density regime $\rho\epsilon^d \ll 1$ (Methods), we show that the ERM model can quantitatively explain the improved scale invariance in the covariance matrix compared to the correlation matrix (*Figure 4—figure supplement 2B*; *Table 1*).

Lastly, we examine factors that turn out to have minimal impact on the scale invariance of the covariance spectrum. First, the shape of the kernel function $f(\vec{x})$ over a small distance (small distance means $f(x)$ near $x = 0$ in the functional space, *Appendix 1—figure 3*) does not affect the distribution of large eigenvalues (*Appendix 1—figure 3*, Table 3, *Appendix 1—figure 2*, *Appendix 1—figure 1A*). This supports our use of a specific $f(\vec{x})$ to represent a class of slow-decaying kernels. Second, altering the spatial distribution of neurons in the functional space, whether using a Gaussian, uniform, or clustered distribution, does not affect large covariance eigenvalues, except possibly the leading ones (*Appendix 1—figure 1B*, Appendix 1). Third, different geometries of the functional space, such as a flat square, a sphere, or a hemisphere, result in eigenspectra similar to the original ERM model

**Table 1.** Table of notations.

| Notation | Description |
| --- | --- |
| $C$ | Covariance matrix, **Equation 2** |
| $C_{ij}$ | Pairwise covariance between neuron $i$, $j$; entries of $C$ |
| $D_{\mathrm{PR}}$ | Participation ratio dimension, **Equation 5** |
| $D_{\mathrm{PR}}^{\mathrm{ASap}}$ | Anatomical sampling dimension, **Equation 4** |
| $\lambda$ | Eigenvalue of a covariance matrix $C$ |
| $p(\lambda)$ | Probability density function of covariance eigenvalues, **Equation 8** |
| $r$ | Rank of an eigenvalue in descending order, **Equation 3** |
| $q$ | Fraction of eigenvalues up to $\lambda$ and $q = r/N$; **Equation 13** |
| $f(\vec{x}) = f(\|\vec{x}_i - \vec{x}_j\|)$ | Kernel function or distance-correlation function, **Equation 11** |
| $\tilde{f}(\vec{k})$ | Fourier transform of $f(\vec{x})$, $\tilde{f}(\vec{k}) = \int_{\mathbb{R}^d} f(\vec{x}) e^{-i\vec{x}\cdot\vec{k}} d^d\vec{x}$ |
| $\mu$ | Power-law exponent in $f(\vec{x})$, **Equation 11** |
| $\varepsilon$ | Resolution parameter in $f(\vec{x})$ to smooth the singularity near 0, **Equation 11** |
| $N$ | Number of neurons |
| $N_0$ | The total number of neurons prior to sampling |
| $k$ | $N/N_0$ the fraction of sampled neurons |
| $L$ | Linear box size of the functional space |
| $\rho$ | Density of neurons in the functional space, **Equation 3** |
| $d$ | Dimension of the functional space, **Equation 3** |
| $a_i(t)$ | Neural activity of neuron $i$ at time $t$ |
| $\sigma_i^2$ | Temporal variance of neural activity, **Equation 2** |
| CI | Collapse index for measuring scale invariance, **Equation 13** |
| $\alpha$ | Power-law coefficient of eigenspectrum in the rank plot, see Discussion |
| $\vec{x}_i, \vec{y}_i$ | Neuron $i$'s coordinate in the functional and anatomical space, respectively |
| $\vec{v}_{func}, \vec{v}_{anat}$ | The first canonical directions in the functional and anatomical space, respectively |
| $R_{\mathrm{CCA}}$ | The first canonical correlation |
| $R_{\mathrm{ASap}}$ | Correlation between anatomical and functional coordinates along ASap direction, **Equation 4** |

(**Appendix 1—figure 1C**). These findings indicate that our theory for the covariance spectrum's scale invariance is robust to various modeling details.

## Connection among random sampling, functional sampling, and anatomical sampling

So far, we have focused on random sampling of neurons, but how does the neural activity space change with different sampling methods? To this end, we consider three methods (**Figure 5A1**): random sampling (RSap), anatomical sampling (ASap) where neurons in a brain region are captured by optical imaging (**Grewe and Helmchen, 2009**; **Gauthier and Tank, 2018**; **Stringer et al., 2019a**), and functional sampling (FSap) where neurons are selected based on activity similarity (**Meshulam et al., 2019**). In ASap or FSap, sampling involves expanding regions of interest in anatomical space or functional space while measuring all neural activity within those regions (Appendix 1). The difference

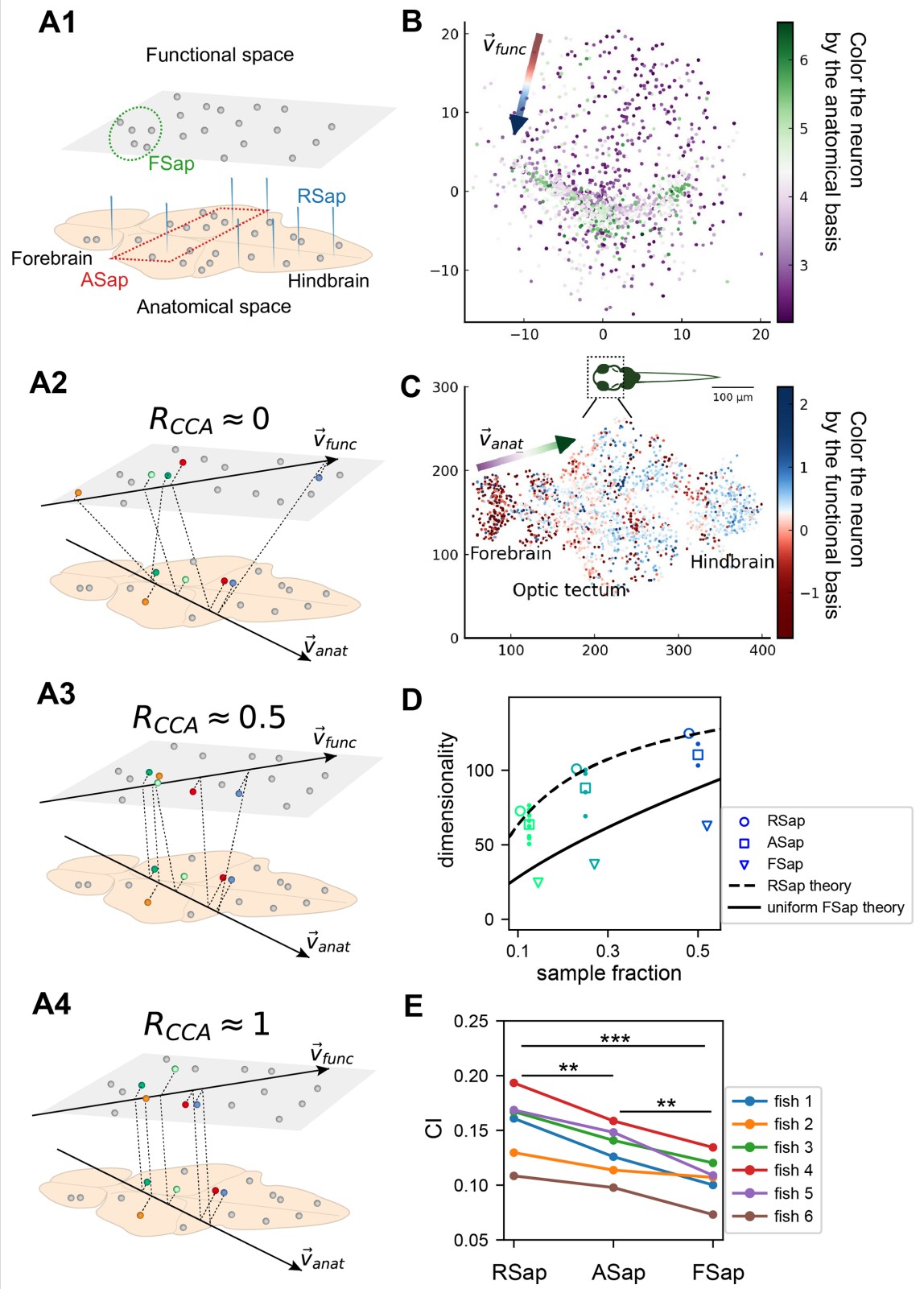

**Figure 5.** The relationship between the functional and anatomical space and theoretical predictions. (**A**) Three sampling methods (**A1**) and $R_{CCA}$ (see text). When $R_{CCA} \approx 0$ (**A2**), the anatomical sampling (ASap) resembles the random sampling (RSap), and while when $R_{CCA} \approx 1$ (**A4**), ASap is similar to the functional sampling (FSap). (**B**) Distribution of neurons in the functional space inferred by MDS. Each neuron is color-coded by its projection along the first canonical direction $\vec{v}_{anat}$ in the anatomical space (see text). Data based on fish 6, same for (**C-E**). (**C**) Similar to (**B**) but plotting neurons in the

Figure 5 continued

anatomical space with color based on their projection along $\vec{v}_{\text{func}}$ in the functional space (see text). (**D**) Dimensionality ($D_{\text{PR}}$) across sampling methods: average $D_{\text{PR}}$ under RSap (circles), average and individual brain region $D_{\text{PR}}$ under ASap (squares and dots), and $D_{\text{PR}}$ under FSap for the most correlated neuron cluster (triangles; Methods). Dashed and solid lines are theoretical predictions for $D_{\text{PR}}$ under RSap and FSap, respectively (Methods). (**E**) The CI of correlation matrices under three sampling methods in six animals (colors). **p < 0.01; ***p < 0.001; one-sided paired $t$ tests: RSap versus ASap, p = 0.0010; RSap versus FSap, p = 0.0004; ASap versus FSap, p = 0.0014.

The online version of this article includes the following figure supplement(s) for figure 5:

**Figure supplement 1.** Fitting Euclidean Random Matrix (ERM) to zebrafish data from our experiments (part 1).

**Figure supplement 2.** Fitting Euclidean Random Matrix (ERM) to all six zebrafish data from our experiments (part 2).

**Figure supplement 3.** Fitting Euclidean Random Matrix (ERM) to all six zebrafish data from our experiments (part 3).

**Figure supplement 4.** Fitting Euclidean Random Matrix (ERM) to all six zebrafish data from our experiments (part 4).

**Figure supplement 5.** Relationship between the functional space and anatomical space for each zebrafish dataset from our experiments.

**Figure supplement 6.** Dimensionality ($D_{\text{PR}}$) across sampling methods in fish data.

**Figure supplement 7.** Dimensionality ($D_{\text{PR}}$) across sampling methods in Euclidean Random Matrix (ERM).

among sampling methods depends on the neuron organization throughout the brain. If anatomically localized neurons also cluster functionally (*Figure 5A4*), ASap ≈ FSap; if they are spread in the functional space (*Figure 5A2*), ASap ≈ RSap. Generally, the anatomical–functional relationship is in-between and can be quantified using the Canonical Correlation Analysis (CCA). This technique finds axes (CCA vectors $\vec{v}_{\text{anat}}$ and $\vec{v}_{\text{func}}$) in anatomical and functional spaces such that the neurons' projection along these axes has the maximum correlation, $R_{\text{CCA}}$. The extreme scenarios described above correspond to $R_{\text{CCA}} = 1$ and $R_{\text{CCA}} = 0$.

To determine the anatomical–functional relationship in neural data, we infer the functional coordinates $\vec{x}_i$ of each neuron by fitting the ERM using multidimensional scaling (MDS) (*Cox and Cox, 2000*) (Methods). For simplicity and better visualization, we use a low-dimensional functional space where $d = 2$. The fitted functional coordinates confirm the slow decay kernel function in ERM except for a small distance (*Figure 5—figure supplement 3*). The ERM with inferred coordinates $\vec{x}_i$ also reproduces the experimental covariance matrix, including cluster structures (*Figure 5—figure supplement 2*) and its sampling eigenspectra (*Figure 5—figure supplement 1*).

Equipped with the functional and anatomical coordinates, we next use CCA to determine which scenarios illustrated in *Figure 5A* align better with the neural data. *Figure 5B, C* shows a representative fish with a significant $R_{\text{CCA}} = 0.327$ (p-value = 0.0042, Anderson–Darling test). Notably, the CCA vector in the anatomical space, $\vec{v}_{\text{anat}}$, aligns with the rostrocaudal axis. Coloring each neuron in the functional space by its projection along $\vec{v}_{\text{anat}}$ shows a correspondence between clustering and anatomical coordinates (*Figure 5B*). Similarly, coloring neurons in the anatomical space (*Figure 5C*) by their projection along $\vec{v}_{\text{func}}$ reveals distinct localizations in regions like the forebrain and optic tectum. Across animals, functionally clustered neurons show anatomical segregation (*Chen et al., 2018*), with an average $R_{\text{CCA}}$ of 0.335 ± 0.054 (mean ± SD).

Next, we investigate the effects of different sampling methods (*Figure 5A1*) on the geometry of the neural activity space when there is a significant but moderate anatomical–functional correlation as in the experimental data. Interestingly, dimensionality $D_{\text{PR}}^{\text{ASap}}$ in data under anatomical sampling consistently falls between random and functional sampling values (*Figure 5D*). This phenomenon can be intuitively explained by the ERM theory. Recall that for large $N$, the key term in *Equation 1* is $\text{E}_{i \neq j}(C_{ij}^2)$. For a fixed number of sampled neurons, this average squared covariance is maximized when neurons are selected closely in the functional space (FSap) and minimized when distributed randomly (RSap). Thus, RSap and FSap $D_{\text{PR}}$ set the upper and lower bounds of dimensionality, with ASap expected to fall in between. This reasoning can be precisely formulated to obtain quantitative predictions of the bounds (Methods). We predict the ASap dimension at large $N$ as

$$D_{\text{PR}}^{\text{ASap}} \approx (1 - R_{\text{ASap}}^2 + k^2 R_{\text{ASap}}^2)^{\mu/d} D_{\text{PR}}. \tag{4}$$

Here, $D_{\text{PR}}$ is the dimensionality under RSap (*Equation 1*), $k$ represents the fraction of sampled neurons. $R_{\text{ASap}}$ is the correlation between anatomical and functional coordinates along the direction where the anatomical subregions are divided (Methods), and it is bounded by the canonical correlation

$R_{\text{ASap}} \leq R_{\text{CCA}}$. When $R_{\text{ASap}} = 0$, we get the upper bound $D_{\text{PR}}^{\text{ASap}} = D_{\text{PR}}$ (*Figure 5D*, dashed line). The lower bound is reached when $R_{\text{ASap}} = R_{\text{CCA}} = 1$ (*Figure 5A4*), where *Equation 4* shows a scaling relationship $D_{\text{PR}}^{\text{ASap}} = D_{\text{PR}}^{\text{FSap}} \sim k^{2\mu/d} D_{\text{PR}}$ that depends on the sampling fraction $k$ (*Figure 5D*, solid line). This contrasts with the $k$-independent dimensionality of RSap in *Equation 1*. Furthermore, if $R_{\text{ASap}}$ and its upper bound is not close to 1 (precisely $R_{\text{ASap}} \leq 0.84$ for the ERM model in *Figure 5D*), $D_{\text{PR}}^{\text{ASap}}$ align closer to the upper bound of RSap. This prediction agrees well with our observations in data across animals (*Figure 5D*, *Figure 5—figure supplement 6*, and *Figure 5—figure supplement 7*).

Beyond dimensionality, our theory predicts the difference in the covariance spectrum between sampling methods based on the neuronal density $\rho$ in the functional space (*Equation 3*). This density $\rho$ remains constant during FSap (*Figure 5A1*) and decreases under RSap; the average density across anatomical regions $\langle \rho \rangle$ in ASap lies between those of FSap and RSap. Analogous to *Equation 4*, the relationship in $\rho$ orders the spectra: ASap's spectrum lies between those of FSap and RSap (Methods). This further implies that the level of scale invariance under ASap should fall between that of RSap and FSap, which is confirmed by our experimental data (*Figure 5E*).

## Discussion

### Impact of hunting behavior on scale invariance and functional space organization

How does task-related neural activity shape the covariance spectrum and brain-wide functional organization? We examine the hunting behavior in larval zebrafish, marked by eye convergence (both eyes move inward to focus on the central visual field) (*Bianco et al., 2011*). We find that scale invariance of the eigenspectra persists and is enhanced even after removing the hunting frames from the Ca²⁺ imaging data (*Figure 4C*, *Appendix 1—figure 7A, B*, Appendix 1). This is consistent with the scale-invariant spectrum found in other datasets during spontaneous behaviors (*Figure 5—figure supplement 1F*, *Figure 2—figure supplement 2G, H*), suggesting scale invariance is a general phenomenon.

Interestingly, in the inferred functional space, we observe reorganizations of neurons after removing hunting behavior (*Appendix 1—figure 7C, D*). Neurons in one cluster disperse from their center of mass (*Appendix 1—figure 7D*) and decreases the local neuronal density $\rho$ (Appendix 1 and *Appendix 1—figure 7E*). The neurons in this dispersed cluster have a consistent anatomical distribution from the midbrain to the hindbrain in four out of five fish (*Appendix 1—figure 9*). During hunting, the cluster has robust activations that are widespread in the anatomical space but localized in the functional space (Appendix 1, *Appendix 1—Video 1*).

Our findings suggest that the functional space could be defined by latent variables that represent cognitive factors such as decision-making, memory, and attention. These variables set the space's dimensions, with neural activity patterns reflecting cognitive state dynamics. Functionally related neurons – through sensory tuning, movement parameters, internal conditions, or cognitive factors – become closer in this space, leading to stronger activity correlations.

### Criticality and power law

What drives brain dynamics with a slow-decaying distance–correlation function $f(\vec{x})$ in functional space? Long-range connections and a slow decline in projection strength over distance (*Kunst et al., 2019*) may cause extensive correlations, enhancing global activity patterns. This behavior is also reminiscent of phase transitions in statistical mechanics (*Kardar, 2007*), where local interactions lead to expansive correlated behaviors. Studies suggest that critical brains optimize information processing (*Beggs and Plenz, 2003*; *Dahmen et al., 2019*). The link between neural correlation structures and neuronal connectivity topology is an exciting area for future exploration.

In the high-density regime of the ERM model, the rank plot (*Equation 3*) for large eigenvalues ($\lambda > 1$) follows a power law $\lambda \sim r^{-\alpha}$, with $\alpha = 1 - \mu/d < 1$. The scale-invariant spectrum occurs when $\alpha$ is close to 1. Experimental data, however, align more closely with the model in the intermediate-density regime, where the power-law spectrum is an approximation and the decay is slower (for ERM model, *Figure 4—figure supplement 1BC*, and for data $\alpha = 0.47 \pm 0.08$, mean ± SD, $n = 6$ fish). *Stringer et al., 2019a* found an $\alpha \gtrsim 1$ decay in the mouse visual cortex's stimulus trial averaged covariance spectrum, and they argued that this decay optimizes visual code efficiency and smoothness.

Our study differs in two fundamental ways. First, we recorded brain-wide activity during spontaneous or hunting behavior, calculating neural covariance from single-trial activity. Much of the neural activity was not driven by sensory stimulus and unrelated to specific tasks, requiring a different interpretation of the neural covariance spectrum. Second, without loss of generality, we normalized the mean variance of neural activity $\mathrm{E}(\sigma^2)$ by scaling the covariance matrix so that its eigenvalues sum up to $N$. This normalization imposes a constraint on the spectrum. In particular, large and small eigenvalues may have different behaviors and do not need to obey a single power law $\lambda \sim r^{-\alpha}$ for all $N$ eigenvalues (*Pospisil and Pillow, 2024*) (Methods). *Stringer et al., 2019a* did not take this possibility into account, making their theory less applicable to our analysis.

We draw inspiration from the renormalization group (RG) approach to navigate neural covariance across scales, which has also been explored in the recent literature. Following Kadanoff's block spin transformation (*Kardar, 2007*, *Meshulam et al., 2019*) formed size-dependent neuron clusters and their covariance matrices by iteratively pairing the most correlated neurons and placing them adjacent on a lattice. The groups expanded until the largest reached the system size. The RG process, akin to spatial sampling in functional space (FSap), maintains constant neuron density $\rho$. Thus, for any kernel function $f(\vec{x})$, including the power law and exponential, the covariance eigenspectrum remains invariant across scales (*Appendix 1—figure 5A, B, D, E*).

*Morrell et al., 2024*; *Morrell et al., 2021* proposed a simple model in which a few time-varying latent factors impact the whole neural population. We evaluated if this model could account for the scale invariance seen in our data. Simulations showed that the resulting eigenspectra differed considerably from our findings (*Appendix 1—figure 6*). Although the Morrell model demonstrated a degree of scale invariance under functional sampling (or RG), it did not align with the scale-invariant features under random sampling, suggesting that this simple model might not capture all crucial features in our observations.

We emphasize that the covariance spectrum being a power law is distinct from the scale invariance we define in this study, namely the collapse of spectrum curves under random neuron sampling. The random RNN model in *Figure 2I* shows a power-law behavior, but lacks true scale invariance as spectrum curves for different sizes do not collapse. When connection strength $g$ approaches 1, the system exhibits a power-law spectrum of $\lambda \propto \left(\frac{r}{N}\right)^{-\frac{3}{2}}$. Subsampling causes the spectrum to shift by $\lambda \propto k^{-\frac{1}{2}} \left(\frac{r}{N}\right)^{-\frac{3}{2}}$, where $k = N_s/N$ is the sampling fraction (derived from Equation 24 in *Hu and Sompolinsky, 2022*).

## Bounded dimensionality under random sampling

The saturation of the dimensionality $D_{\mathrm{PR}}$ at large sample sizes indicates a limit to neural assembly complexity, evidenced by the finite mean square covariance. This is in contrast with neural dynamics models such as the balanced excitatory–inhibitory (E–I) neural network (*Renart et al., 2010*), where $\mathrm{E}_{i \neq j}(C_{ij}^2) \sim 1/N$ resulting in an unbounded dimensionality (see Appendix 2). Our results suggest that the brain encodes experiences, sensations, and thoughts using a finite set of dimensions instead of an infinitely complex neural activity space.

We found that the relationship between dimensionality and the number of recorded neurons depends on the sampling method. For functional sampling, the dimensionality scales with the sampling fraction $k : D_{PR}^{FSap} \sim k^{2\mu/d} D_{PR}$. This suggests that if anatomically sampled neurons are functionally clustered, as with cortical neurons forming functional maps, the increase in dimensionality with neuron number may seem unbounded. This offers new insights for interpreting large-scale neural activity data recorded under various techniques.

*Manley et al., 2024* found that, unlike in our study, neural activity dimensionality in head-fixed, spontaneously behaving mice did not saturate. They used shared variance component analysis (SVCA) and noted that PCA-based estimates often show dimensionality saturation, which is consistent with our findings. We intentionally chose PCA in our study for several reasons. First, PCA is a trusted and widely used method in neuroscience, proven to uncover meaningful patterns in neural data. Second, its mathematical properties are well understood, making it particularly suitable for our theoretical analysis. Although newer methods such as SVCA might offer valuable insights, we believe PCA remains the most appropriate method for our research questions.

It is important to note that the scale invariance of dimensionality and covariance spectrum are distinct phenomena with different underlying requirements. Dimensionality invariance relies on finite

mean square covariance, causing saturation at large sample sizes. In contrast, spectral invariance requires a slow-decaying correlation kernel (small $\mu$) and/or a high-dimensional functional space (large $d$). Although both features appear in our data, they result from distinct mechanisms. A neural system could show saturating dimensionality without spectral invariance if it has finite mean square covariance but rapidly decaying correlations with functional distance. Understanding these requirements clarifies how neural organization affects different scale-invariant properties.

## Computational benefits of a scale-invariant covariance spectrum

Our findings are validated across multiple datasets obtained through various recording techniques and animal models, ranging from single-neuron calcium imaging in larval zebrafish to single-neuron multi-electrode recordings in the mouse brain (see *Figure 2—figure supplement 2*). The conclusion remains robust when the multi-electrode recording data are reanalyzed under different sampling rates (6–24 Hz, *Figure 2—figure supplement 4*). We also confirm that substituting a few negative covariances with zero retains the spectrum of the data covariance matrix (*Figure 2—figure supplement 3* and Methods).

The scale invariance of neural activity across different neuron assembly sizes could support efficient multiscale information encoding and processing. This indicates that the neural code is robust and requires minimal adjustments despite changes in population size. One recent study shows that randomly sampled and coarse-grained macrovoxels can predict population neural activity (*Hoffmann et al., 2023*), reinforcing that a random neuron subset may capture overall activity patterns. This enables downstream circuits to readout and process information through random projections (*Gao et al., 2017*). A recent study demonstrates that a scale-invariant noise covariance spectrum with a specific slope $\alpha < 1$ enables neurons to convey unlimited stimulus information as the population size increases (*Moosavi et al., 2024*). The linear Fisher information, in this context, grows at least as $N^{1-\alpha}$.

Understanding how dimensionality and spectrum change with sample size also suggests the possibility of extrapolating from small samples to overcome experimental limitations. This is particularly feasible when $\mu/d \to 0$, where the dimensionality and spectrum under anatomical, random, and functional sampling coincide (*Equations 3 and 4*). Developing extrapolation methods and exploring the benefits of scale-invariant neural code are promising future research directions.

# Materials and methods

**Key resources table**

| Reagent type (species) or resource | Designation | Source or reference | Identifiers | Additional information |
|---|---|---|---|---|
| Strain, strain background (*Danio rerio*) | Tg(elavl3: H2B-GCaMP6f) | https://doi.org/10.7554/eLife.12741 | | Jiu-Lin Du, Institute of Neuroscience, Chinese Academy of Sciences, Shanghai |
| Software, algorithm | julia1.7 | https://julialang.org/ | | |
| Software, algorithm | MATLAB | https://ww2.mathworks.cn/ | | |
| Software, algorithm | Mathematica | https://www.wolfram.com/mathematica/ | | |

## Experimental methods

The handling and care of the zebrafish complied with the guidelines and regulations of the Animal Resources Center of the University of Science and Technology of China (USTC). All larval zebrafish (huc:h2b -GCaMP6f *Cong et al., 2017*) were raised in E2 embryo medium (comprising 7.5 mM NaCl, 0.25 mM KCl, 0.5 mM MgSO$_4$, 0.075 mM KH$_2$PO$_4$, 0.025 mM Na$_2$HPO$_4$, 0.5 mM CaCl$_2$, and 0.35 mM NaHCO$_3$; containing 0.5 mg/l methylene blue) at 28.5°C and with a 14-hr light and 10-hr dark cycle.

To induce hunting behavior (composed of motor sequences like eye convergence and J turn) in larval zebrafish, we fed them a large amount of paramecia over a period of 4–5 days post-fertilization (dpf). The animals were then subjected to a 24-hr starvation period, after which they were transferred to a specialized experimental chamber. The experimental chamber was 20 mm in diameter and 1 mm in depth, and the head of each zebrafish was immobilized by applying 2% low melting point agarose. The careful removal of the agarose from the eyes and tail of the fish ensured that these body regions remained free to move during hunting behavior. Thus, characteristic behavioral features such

as J-turns and eye convergence could be observed and analyzed. Subsequently, the zebrafish were transferred to an incubator and stayed overnight. At 7 dpf, several paramecia were introduced in front of the previously immobilized animals, each of which was monitored by a stereomicroscope. Those displaying binocular convergence were selected for subsequent $Ca^{2+}$ imaging experiments.

We developed a novel optomagnetic system that allows (1) precise control of the trajectory of the paramecium and (2) imaging brain-wide $Ca^{2+}$ activity during the hunting behavior of zebrafish. To control the movement of the paramecium, we treated these microorganisms with a suspension of ferric tetroxide for 30 min and selected those that responded to its magnetic attraction. A magnetic paramecium was then placed in front of a selected larva, and its movement was controlled by changing the magnetic field generated by Helmholtz coils that were integrated into the imaging system. The real-time position of the paramecium, captured by an infrared camera, was identified by online image processing. The positional vector relative to a predetermined target position was calculated. The magnitude and direction of the current in the Helmholtz coils were adjusted accordingly, allowing for precise control of the magnetic field and hence the movement of the paramecium. Multiple target positions could be set to drive the paramecium back and forth between multiple locations.

The experimental setup consisted of head-fixed larval zebrafish undergoing two different types of behavior: induced hunting behavior by a moving paramecium in front of a fish (fish 1–5), and spontaneous behavior without any visual stimulus as a control (fish 6). Experiments were carried out at ambient temperature (ranging from 23 to 25°C). The behavior of the zebrafish was monitored by a high-speed infrared camera (Basler acA2000-165umNIR, 0.66×) behind a 4F optical system and recorded at 50 Hz. Brain-wide $Ca^{2+}$ imaging was achieved using XLFM. Light-field images were acquired at 10 Hz, using customized LabVIEW software (National Instruments, USA) or Solis software (Oxford Instruments, UK), with the help of a high-speed data acquisition card (PCIe-6321, National Instruments, USA) to synchronize the fluorescence with behavioral imaging.

## Behavior analysis

The background of each behavior video was removed using the clone stamp tool in Adobe Photoshop CS6. Individual images were then processed by an adaptive thresholding algorithm, and fish head and yolk were selected manually to determine the head orientation. The entire body centerline, extending from head to tail, was divided into 20 segments. The amplitude of a bending segment was defined as the angle between the segment and the head orientation. To identify the paramecium in a noisy environment, we subtracted a background image, averaged over a time window of 100 s, from all the frames. The major axis of the left or right eye was identified using DeepLabCut (*Mathis et al., 2018*). The eye orientation was defined as the angle between the rostrocaudal axis and the major axis of an eye. The convergence angle was defined as the angle between the major axes of the left and right eyes. An eye-convergence event was defined as a period of time where the angle between the long axis of the eyes stayed above 50° (*Bianco et al., 2011*).

## Imaging data acquisition and processing

We used a fast eXtended light-field microscope (XLFM, with a volume rate of 10 Hz) to record $Ca^{2+}$ activity throughout the brain of head-fixed larval zebrafish. Fish were ordered by the dates of experiments. As previously described (*Cong et al., 2017*), we adopted the Richardson–Lucy deconvolution method to iteratively reconstruct 3D fluorescence stacks (600 × 600 × 250) from the acquired 2D images (2048 × 2048). This algorithm requires an experimentally measured point spread function of the XLFM system. The entire recording for each fish is 15.3 ± 4.3 min (mean ± SD).

To perform image registration and segmentation, we first cropped and resized the original image stack to 400 × 308 × 210, which corresponded to the size of a standard zebrafish brain (zbb) atlas (*Tabor et al., 2019*). This step aimed to reduce substantial memory requirements and computational costs in subsequent operations. Next, we picked a typical volume frame and aligned it with the zbb atlas using a basic 3D affine transformation. This transformed frame was used as a template. We aligned each volume with the template using rigid 3D intensity-based registration (*Studholme et al., 1997*) and non-rigid pairwise registration (*Rueckert et al., 1999*) in the Computational Morphometry Toolkit (CMTK) (https://www.nitrc.org/projects/cmtk/). After voxel registration, we computed the pairwise correlation between nearby voxel intensities and performed the watershed algorithm on the correlation map to cluster and segment voxels into consistent ROIs across all volumes. We defined

**Table 2.** Resources for additional experimental datasets.

| Dataset | Data reference |
| --- | --- |
| Light-sheet imaging of larval zebrafish (*Chen et al., 2018*) | https://janelia.figshare.com/articles/dataset/Whole-brain_light-sheet_imaging_data/7272617 |
| Neuropixels recordings in mice (*Stringer et al., 2019b*) | https://janelia.figshare.com/articles/dataset/Eight-probe_Neuropixels_recordings_during_spontaneous_behaviors/7739750 |
| Two-photon imaging in mice (*Stringer et al., 2019b*) | https://janelia.figshare.com/articles/dataset/Recordings_of_ten_thousand_neurons_in_visual_cortex_during_spontaneous_behaviors/6163622 |

the diameter of each ROI using the maximum Feret diameter (the longest distance between any two voxels within a single ROI).

Finally, we adopted the 'OASIS' deconvolution method to denoise and infer neural activity from the fluorescence time sequence (*Friedrich et al., 2017*). The deconvolved $\Delta F/F$ of each ROI was used to infer firing rates for subsequent analysis.

## Other experimental datasets analyzed

To validate our findings across different recording methods and animal models, we also analyzed three additional datasets (*Table 2*). We include a brief description below for completeness. Further details can be found in the respective reference. The first dataset includes whole-brain light-sheet $Ca^{2+}$ imaging of immobilized larval zebrafish in the presence of visual stimuli as well as in a spontaneous state (*Chen et al., 2018*). Each volume of the brain was scanned through $2.11 \pm 0.21$ planes per second, providing a near-simultaneous readout of neuronal $Ca^{2+}$ signals. We analyzed fish 8 (69,207 neurons × 7890 frames), 9 (79,704 neurons × 7720 frames), and 11 (101,729 neurons × 8528 frames), which are the first three fish data with more than 7200 frames. For simplicity, we labeled them l2, l3, and l1(fl). The second dataset consists of Neuropixels recordings from approximately ten different brain areas in mice during spontaneous behavior (*Stringer et al., 2019b*). Data from the three mice, *Kerbs*, *Robbins*, and *Waksman*, include the firing rate matrices of 1462 neurons × 39,053 frames, 2296 neurons × 66,409 frames, and 2688 neurons × 74,368 frames, respectively. The last dataset comprises two-photon $Ca^{2+}$ imaging data (2–3 Hz) obtained from the visual cortex of mice during spontaneous behavior. While this dataset includes numerous animals, we focused on the first three animals that exhibited spontaneous behavior. While this dataset includes numerous animals, we focused on the first three animals that exhibited spontaneous behavior:spont_M150824_MP019_2016-04-05 (11,983 neurons × 21,055 frames), spont_M160825_MP027_2016-12-12 (11,624 neurons × 23,259 frames), and spont_M160907_MP028_2016-09-26 (9392 neurons × 10,301 frames) (*Stringer et al., 2019b*).

## Covariance matrix, eigenspectrum, and sampling procedures

To begin, we multiplied the inferred firing rate of each neuron (see Methods) by a constant such that in the resulting activity trace $a_i$, the mean of $a_i(t)$ *over the nonzero time frames* equaled one (*Meshulam et al., 2019*). Consistent with the literature (*Meshulam et al., 2019*), this step aimed to eliminate possible confounding factors in the raw activity traces, such as the heterogeneous expression level of the fluorescence protein within neurons and the nonlinear conversion of the electrical signal to $Ca^{2+}$ concentration. Note that after this scaling, neurons could still have different activity levels characterized by the variance of $a_i(t)$ over time, due to differences in the sparsity of activity (proportion of nonzero frames) and the distribution of nonzero $a_i(t)$ values. Without normalization, the covariance matrix becomes nearly diagonal, causing significant underestimation of the covariance structures.

The three models of covariance in *Figure 2G–I* were constructed as follows. For model in *Figure 2G*, the entries of matrix $G$ (with dimensions $N \times T$) were sampled from an i.i.d. Gaussian distribution with zero mean and standard deviation $\sigma = 1$. In *Figure 2H*, we constructed the composite covariance matrix for fish 1 achieved by maintaining the eigenvalues from the fish 1 data covariance matrix and replacing the eigenvectors $U$ with a set of random orthonormal basis. Lastly, the covariance matrix in *Figure 2I* was generated from a randomly connected recurrent network of linear rate neurons. The entries in the synaptic weight matrix are normally distributed with $J_{ij} \sim \mathcal{N}(0, g^2/N)$, with a coupling strength $g = 0.95$ (*Hu and Sompolinsky, 2022*; *Morales et al., 2023*). For consistency, we used the same number of time frames $T = 7200$ when comparing CI across all the datasets (*Figures 4B, C and*

*5D, E*, *Figure 4—figure supplement 2C*). For other cases, we analyzed the full length of the data (number of time frames: fish 1 – 7495, fish 2 – 9774, fish 3 – 13,904, fish 4 – 7318, fish 5 – 7200, and fish 6 – 9388). Next, the covariance matrix was calculated as $C_{ij} = \frac{1}{T-1} \sum_{t=1}^{T} \left( a_i(t) - \bar{a}_i \right) \left( a_j(t) - \bar{a}_j \right)$, where $\bar{a}_i$ is the mean of $a_i(t)$ over time. Finally, to visualize covariance matrices on a common scale, we multiplied matrix $C$ by a constant such that the average of its diagonal entries equaled one, that is, $\mathrm{Tr}(C)/N = 1$. This scaling did not alter the shape of covariance eigenvalue distribution, but set the mean at 1 (see also *Equation 8*).

To maintain consistency across datasets, we fixed the same initial number of neurons at $N_0 = 1,024$. These $N_0$ neurons were randomly chosen once for each zebrafish dataset and then used throughout the subsequent analyses. We adopted this setting for all analyses except in two particular instances: (1) for comparisons among the three sampling methods (RSap, ASap, and FSap), we specifically chose 1024 neurons centered along the anterior–posterior axis, mainly from the midbrain to the anterior hindbrain regions (*Figure 5DE*, *Figure 5—figure supplement 6*). (2) When investigating the impact of hunting behavior on scale invariance, we included the entire neuronal population (Appendix 1).

We used an iterative procedure to sample the covariance matrix $C$ (calculated from data or as simulated ERMs). For RSap, in the first iteration, we randomly selected half of the neurons. The covariance matrix for these selected neurons was a $N/2 \times N/2$ diagonal block of $C$. Similarly, the covariance matrix of the unselected neurons was another diagonal block of the same size. In the next iteration, we similarly created two new sampled blocks with half the number of neurons for each of the blocks we had. Repeating this process for $n$ iterations resulted in $2^n$ blocks, each containing $N := N_0/2^n$ neurons. At each iteration, the eigenvalues of each block were calculated and averaged across the blocks after being sorted in descending order. Finally, the averaged eigenvalues were plotted against rank/$N$ on a log–log scale.

In the case of ASap and FSap, the process of selecting neurons was different, although the remaining procedures followed the RSap protocol. In ASap, the selection of neurons was based on a spatial criterion: neurons close to the anterior end on the anterior–posterior axis were grouped to create a diagonal block of size $\frac{N}{2} \times \frac{N}{2}$, with the remaining neurons forming a separate block. FSap, on the other hand, used the RG framework (*Meshulam et al., 2019*) to define the blocks (details in Appendix 1). In each iteration, the cluster of neurons within a block that showed the highest average correlation ($\mathrm{E}_{i \neq j}(C_{ij}^2)$) was identified and labeled as the most correlated cluster (refer to *Figure 5D*, *Figure 5—figure supplement 6*, and *Figure 5—figure supplement 7*).

In the ERM model, as part of implementing ASap, we generated anatomical and functional coordinates for neurons with a specified CCA properties as described in Methods. Mirroring the approach taken with our data, ASap segmented neurons into groups based on the first dimension of their anatomical coordinates, akin to the anterier–posterior axis. FSap employed the same RG procedures outlined earlier (Appendix 1).

To determine the overall power-law coefficient of the eigenspectra, $\alpha$, throughout sampling, we fitted a straight line in the log–log rank plot to the large eigenvalues that combined the original and three iterations of sampled covariance matrices (selecting the top 10% eigenvalues for each matrix and excluding the first four largest ones for each matrix). We averaged the estimated $\alpha$ over 10 repetitions of the entire sampling procedure. $R^2$ of the power-law fit was computed in a similar way. To visualize the statistical structures of the original and sampled covariance matrices, the orders of the neurons (i.e., columns and rows) are determined by the following algorithm. We first construct a symmetric Toeplitz matrix $\mathcal{T}$, with entries $\mathcal{T}_{i,j} = t_{i-j}$ and $t_{i-j} \equiv t_{j-i}$. The vector $\vec{t} = [t_0, t_1, \ldots, t_{N-1}]$ is equal to the mean covariance vector of each neuron calculated below. Let $\vec{c}_i$ be a row vector of the data covariance matrix; we identify $\vec{t} = \frac{1}{N} \sum_{i=1}^{N} D(\vec{c}_i)$, where $D(\cdot)$ denotes a numerical ordering operator, namely rearranging the elements in a vector $\vec{c}$ such that $c_0 \geq c_1 \geq \ldots \geq c_{N-1}$. The second step is to find a permutation matrix $P$ such that $\|\mathcal{T} - PCP^T\|_F$ is minimized, where $\| \ \|_F$ denotes the Frobenius norm. This quadratic assignment problem is solved by simulated annealing. Note that after sampling, the smaller matrix will appear different from the larger one. We need to perform the above reordering algorithm for every sampled matrix so that matrices of different sizes become similar in *Figure 2E*.

The composite covariance matrix with substituted eigenvectors in *Figure 2H* was created as described in the following steps. First, we generated a random orthogonal matrix $U_r$ (based on the Haar measure) for the new eigenvectors. This was achieved by QR decomposition $A = U_r R$ of a random matrix $A$ with i.i.d. entries $A_{ij} \sim \mathcal{N}(0, 1/N)$. The composite covariance matrix $C_r$ was then

defined as $C_r := U_r \Lambda U_r^T$, where $\Lambda$ is a diagonal matrix that contains the eigenvalues of $C$. Note that since all the eigenvalues are real and $U_r$ is orthogonal, the resulting $C_r$ is a real and symmetric matrix. By construction, $C_r$ and $C$ have the same eigenvalues, but their *sampled* eigenspectra can differ.

## Dimensionality

In this section, we introduce the participation ratio ($D_{\mathrm{PR}}$) as a metric for effective dimensionality of a system, based on *Recanatesi et al., 2019*; *Litwin-Kumar et al., 2017*; *Gao and Ganguli, 2015*; *Gao et al., 2017*; *Clark et al., 2023*; *Dahmen et al., 2020*. $D_{\mathrm{PR}}$ is defined as:

$$D_{\mathrm{PR}}(C) = \frac{\left(\sum_i \lambda_i\right)^2}{\sum_i \lambda_i^2} = \frac{\left(\mathrm{Tr}(C)\right)^2}{\mathrm{Tr}(C^2)} = \frac{N^2 \mathrm{E}(\sigma^2)^2}{N\mathrm{E}(\sigma^4) + N(N-1)\mathrm{E}_{i \neq j}(C_{ij}^2)} \tag{5}$$

Here, $\lambda_i$ are the eigenvalues of the covariance matrix $C$, representing variances of neural activities. $\mathrm{Tr}(\cdot)$ denotes the trace of the matrix. The term $\mathrm{E}_{i \neq j}(C_{ij}^2)$ denotes the expected value of the squared elements that lie off the main diagonal of $C$. This represents the average squared covariance between the activities of distinct pairs of neurons.

With these definitions, we explore the asymptotic behavior of $D_{\mathrm{PR}}$ as the number of neurons $N$ approaches infinity:

$$\lim_{N \to \infty} D_{\mathrm{PR}}(C) = \frac{\mathrm{E}(\sigma^2)^2}{\mathrm{E}_{i \neq j}(C_{ij}^2)}$$

This limit highlights the relationship between the PR dimension and the average squared covariance among different pairs of neurons. To predict how $D_{\mathrm{PR}}$ scales with the number of neurons (*Figure 2D*), we first estimated these statistical quantities ($\mathrm{E}_{i \neq j}(C_{ij}^2)$, $\mathrm{E}(\sigma^2)$, and $\mathrm{E}(\sigma^4)$) using all available neurons, then applied *Equation 5* for different values of $N$. It is worth mentioning that a similar theoretical finding is established by *Dahmen et al., 2020*. The transition from increasing $D_{\mathrm{PR}}$ with $N$ to approaching the saturation point occurs when $N$ is significantly larger than $D_{\mathrm{PR}}$.

## ERM model

We consider the eigenvalue distribution or spectrum of the matrix $C$ at the limit of $N \gg 1$ and $L \gg 1$. This spectrum can be analytically calculated in both high- and intermediate-density scenarios using the replica method (*Mézard et al., 1999*). The following sketch shows our approach, and detailed derivations can be found in Appendix 2. To calculate the probability density function of the eigenvalues (or eigendensity), we first compute the resolvent or Stieltjes transform $g(z) = -\frac{2}{N} \partial_z \left\langle \ln \det(zI - C)^{-1/2} \right\rangle$, $z \in \mathbb{C}$. Here, $\langle \ldots \rangle$ is the average across the realizations of $C$ (i.e., random $\vec{x}_i$'s and $\sigma_i^{2'}$'s). The relationship between the resolvent and the eigendensity is given by the Sokhotski–Plemelj formula:

$$p(\lambda) = -\frac{1}{\pi} \lim_{\eta \to 0^+} \mathbf{Im}\, g(\lambda + i\eta), \tag{6}$$

where $\mathbf{Im}$ means imaginary part.

Here we follow the field-theoretic approach (*Mézard et al., 1999*), which turns the problem of calculating the resolvent to a calculation of the partition function in statistical physics by using the replica method. In the limit $N \to \infty$, $L^d \to \infty$, $\rho$ being finite, by performing a leading order expansion of the canonical partition function at large $z$ (Appendix 2), we find the resolvent is given by

$$g(z) = \frac{1}{\rho} \int \frac{\mathrm{d}^d \vec{k}}{(2\pi)^d} \frac{1}{z - \rho \mathrm{E}(\sigma^2)\tilde{f}(\vec{k})} \tag{7}$$

In the *high-density* regime, the probability density function (pdf) of the covariance eigenvalues can be approximated and expressed from *Equations 6 and 7* using the Fourier transform of the kernel function $\tilde{f}(\vec{k})$:

$$p(\lambda) = \frac{1}{\rho \mathrm{E}(\sigma^2)} \int_{\mathbb{R}^d} \frac{\mathrm{d}^d \vec{k}}{(2\pi)^d} \delta \left( \frac{\lambda}{\mathrm{E}(\sigma^2)} - \rho \tilde{f}(\vec{k}) \right), \tag{8}$$

where $\delta(x)$ is the Dirac delta function and $\mathrm{E}(\sigma^2)$ is the expected value of the variances of neural activity. Intuitively, *Equation 8* means that $\lambda/\rho$ are distributed with a density proportional to the area of $\tilde{f}(\vec{k})'$ level sets (i.e., isosurfaces).

In Results, we found that the covariance matrix consistently shows greater scale invariance compared to the correlation matrix across all datasets. This suggests that the variability in neuronal activity significantly influences the eigenspectrum. This finding, however, cannot be explained by the high-density theory, which predicts that the eigenspectrum of the covariance matrix is simply a rescaling of the correlation eigenspectrum by $\mathrm{E}(\sigma_i^2)$, the expected value of the variances of neural activity. Without loss of generality, we can always standardize the fluctuation level of neural activity by setting $\mathrm{E}(\sigma^2) = 1$. This is equivalent to multiplying the covariance matrix $C$ by a constant such that $\mathrm{Tr}(C)/N = 1$, which in turn scales all the eigenvalues of $C$ by the same factor. Consequently, the heterogeneity of $\sigma_i^2$ has no effect on the scale invariance of the eigenspectrum (see *Equation 8*). This theoretical prediction is indeed correct and is confirmed by direct numerical simulations and quantifying the scale invariance using the CI (*Figure 4—figure supplement 2A*).

Fortunately, the inconsistency between theory and experimental results can be resolved by focusing the ERM within the intermediate density regime $\rho\epsilon^d \ll 1$, where neurons are positioned at a moderate distance from each other. As mentioned above, we set $\mathrm{E}(\sigma^2) = 1$ in our model and vary the diversity of activity fluctuations among neurons represented by $\mathrm{E}(\sigma^4)$. Consistent with the experimental observations, we find that the CI decreases with $\mathrm{E}(\sigma^4)$ (see *Figure 4—figure supplement 2B*). This agreement indicates that the neural data are better explained by the ERM in the intermediate density regime.

To gain a deeper understanding of this behavior, we use the Gaussian variational method (*Mézard et al., 1999*) to calculate the eigenspectrum. Unlike the high-density theory where the eigendensity has an explicit expression, in the intermediate density the resolvent $g(z)$ no longer has an explicit expression and is given by the following equation:

$$g(z) = \left\langle \frac{1}{z - \sigma^2 \int \mathrm{D}\vec{k}\,\tilde{G}(\vec{k}, z)} \right\rangle_\sigma , \qquad (9)$$

where $\langle \ldots \rangle_\sigma$ computes the expectation value of the term within the bracket with respect to σ, namely $\langle \ldots \rangle_\sigma \equiv \int \ldots p(\sigma)\mathrm{d}\sigma$. Here and in the following, we denote $\int \mathrm{D}\vec{k} \equiv \int \frac{\mathrm{d}^d \vec{k}}{(2\pi)^d}$. The function $G(\vec{k}, z)$ is determined by a self-consistent equation,

$$\frac{1}{\tilde{f}(\vec{k})} = \frac{1}{\tilde{G}(\vec{k}, z)} + \left\langle \frac{\rho\sigma^2}{z - \sigma^2 \int \mathrm{D}\vec{k}\,\tilde{G}(\vec{k}, z)} \right\rangle_\sigma \qquad (10)$$

We can solve $\int \mathrm{D}\vec{k}\,G(\vec{k}, z)$ from *Equation 10* numerically and below is an outline, and the details are explained in Appendix 2. Let us define the integral $\mathcal{G} \equiv \int \mathrm{D}\vec{k}\,\tilde{G}(\vec{k}, z)$. First, we substitute $z \equiv \lambda + i\eta$ into *Equation 10* and write $\mathcal{G} = \mathbf{Re}\mathcal{G} + i\mathbf{Im}\mathcal{G}$. *Equation 10* can thus be decomposed into its real part and imaginary part, and a set of nonlinear and integral equations, each of which involves both $\mathbf{Re}\mathcal{G}$ and $\mathbf{Im}\mathcal{G}$. We solve these equations at the limit $\eta \to 0$ using a fixed-point iteration that alternates between updating $\mathbf{Re}\mathcal{G}$ and $\mathbf{Im}\mathcal{G}$ until convergence.

We find that the variational approximations exhibit excellent agreement with the numerical simulation for both large and intermediate $\rho$ where the high-density theory starts to deviate significantly (for $\rho = 256$ and $\rho = 10.24$, $\epsilon = 0.03125$, *Figure 4—figure supplement 1*). Note that the departure of the leading eigenvalues in these plots is expected, since the power-law kernel function we use is not integrable (see Methods).

To elucidate the connection between the two different methods, we estimate the condition when the result of the high-density theory (*Equation 8*) matches that of the variational method (*Equations 9 and 10*; Appendix 2). The transition between these two density regimes can also be understood (see *Equation 22* and Appendix 2).

Importantly, the scale invariance of the spectrum at $\mu/d \to 0$ previously derived using the high-density result (*Equation 3*) can be extended to the intermediate-density regime by proving the $\rho$-independence using the variational method (Appendix 2).

Finally, using the variational method and the integration limit estimated by simulation (see Methods), we show that the heterogeneity of the variance of neural activity, quantified by $\mathrm{E}(\sigma^4)$, indeed improves

**Table 3.** Modifications of the shape of $f(\vec{x})$ near $\|\vec{x}\| = 0$ used in *Appendix 1—figures 1–3*.
Flat: when $\|\vec{x}\| < \epsilon$, $f(\vec{x}) = 1$. Tangent: when $\|\vec{x}\| < c\epsilon$, $f(\vec{x})$ follows a tangent line of the exact power law ($b\|\vec{x}\| + 1$ and $\frac{\epsilon^\mu}{\|\vec{x}\|^\mu}$ have a same first-order derivative when $\|\vec{x}\| = c\epsilon$). $b$ and $c$ are constants. Tent: when $\|\vec{x}\| < c\epsilon$, $f(\vec{x})$ follows a straight line while the slope is not the same as the tangent case. Parabola: when $\|\vec{x}\| < c\epsilon$, $f(\vec{x})$ follows a quadratic function ($ax^2 + 1$ and $\frac{\epsilon^\mu}{\|\vec{x}\|^\mu}$ have same first-order derivative). t pdf: mimic the smoothing treatment like the t distribution. All the constant parameters are set such that $f(0) = 1$.

| $f(\vec{x})$ | Definition |
| --- | --- |
| Flat | $f(\vec{x}) = \begin{cases} 1, & \|\vec{x}\| < \epsilon \\ \frac{e^\mu}{\|\vec{x}\|^\mu}, & \|\vec{x}\| \geq \epsilon \end{cases}$ |
| Tangent | $f(\vec{x}) = \begin{cases} b\|\vec{x}\| + 1, & \|\vec{x}\| < c\epsilon, f'(c\epsilon) = b \\ \frac{e^\mu}{\|\vec{x}\|^\mu}, & \|\vec{x}\| \geq c\epsilon \end{cases}$ |
| Tent | $f(\vec{x}) = \begin{cases} b\|\vec{x}\| + 1, & \|\vec{x}\| < c\epsilon, f'(c\epsilon) \neq b \\ \frac{e^\mu}{\|\vec{x}\|^\mu}, & \|\vec{x}\| \geq c\epsilon \end{cases}$ |
| Parabola | $f(\vec{x}) = \begin{cases} b\|\vec{x}\|^2 + 1, & \|\vec{x}\| < c\epsilon, f'(c\epsilon) = 2bc\epsilon \\ \frac{e^\mu}{\|\vec{x}\|^\mu}, & \|\vec{x}\| \geq c\epsilon \end{cases}$ |
| t pdf | $f(\vec{x}) = \varepsilon^\mu(\varepsilon^2 + \|\vec{x}\|^2)^{-\mu/2}$ |

the collapse of the eigenspectra for intermediate $\rho$ (Appendix 2). Our theoretical results agree excellently with the ERM simulation (*Figure 4—figure supplement 2A, B*).

## Kernel function

Throughout the paper, we have mainly considered a particular approximate power-law kernel function inspired by the Student's *t* distribution

$$f(\vec{x}) = \epsilon^\mu(\epsilon^2 + \|\vec{x}\|^2)^{-\mu/2}. \tag{11}$$

To understand how to choose $\epsilon$ and $\mu$, see Methods. Variations of *Equation 11* near $x = 0$ have also been explored; see a summary in *Table 3*.

It is worth mentioning that a power law is not the only slow-decaying function that can produce a scale-invariant covariance spectrum (*Figure 3—figure supplement 2*). We choose it for its analytical tractability in calculating the eigenspectrum. Importantly, we find numerically that the two contributing factors to scale invariance – namely, slow spatial decay and higher functional space – can be generalized to other *nonpower-law* functions. An example is the stretched exponential function $f(\vec{x}) = e^{-\|\vec{x}\|^\eta}$ with $0 < \eta < 1$. When $\eta$ is small and $d$ is large, the covariance eigenspectra also display a similar collapse upon random sampling (*Figure 3—figure supplement 2*).

This approximate power-law $f(\vec{x})$ has the advantage of having an analytical expression for its Fourier transform, which is crucial for the high-density theory (*Equation 8*),

$$f(\vec{k}) = \frac{2^{\frac{d-\mu+2}{2}} \pi^{\frac{d}{2}} k^{\frac{\mu-d}{2}} \epsilon^{\frac{\mu+d}{2}} K_{(d-\mu)/2}(k\epsilon)}{\Gamma(\mu/2)}, \quad k = \|\vec{k}\| \tag{12}$$

Here, $K_\alpha(x)$ is the modified Bessel function of the second kind, and $\Gamma(x)$ is the Gamma function. We calculated the above formulas analytically for $d = 1, 2, 3$ with the assistance of Mathematica and conjectured the case for general dimension $d$, which we confirmed numerically for $d \leq 10$.

We want to explain two technical points relevant to the interpretation of our numerical results and the choice of $f(\vec{x})$. Unlike the case in the usual ERM, here we allow $f(\vec{x})$ to be non-integrable (over $\mathbb{R}^d$), which is crucial to allow power law $f(\vec{x})$. The nonintegrability violates a condition in the classical convergence results of the ERM spectrum (*Bordenave, 2008*) as $N \to \infty$. We believe that this is exactly the reason for the departure of the first few eigenvalues from our theoretical spectrum (e.g., in *Figure 3*).

Our hypothesis is also supported by ERM simulations with integrable $f(\vec{x})$ (**Figure 3—figure supplement 1**), where the numerical eigenspectrum matches closely with our theoretical one, including the leading eigenvalues. For ERM to be a legitimate model for covariance matrices, we need to ensure that the resulting matrix $C$ is positive semidefinite. According to the Bochner theorem (**Rudin, 1990**), this is equivalent to the Fourier transform (FT) of the kernel function $\tilde{f}(\vec{k})$ being nonnegative for all frequencies. For example, in 1D, a rectangle function $\text{rect}(x) = \begin{cases} 1, & \text{if } |x| \leq \frac{1}{2} \\ 0, & \text{otherwise} \end{cases}$ does not meet the condition (its FT is $\text{sinc}(x) = \frac{\sin(x)}{x}$), but a tent function $\text{tent}(x) = \begin{cases} 1 - |x|, & \text{if } |x| \leq 1 \\ 0, & \text{otherwise} \end{cases}$ does (its FT is $\text{sinc}^2(x)$). For the particular kernel function $f(\vec{x})$ in **Equation 11**, this condition can be easily verified using the analytical expressions of its Fourier transform (**Equation 12**). The integral expression for $K_\alpha(x)$, given as $K_\alpha(x) = \int_0^\infty e^{-x\cosh t} \cosh(\alpha t)dt$, shows that $K_\alpha(x)$ is positive for all $x > 0$. Likewise, the Gamma function $\Gamma(x) > 0$. Therefore, the Fourier transform of **Equation 11** is positive and the resulting matrix $C$ (of any size and values of $\vec{x}_i$) is guaranteed to be positive definite.

Building upon the theory outlined above, numerical simulations further validated the empirical robustness of our ERM model, as showcased in **Figures 3B–D and 4A**. In **Figure 3B–D**, the ERM was characterized by the parameters $N = 1024$, $d = 2$, $L = 10$, $\rho = 10.24$, and $\mu = 0.5$ and $\epsilon = 0.03125$ for $f(\vec{x})$. To numerically compute the eigenvalue probability density function, we generated the ERM 100 times, each sampled using the method described in Methods. The pdf was computed by calculating the pdf of each ERM realization and averaging these across the instances. The curves in **Figure 3D** showed the average of over 100 ERM simulations. The shaded area (most of which is smaller than the marker size) represented the SEM. For **Figure 4A**, the columns from left to right were corresponded to $\mu = 0.5, 0.9, 1.3$, and the rows from top to bottom were corresponded to $d = 1, 2, 3$. Other ERM simulation parameters: $N = 4096$, $\rho = 256$, $L = (N/\rho)^{1/d}$, $\epsilon = 0.03125$, and $\sigma_i^2 = 1$. It should be noted that for **Figure 4A**, the presented data pertain to a single ERM realization.

## Collapse index

We quantify the extent of scale invariance using CI defined as the area between two spectrum curves (**Figure 4A**, upper right), providing an intuitive measure of the shift of the eigenspectrum when varying the number of sampled neurons. We chose the CI over other measures of distance between distributions for several reasons. First, it directly quantifies the shift of the eigenspectrum, providing a clear and interpretable measure of scale invariance. Second, unlike methods that rely on estimating the full distribution, the CI avoids potential inaccuracies in estimating the probability of the top leading eigenvalues. Finally, the use of CI is motivated by theoretical considerations, namely the ERM in the high-density regime, which provides an analytical expression for the covariance spectrum (**Equation 3**) valid for large eigenvalues.

$$\text{CI} := \frac{1}{\log(q_0/q_1)} \int_{\log q_1}^{\log q_0} \left| \frac{\partial \log \lambda(q)}{\partial \log \rho} \right| d \log q, \tag{13}$$

we set $q_1$ such that $\lambda(q_1) = 1$, which is the mean of the eigenvalues of a normalized covariance matrix. The other integration limit $q_0$ is set to 0.01 such that $\lambda(q_0)$ is the 1% largest eigenvalue.

Here, we provide numerical details on calculating CI for the ERM simulations and experimental data.

### A calculation of CI for experimental datasets/ERM model

To calculate CI for a covariance matrix $C$ of size $N_0$, we first computed its eigenvalues $\lambda_i^0$ and those of the sampled block $C_s$ of size $N_s = N_0/2$, denoted as $\lambda_i^s$ (averaged over 20 times for the ERM simulation and 2000 times in experimental data). Next, we estimated $\log \lambda(q)$ using the eigenvalues of $C_0$ and $C_s$ at $q = i/N_s$, $i = 1, 2, \ldots, N_s$. For the sampled $C_s$, we simply had $\log \lambda(q = i/N_s) = \log \lambda_i^s$, its $i$th largest eigenvalue. For the original $C_0$, $\log \lambda(q = i/N_s)$ was estimated by a linear interpolation, on the $\log \lambda - \log q$ scale, using the value of $\log \lambda(q)$ in the nearest neighboring $q = i/N_0$'s (which again are simply $\log \lambda_i^0$). Finally, the integral (**Equation 13**) was computed using the trapezoidal rule, discretized at $q = i/N_s$'s, using the

finite difference $\frac{\partial \log \lambda(q)}{\partial \log \rho} \approx \frac{1}{\log(N_0/N_s)} |\Delta \log \lambda(q)|$, where $\Delta$ denotes the difference between the original eigenvalues of $C_0$ and those of sampled $C_s$.

## Estimating CI using the variational method

In the definition of CI (*Equation 13*), calculating $\lambda(q)$ and $\frac{\partial \log \lambda(q)}{\partial \log \rho}$ directly using the variational method is difficult, but we can make use of an implicit differentiation

$$\frac{\partial \log \lambda(q, \rho)}{\partial \log \rho} = \frac{\rho}{\lambda} \frac{\partial \lambda(q, \rho)}{\partial \rho} = -\frac{\rho}{\lambda} \frac{\dfrac{\partial q(\rho, \lambda)}{\partial \rho}}{\dfrac{\partial q(\rho, \lambda)}{\partial \lambda}}, \tag{14}$$

where $q(\lambda) := \int_\lambda^\infty p(\lambda) \mathrm{d}\lambda$ is the complementary cdf (the inverse function of $\lambda(q)$ in Methods). Using this, the integral in CI (*Equation 13*) can be rewritten as

$$\int_{\log q_1}^{\log q_0} \left| \frac{\partial \log \lambda(q, \rho)}{\partial \log \rho} \right| \mathrm{d} \log q = \int_{q_1}^{q_0} \left| -\frac{\rho}{q\lambda} \frac{\dfrac{\partial q}{\partial \rho}}{\dfrac{\partial q}{\partial \lambda}} \right| \mathrm{d}q$$

$$= \int_{\lambda(q_1)}^{\lambda(q_0)} \left| -\frac{\rho}{q\lambda} \frac{\dfrac{\partial q}{\partial \rho} y}{\dfrac{\partial q}{\partial \lambda}} \right| \frac{\partial q}{\partial \lambda} \mathrm{d}\lambda = \int_{\lambda(q_0)}^{\lambda(q_1)} \left| \frac{1}{\lambda} \frac{\partial \log q}{\partial \log \rho} \right| \mathrm{d}\lambda. \tag{15}$$

Since $\frac{\partial q}{\partial \lambda} = -p(\lambda) < 0$, we switch the order of the integration interval in the final expression of *Equation 15*.

First, we explain how to compute the complementary cdf $q(\lambda)$ numerically using the variational method. The key is to integrate the probability density function $p(\lambda)$ from $\lambda$ to a finite $\lambda(q_s)$ rather than to infinity,

$$q(\lambda) = \int_\lambda^\infty p(\lambda) \mathrm{d}\lambda = \int_{\lambda(q_s)}^\infty p(\lambda) \mathrm{d}\lambda + \int_\lambda^{\lambda(q_s)} p(\lambda) \mathrm{d}\lambda = q_s + \int_\lambda^{\lambda(q_s)} p(\lambda) \mathrm{d}\lambda. \tag{16}$$

The integration limit $\lambda(q_s)$ cannot be calculated directly using the variational method. We thus used the value of $\lambda^s(q_s \approx q_0)$ (Methods) from simulations of the ERM with a large $N = 1024$ as an approximation. Furthermore, we employed a smoothing technique to reduce bias in the estimation of $\lambda^s(q_s)$ due to the leading zigzag eigenvalues (i.e., the largest eigenvalues) of the eigenspectrum. Specifically, we determined the nearest rank $j < Nq_0$ and then smoothed the eigenvalue $\log \lambda^s(q_s)$ on the log–log scale using the formula $\log \lambda^s(q_s) = \frac{1}{3} \sum_{i=0}^2 \log \lambda^s(\frac{j+i}{N})$ and $\log q_s = \frac{1}{3} \sum_{i=0}^2 \log \frac{j+i}{N}$, averaging over 100 ERM simulations.

Note that we can alternatively use the high-density theory (Appendix 2) to compute the integration limit $\lambda(q_s = 1/N)$ instead of resorting to simulations. However, since the true value deviates from the $\lambda^h(q_s = 1/N)$ derived from high-density theory, this approach introduces a constant bias (*Figure 4—figure supplement 2*) when computing the integral in *Equation 16*. Therefore we used the simulation value $\lambda^s(q_s \approx q_0)$ when producing *Figure 4—figure supplement 2AB*.

Next, we describe how each term within the integral of *Equation 15* was numerically estimated. First, we calculated $\frac{\partial \log q}{\partial \log \rho}$ with a similar method described in Methods. Briefly, we calculated $q_0(\lambda)$ for density $\rho_0 = \frac{N_0}{L^d}$ and $q_s(\lambda)$ for density $\rho_s = \frac{N_s}{L^d}$, and then used the finite difference $\frac{1}{\log(\rho_0/\rho_s)} |\Delta \log q(\lambda)|$. Second, $\frac{\partial \log q(\lambda)}{\partial \log \rho}$ was evaluated at $\lambda = \lambda(q_1) + i \frac{\lambda(q_0) - \lambda(q_1)}{k-1}$, where $i = 0, 1, 2, \ldots, k-1$, and we used $k = 20$. Finally, we performed a cubic spline interpolation of the term $\frac{\partial \log q}{\partial \log \rho}$, and obtained the theoretical CI by an integration of *Equation 15*. *Figure 4—figure supplement 2A, B* shows a comparison between theoretical CI and that obtained by numerical simulations of ERM (Methods).

## Fitting ERM to data

### Estimating the ERM parameters

Our ERM model has four parameters: $\mu$ and $\epsilon$ dictate the kernel function $f(\vec{x})$, whereas the box size $L$ and the embedding dimension $d$ determine the neuronal density $\rho$. In the following, we describe

an approximate method to estimate these parameters from pairwise correlations measured experimentally $R_{ij} = \frac{C_{ij}}{\sigma_i \sigma_j}$. We proceed by deriving a relationship between the correlation probability density distribution $h(R)$ and the pairwise distance probability density distribution $g(u) := g(\|\vec{x}_1 - \vec{x}_2\|)$ in the functional space, from which the parameters of the ERM can be estimated.

Consider a distribution of neurons in the functional space with a coordinate distribution $p(\vec{x})$. The pairwise distance density function $g(u)$ is related to the spatial point density by the following formula:

$$g(u) = \int_{[0,L]^d} p(\vec{x}_1)p(\vec{x}_2)\delta(\|\vec{x}_1 - \vec{x}_2\| - u)\mathrm{d}\vec{x}_1\mathrm{d}\vec{x}_2 \tag{17}$$

For ease of notation, we subsequently omit the region of integration, which is the same as here. In the case of a uniform distribution, $p(\vec{x}_1) = p(\vec{x}_2) = 1/V = 1/L^d$. For other spatial distributions, *Equation 17* cannot be explicitly evaluated. We therefore make a similar approximation by focusing on a small pairwise distance (i.e., large correlation):

$$p(\vec{x}_1) \approx p(\vec{x}_2) \approx p(\frac{\vec{x}_1 + \vec{x}_2}{2}) \tag{18}$$

By a change of variables:

$$\vec{X} = \frac{\vec{x}_1 + \vec{x}_2}{2}, \quad \vec{u} = \vec{x}_1 - \vec{x}_2,$$

*Equation 17* can be rewritten as

$$g(u) \approx \int p^2(\vec{X})\delta(\|\vec{u}\| - u)\mathrm{d}\vec{X}\mathrm{d}\vec{u} = S_{d-1}(u)\int p^2(\vec{X})\mathrm{d}\vec{X} \tag{19}$$

where $S_{d-1}(u)$ is the surface area of $d-1$ sphere with radius $u$. Note that the approximation of $g(u)$ is not normalized to 1, as *Equation 19* provides an approximation valid only for small pairwise distances (i.e., large correlation). Therefore, we believe this does not pose an issue.

With the approximate power-law kernel function $R = f(u) \approx (\frac{\epsilon}{u})^\mu$, the probability density function of pairwise correlation $h(R)$ is given by:

$$h(R) = g(u)\left|\frac{\mathrm{d}u}{\mathrm{d}R}\right| = \frac{2\pi^{\frac{d}{2}}\epsilon^d}{\Gamma(\frac{d}{2})\mu R^{(\mu+d)/\mu}}\int p^2(\vec{X})\mathrm{d}\vec{X} \tag{20}$$

Taking the logarithm on both sides

$$\log h(R) = \log\left(\epsilon^d\int p^2(\vec{X})\mathrm{d}\vec{X}\right) + \log\frac{2\pi^{\frac{d}{2}}}{\Gamma(\frac{d}{2})\mu} - \frac{\mu+d}{\mu}\log R \tag{21}$$

*Equation 21* is the key formula for ERM parameters estimation. In the case of a uniform spatial distribution, $\epsilon^d\int p^2(\vec{X})\mathrm{d}\vec{X} = \epsilon^d/V = (\epsilon/L)^d$. For a given dimension $d$, we can therefore estimate $\mu$ and $(\epsilon/L)^d$ separately by fitting $h(R)$ on the log–log scale using the linear least squares. Lastly, we fit the distribution of $\sigma^2$ (the diagonal entries of the covariance matrix $C$) to a log-normal distribution by estimating the maximum likelihood.

There is a redundancy between the unit of the functional space (using a rescaled $\epsilon_\delta \equiv \epsilon/\delta$) and the unit of $f(\vec{x})$ (using a rescaled $f_\delta(\vec{x}) \equiv f(\vec{x}/\delta)$), thus $\epsilon$ and $L$ are a pair of redundant parameters: once $\epsilon$ is given, $L$ is also determined. We set $\epsilon = 0.03125$ throughout the article. In summary, for a given dimension $d$ and $\epsilon$, $\mu$ of $f(\vec{x})$ (*Equation 11*), the distribution of $\sigma^2$ and $\rho$ (or equivalently $L$) can be fitted by comparing the distribution of pairwise correlations in experimental data and ERM. Furthermore, knowing $(\epsilon/L)^d$ enables us to determine *a fundamental dimensionless parameter*

$$\rho\epsilon^d := N(\epsilon/L)^d, \tag{22}$$

which tells us whether the experimental data are better described by the high-density theory or the Gaussian variational method (Appendix 2). Indeed, the fitted $\rho\epsilon^d \sim 10^{-3} - 10^0$ is much smaller than 1, consistent with our earlier conclusion that neural data are better described by an ERM model in the intermediate-density regime.

Notably, we found that a smaller embedding dimension $d \leq 5$ gave a better fit to the overall pairwise correlation distribution. The following is an empirical explanation. As $d$ grows, to best fit the slope of $\log h(R) - \log R$, $\mu$ will also grow. However, for very high dimensions $d$, the $y$-intercept would become very negative, or equivalently, the fitted correlation would become extremely small. This can be verified by examining the leading order $\log R$ independent term in *Equation 21*, which can be approximated as $d \log \frac{\epsilon}{L} + \frac{d}{2}\left(\log \pi + 1 - \log \frac{d}{2}\right)$. It becomes very negative for large $d$ since $\epsilon \ll L$ by construction. Throughout this article, we use $d = 2$ when fitting the experimental data with our ERM model.

The above calculation can be extended to the cases where the coordinate distribution $p(\vec{x})$ becomes dependent on other parameters. To estimate the parameters in coordinate distributions that can generate ERMs with a similar pairwise correlation distribution (*Appendix 1—figure 1*), we fixed the integral value $\int p^2(\vec{x})d\vec{x}$. Consider, for example, a transformation of the uniform coordinate distribution to the normal distribution $\mathcal{N}(\mu_p = 0, \sigma_p^2 \mathbf{I})$ in $\mathbb{R}^2$. We imposed $\int p^2(\vec{x})d\vec{x} = 1/(4\pi\sigma_p^2) = 1/L^2$. For the lognormal distribution, a similar calculation led to $L\exp(\sigma_p^2/4 - \mu_p) = 2\sqrt{\pi}\sigma_p$. The numerical values for these parameters are shown in Appendix 1. However, note that due to the approximation we used (*Equation 18*), our estimate of the ERM parameters becomes less accurate if the density function $p(\vec{x})$ changes rapidly over a short distance in the functional space. More sophisticated methods, such as grid search, may be needed to tackle such a scenario.

After determining the parameters of the ERM, we first examine the spectrum of the ERM with uniformly distributed random functional coordinates $\vec{x}_i \in [0, L]^d$ (*Figure 5—figure supplement 1M–R*). Second, we use $f(\vec{x})$ to translate experimental pairwise correlations into pairwise distances for all neurons in the functional space (*Figure 5—figure supplement 2, Figure 5—figure supplement 1G–L*). The embedding coordinates $\vec{x}_i$ in the functional space can then be solved through multidimensional scaling (MDS) by minimizing the Sammon error (Methods). The similarity between the spectra of the uniformly distributed coordinates (*Figure 5—figure supplement 1M–R*) and those of the embedding coordinates (*Figure 5—figure supplement 1G–L*) is also consistent with the notion that specific coordinate distributions in the functional space have little impact on the shape of the eigenspectrum (*Appendix 1—figure 1*).

## Nonnegativity of data covariance

To use ERM to model the covariance matrix, the pairwise correlation is given by a *non-negative* kernel function $f(\vec{x})$ that monotonically decreases with the distance between neurons in the functional space. This nonnegativeness brings about a potential issue when applied to experimental data, where, in fact, a small fraction of pairwise correlations/covariances are negative. We have verified that the spectrum of the data covariance matrix (*Figure 2—figure supplement 3*) remains virtually unchanged when replacing these negative covariances with zero (*Figure 2—figure supplement 3*). This confirms that the ERM remains a good model when the neural dynamics is in a regime where pairwise covariances are mostly positive *Dahmen et al., 2019* (see also *Figure 2—figure supplement 2B, Figure 2—figure supplement 2B–D*).

## Multidimensional scaling

With the estimated ERM parameters ($\mu$ in $f(\vec{x})$ and the box size $L$ for given $\epsilon$ and $d$, see Methods), we performed MDS to infer neuronal coordinates $\vec{x}_i$ in functional space. First, we computed a pairwise correlation $R_{ij} = \frac{C_{ij}}{\sigma_i\sigma_j}$ from the data covariances. Next, we calculated the pairwise distance, denoted by $u_{ij}^*$, by computing the inverse function of $f(\vec{x})$ with respect to the absolute value of $R_{ij}$, $u_{ij}^* = f^{-1}(|R_{ij}|)$. We used the absolute value $|R_{ij}|$ instead of $R_{ij}$ as a small percentage of $R_{ij}$ are negative (*Figure 2—figure supplement 2A–D*) where the distance is undefined. This substitution by the absolute value serves as a simple workaround for the issue and is only used here in the analysis to infer the neuronal coordinates by MDS. Finally, we estimated the embedding coordinates $\vec{x}_i$ for each neuron by the SMACOF algorithm (Scaling by MAjorizing a COmplicated Function), which minimizes the Sammon error

$$E = \frac{1}{\sum\limits_{i<j} u_{ij}^*} \sum\limits_{i<j} \frac{(u_{ij}^* - u_{ij})^2}{u_{ij}^*} \tag{23}$$

where $u_{ij} = \|\vec{x}_i - \vec{x}_j\|$ is the pairwise distance in the embedding space calculated above. To reduce errors at large distances (i.e., small correlations with $R_{ij} < f(L)$, where $L$ is the estimated box size), we performed a soft cut-off at a large distance:

$$\begin{aligned} u_{ij}^* &= f^{-1}(|R_{ij}|), & R_{ij} \geq f(L) \\ u_{ij}^* &= L\log(f^{-1}(|R_{ij}|)/L) + L, & R_{ij} < f(L) \end{aligned} \tag{24}$$

During the optimization process, we started at the embedding coordinates estimated by the classical MDS (*Cox and Cox, 2000*), with an initial sum of squares distance error that can be calculated directly, and ended with an error or its gradient smaller than $10^{-4}$.

The fitted ERM with the embedding coordinates $\vec{x}_i$ reproduced the experimental covariance matrix including the cluster structures (*Figure 5—figure supplement 2*) and its sampling eigenspectra (*Figure 5—figure supplement 1*).

## Canonical correlation analysis

Here we briefly explain the CCA method (*Knapp, 1978*) for completeness. The basis vectors $\vec{v}_{\text{func}}$ and $\vec{v}_{\text{anat}}$, in functional and anatomical space, respectively, were found by maximizing the correlation $R_{\text{CCA}} = corr(\{\vec{v}_{\text{func}} \cdot \vec{x}_i\}, \{\vec{v}_{\text{anat}} \cdot \vec{y}_i\})$. These basis vectors satisfy the condition that the projections of the neuron coordinates along them, $\{\vec{x}_i \cdot \vec{v}_{\text{func}}\}$ and $\{\vec{y}_i \cdot \vec{v}_{\text{anat}}\}$, are maximally correlated among all possible choices of $\vec{v}_{\text{func}}$ and $\vec{v}_{\text{anat}}$. Here, $\{\vec{x}_i\}$, $\{\vec{y}_i\}$ represent the coordinates in functional and anatomical spaces, respectively. The resulting maximum correlation is $R_{\text{CCA}}$. To check the significance of the canonical correlation, we shuffled the functional space coordinates $\{\vec{x}_i\}$ across neurons' identity and re-calculated the canonical correlation with the anatomical coordinates, as shown in *Figure 5—figure supplement 4*.

To study the effect of functional–anatomical relation described by $R_{\text{CCA}}$ in the ERM model, we generated three-dimensional anatomical coordinates $\{\vec{y}_i\}$ and two-dimensional functional coordinates $\{\vec{x}_i\}$ for each neuron which are jointly five-dimensional zero-mean multivariate Gaussian random variables. The coordinates are independent among each other, except for the first dimension $\{\vec{x}_i^1\}$ of the functional coordinates and the first dimension $\{\vec{y}_i^1\}$, which are assigned to have a correlation coefficient equals to $R_{\text{CCA}}$. The variances of the coordinates are $\sigma_{y1}^2 = 1, \sigma_{y2}^2 = 1, \sigma_{y3}^2 = 1$, and $\sigma_{x1}^2 = 2, \sigma_{x2}^2 = 1$ for the numerics in *Figure 5—figure supplement 7*. Under this construction, the first canonical correlation between the anatomical and functional coordinates equals $R_{\text{CCA}}$, and the first canonical direction $\vec{v}_{\text{anat}}$ in the anatomical space is $(1, 0, 0)^T$ and the first canonical direction $\vec{v}_{\text{func}}$ in the functional space is $(1, 0)^T$.

## Spectrum of three types of sampling procedures in ERM model

In Result, we have considered three types of sampling procedures: random sampling (RSap), spatial sampling in the anatomical space (ASap, e.g., recording neurons in a brain region), and spatial sampling in the functional space (FSap), namely spatial sampling in functional space by subdividing the space into smaller regions, is equivalent to the previously reported RG inspired process (*Bradde and Bialek, 2017*). Here, we consider the relationship between the spectrum of three types of sampling procedures.

We assume a uniform random distribution of neurons in a $d$-dimensional functional space, $[0, L]^d$. For RSap procedures, the resulting neuronal density $\rho_R$ is reduced to $\rho_R = k\rho_0$, with $k$ representing the sampling ratio ($k = N/N_0$) and $\rho_0$ being the initial density. In contrast, FSap maintains the original density, $\rho_F = \rho_0$. This constancy in neuronal density under FSap ensures that the covariance eigenspectrum remains invariant across scales for any spatial correlation functions $f(\vec{x})$, such as power law and exponential, as shown in *Appendix 1—figure 5A, B, D, E*. In contrast, RSap reduces $\rho$, thus demanding more rigorous conditions to achieve a scale-invariant covariance spectrum (e.g., compare *Appendix 1—figure 5A, C*).

Under ASap, sampled neurons are not spread out evenly in functional space, whereas our theoretical framework assumes a uniform distribution. To reconcile this discrepancy, we employ a uniform

approximation of the neural distribution. This approach involves introducing an effective density, $\rho'$, defined as the spatial average of the density function $\rho(\vec{x})$. This adjustment allows our theoretical model to accommodate non-uniform distributions encountered in anatomically spatial sampling.

$$\rho' \equiv \langle \rho(\vec{x}) \rangle = \int p(\vec{x})\rho(\vec{x})\mathrm{d}\vec{x} = kN_0 \int p^2(\vec{x})\mathrm{d}\vec{x}, \tag{25}$$

where $p(\vec{x})$ is the normalized density distribution (see Methods).

Using the Cauchy–Schwarz inequality, we have

$$\int p^2(\vec{x})\mathrm{d}\vec{x} \int \mathrm{d}\vec{x} \geq \left( \int p(\vec{x})\mathrm{d}\vec{x} \right)^2 \tag{26}$$

thus $\rho' \geq k\rho_0$.

According to the condition $p(\vec{x}) < \frac{1}{kV}$, we have $\rho' \leq \rho_0$, intuitively, sampling within a uniformly distributed neuron population does not increase the density.

So we have $\rho_0 \geq \rho'_A \geq k\rho_0$, that is, $\rho_F \geq \rho'_A \geq \rho_R$. Thus, the spectrum ASap should be between FSap and RSap.

## Dimensions of three types of sampling procedures in ERM model

### Scaling of dimensions through random sampling

Let us revisit the definition of the participation ratio (PR) dimension as defined in *Equation 5*:

$$D_{\mathrm{PR}}(C) = \frac{\left(\sum_i \lambda_i\right)^2}{\sum_i \lambda_i^2} = \frac{\left(\mathrm{Tr}(C)\right)^2}{\mathrm{Tr}(C^2)} = \frac{N^2 \mathrm{E}(\sigma^2)^2}{N\mathrm{E}(\sigma^4) + N(N-1)\mathrm{E}_{i \neq j}(C_{ij}^2)} \tag{27}$$

During the random sampling process, the expected values $E(\sigma^2)$, $E(\sigma^4)$, and $\mathrm{E}_{i \neq j}(C_{ij}^2)$ remain constant. These constants allow for the estimation of the PR dimension across various scales using:

$$D_{\mathrm{PR}}^{\mathrm{RSap}} = \frac{kN_0\mathrm{E}(\sigma^2)^2}{\mathrm{E}(\sigma^4) + (kN_0 - 1)\mathrm{E}_{i \neq j}(C_{ij}^2)} \tag{28}$$

Here, $k = N/N_0$ represents a scaling factor (fraction) associated with sampling. The key question is to understand how the dimensionality changes with $k$. Under random sampling, as $k$ increases, the dimensionality will quickly approaches a saturating point defined by *Equation 1*.

### Scaling of dimensions through functional sampling

In this section, we leverage the uniform ERM model to estimate dimensions within the context of functional sampling, specifically focusing on the estimation of squared pairwise covariance $\mathrm{E}_{i \neq j}(C_{ij}^2)$ and dimensionality. Adopting an approximation for a power-law kernel function $f(x) \approx \epsilon^\mu \|x\|^{-\mu}$ allows us to express the expected value of the squared covariance $\mathrm{E}_{i \neq j}(C_{ij}^2)$ as follows:

$$
\begin{aligned}
\mathrm{E}_{i \neq j}(C_{ij}^2) &= \int_{[0,L]^d} p(\vec{x}_1)p(\vec{x}_2)f^2(\|\vec{x}_1 - \vec{x}_2\|)\mathrm{d}\vec{x}_1\mathrm{d}\vec{x}_2 \\
&\approx \int_{[0,L]^d} p(\vec{x}_1)p(\vec{x}_2)\epsilon^{2\mu}\|\vec{x}_1 - \vec{x}_2\|^{-2\mu}\mathrm{d}\vec{x}_1\mathrm{d}\vec{x}_2.
\end{aligned}
\tag{29}
$$

For a set subjected to functional sampling with a sampling fraction $k$, this procedure adjusts the size of the functional space in the ERM model by a factor of $k^{-1/d}$. Consequently, the $\mathrm{E}_{i \neq j}^k(C_{ij}^2)$ for the sampled fraction $k$ is given by:

$$
\begin{aligned}
\mathrm{E}_{i\neq j}^{k}(C_{ij}^2) &= \int_{[0,k^{1/d}L]^d} p(\vec{x}_1)p(\vec{x}_2)f^2(\|\vec{x}_1 - \vec{x}_2\|)\mathrm{d}\vec{x}_1\mathrm{d}\vec{x}_2 \\
&= \int_{[0,L]^d} p(\vec{x}_1)p(\vec{x}_2)f^2(k^{1/d}\|\vec{x}_1 - \vec{x}_2\|)\mathrm{d}\vec{x}_1\mathrm{d}\vec{x}_2 \\
&\approx \int_{[0,L]^d} p(\vec{x}_1)p(\vec{x}_2)\epsilon^{2\mu}k^{-2\mu/d}\|\vec{x}_1 - \vec{x}_2\|^{-2\mu}\mathrm{d}\vec{x}_1\mathrm{d}\vec{x}_2 \\
&\approx k^{-2\mu/d}\mathrm{E}_{i\neq j}(C_{ij}^2),
\end{aligned}
\tag{30}
$$

Here, we assume that $E[\sigma^2]$ and $E[\sigma^4]$ are constant across the sampling process. This model enables the estimation of the ratio $\mu/d$ as detailed in the Methods.

$$
D_{\mathrm{PR}}^{\mathrm{FSap}} \approx \frac{kN_0\mathrm{E}(\sigma^2)^2}{\mathrm{E}(\sigma^4) + (kN_0 - 1)k^{-2\mu/d}\mathrm{E}_{i\neq j}(C_{ij}^2)}
\tag{31}
$$

In the large $N$ limit, we observe distinct behaviors in the evolution of dimensionality in both theory and data: it saturates in RSap (dashed line in *Figure 5D*), namely $D_{\mathrm{PR}}^{\mathrm{RSap}} \approx D_{\mathrm{PR}}$ defined in *Equation 1*, whereas it follows a different scaling relationship $D_{\mathrm{PR}}^{\mathrm{FSap}} \approx k^{2\mu/d}D_{\mathrm{PR}}$ in FSap (solid line in *Figure 5D*).

## Comparative analysis of PR dimension across sampling techniques

This section examines the behavior of the PR dimension under three sampling techniques: anatomical sampling, random sampling, and functional sampling. We show that the average PR dimension following anatomical sampling occupies a middle ground between the extremes presented by random and functional sampling.

The PR dimension, denoted $D_{\mathrm{PR}}$, reflects the sampling impact and depends on the distribution $p(\vec{X})$ of the functional coordinates $\vec{X}$. Defining the sampling fraction as $k = 1/q$, the mean $D_{\mathrm{PR}}$ is represented as:

$$
\mathrm{mean}(D_{\mathrm{PR}}) = \frac{1}{q}\sum_{i=1}^{q} D_{\mathrm{PR}}^i = \frac{1}{q}\sum_{i=1}^{q} J(p_i(\vec{X})),
\tag{32}
$$

where the neuron set $1, 2, ..., N$ is segmented into $q$ clusters $\{\vec{X}_1, \vec{X}_2, ..., \vec{X}_q\}$, each comprising $\frac{N}{q}$ neurons. The probability distribution $p_i(\vec{X})$ corresponds to each cluster $\{\vec{X}_i\}$. The probability distribution for each cluster, $p_i(\vec{X})$, emerges naturally from the sampling process.

The equivalence of the mean probability density function across the sampled clusters to the original set's probability density function leads us to the condition:

$$
\frac{1}{q}\sum_{i=1}^{q} p_i(\vec{X}) = p(\vec{X}),
\tag{33}
$$

This condition is a direct consequence of the sampling process, ensuring that the aggregated probability density function of all sampled sets mirrors the overall density distribution of the neurons.

Applying the Lagrange multiplier method to optimize the mean $D_{\mathrm{PR}}$:

$$
L(p, \lambda) = \frac{1}{q}\sum_{i=1}^{q} J(p_i(\vec{X})) + \int_D \mathrm{d}^d\vec{X}\lambda(\vec{X})\left(\frac{1}{q}\sum_{i=1}^{q} p_i(\vec{X}) - p(\vec{X})\right),
\tag{34}
$$

Here, $L(p, \lambda)$ is the Lagrangian, $\lambda(\vec{X})$ is the Lagrange multiplier, we derive the optimal condition:

$$
\frac{\partial L(p, \lambda)}{\partial p_i} = 0,
\tag{35}
$$

yielding:

$$
\frac{1}{q}\frac{\partial J}{\partial p_i(\vec{X})} + \frac{\lambda(\vec{X})}{q} = 0.
\tag{36}
$$

At the optimal mean $D_{\mathrm{PR}}$, each $p(\vec{X}_i)$ is equivalent, leading to $p(\vec{X}_i) = p(\vec{X}_j) = p(\vec{X})$ (representative of random sampling). Hence, the mean $D_{\mathrm{PR}}$ post-random sampling sets the upper limit for the mean $D_{\mathrm{PR}}$ after anatomical sampling.

Let us investigate the lower bound of the mean PR dimension with the ERM model. For the minimization of mean ($D_{\mathrm{PR}}$), a key requirement is the functional spatial proximity of neurons within the same cluster, in other words, the neuron set should be distinctly separated in functional space. Consequently, achieving the minimum mean PR dimension necessitates a functional sampling strategy.

## Derive upper bound of dimension from spectrum

To deduce $D_{PR}$ from the spectrum, for simplicity, we focus on the high-density region, where we have an analytical expression for $\lambda$ that is valid for large eigenvalues:

$$\lambda_r = \gamma \left(\frac{r}{N}\right)^{-1+\frac{\mu}{d}} \cdot \rho^{\frac{\mu}{d}} = \gamma r^{-1+\frac{\mu}{d}} L^{-\mu} N \quad \text{for} \quad r \leq \beta(N), \tag{37}$$

where $L$ is the size of the functional space, $\gamma$ is the coefficient in *Equation 3*, which depends on $d$, $\mu$, and $\mathrm{E}(\sigma^2)$. Note that the eigenvalue $\lambda_r$ decays rapidly after the threshold $r = \beta(N)$. Since we did not discuss small eigenvalues in this article, we represent them here as an unknown function $\eta(r, N, L)$:

$$\lambda_r = \eta(r, N, L) \quad \text{for} \quad r > \beta(N) \tag{38}$$

As discussed in Methods, without changing the properties of the spectrum, we can always impose $\mathrm{E}(\sigma^2) = 1$ such that

$$\sum_{r=1}^{N} \lambda_r = \mathrm{Tr}(C) = N \tag{39}$$

We emphasize that this constraint requires that large and small eigenvalues behave differently because otherwise $\sum_{r=1}^{N} r^{-\alpha}$ with $\alpha < 1$ would scale as $N^{1-\alpha}$, and $\sum_{r=1}^{N} \lambda_r$ is not proportional to $N$.

Using the Cauchy–Schwarz inequality, we have an upper bound of $\sum_{r=1}^{N} \lambda_r^2$:

$$\sum_{r=1}^{N} \lambda_r^2 \leq \left(\sum_r \lambda_r\right)^2 = N^2 \tag{40}$$

On the other hand, $\lambda_1^2$ is a lower bound of $\sum_{r=1}^{N} \lambda_r^2$:

$$\sum_{r=1}^{N} \lambda_r^2 > \lambda_1^2 = L^{-2\mu} N^2 \gamma^2 \tag{41}$$

As a result, the dimensionality

$$D_{\mathrm{PR}} = \frac{\left(\sum_{r=1}^{N} \lambda_r\right)^2}{\sum_{r=1}^{N} \lambda_r^2},$$

is bounded as

$$1 \leq D_{\mathrm{PR}} < L^{2\mu} \gamma^{-2} \tag{42}$$

Under random sampling, $L$ remains fixed. Thus, we must have a bounded dimensionality that is independent of $N$ for our ERM model. A tighter lower bound of $\sum_{r=1}^{N} \lambda_r^2$ is

$$\sum_{r=1}^{N} \lambda_r^2 > \gamma^2 L^{-2\mu} N^2 \sum_{r=1}^{\beta(N)} \left(r^{-2+2\mu/d}\right) \tag{43}$$

A tighter upper bound of participation ratio $D_{\mathrm{PR}}$ can be written as:

$$D_{\text{PR}} = \frac{\left(\sum_{r=1}^{N} \lambda_r\right)^2}{\sum_{r=1}^{N} \lambda_r^2} < \frac{L^{2\mu}\gamma^{-2}}{\sum_{r=1}^{\beta(N)} \left(r^{-2+2\mu/d}\right)} < L^{2\mu}\gamma^{-2} \tag{44}$$

However, in functional sampling, enlarging the region size with constant density $\rho$ results in $L \sim N^{1/d}$. Thus, the upper bound of $D_{\text{PR}}$ should grow as $N^{2\mu/d}$, consistent with the previously derived result (*Equation 31*) in Methods.

## Simulating CCA and anatomical sampling

In this section, we estimate the dimensions of the anatomically sampled neuron set. For simplicity, we assume that the functional coordinates of neurons, $X_i$, and the anatomical coordinates of neurons, $Y_i$, both follow a multivariate Gaussian distribution. We define anatomical sampling, which involves sampling on $Y_i$, along a direction chosen arbitrarily and denote this direction as $Y^A$. Subsequently, we perform sampling on $X_i$ in the direction denoted by $X^A$, which is determined to have the highest correlation with $Y^A$ according to CCA. This process effectively mimics the scenario of functional sampling.

The key to calculating the PR dimension involves computing the expected value $E_{i \neq j}(C_{ij}^2)$. In the ERM model, the distribution of $C_{ij}$ can be estimated by the distribution of points in the functional space. This allows for the calculation of the PR dimension across anatomical sampling by comparing the distribution of $X_i$ after anatomical sampling with that after functional sampling. We can model the distribution of $X^A$ and $Y^A$ as follows:

$$R_{\text{ASap}} = \text{corr}(X^A, Y^A),$$

$$C_{\text{ASap}} = \text{corr}(X^A, Y^A)\sigma_x\sigma_y,$$

$$\begin{bmatrix} X^A \\ Y^A \end{bmatrix} \sim \mathcal{N}\left( \begin{bmatrix} 0 \\ 0 \end{bmatrix}, \begin{bmatrix} \sigma_x^2 & C_{\text{ASap}} \\ C_{\text{ASap}} & \sigma_y^2 \end{bmatrix} \right), \tag{45}$$

Here, we consider only the projection of the functional coordinate onto the direction $X^A$, which exhibits the highest correlation, denoted by $R_{\text{ASap}}$, with $Y^A$. Specifically, when selecting the anatomical direction as the first CCA direction, the correlation between $X^A$ and $Y^A$ reaches its maximum, such that $R_{\text{ASap}} = R_{\text{CCA}}$. In this case, anatomical sampling results in the minimization of the dimensionality.

Now, let us perform anatomical sampling on the neurons. The $\vec{X}_i$ and $\vec{Y}_i$ denote the functional and anatomical coordinates of the $i^{\text{th}}$ neuron cluster after anatomical sampling, respectively.

To approximate, we need to calculate the functional coordinate probability distribution $p(\vec{X}_i) = p(\vec{X}|q_{ik}^y < Y^A < q_{(i+1)k}^y)$, which is the distribution of the $i^{\text{th}}$ neuron cluster after anatomical sampling. $Y^A$ represents the selected direction in anatomical space, and $q_{ik}^y$ denotes the $ik^{\text{th}}$ quantile of $Y^A$, where $k$ is the sampled fraction. Note the following relationships and distributions:

$$p(X^A|Y^A = y) = \frac{p(X^A, Y^A = y)}{p(Y^A = y)},$$

$$p(X^A|Y^A = y) \sim \mathcal{N}\left( y\frac{\sigma_x}{\sigma_y}R_{\text{ASap}}, \sigma_x^2(1 - R_{\text{ASap}}^2) \right). \tag{46}$$

$$p(X_i^A) = p(X^A|q_{ik}^y < Y^A < q_{(i+1)k}^y) = \frac{1}{k}\int_{q_{ik}^y}^{q_{(i+1)k}^y} p(X^A|Y^A = y)dy \tag{47}$$

The conditional probability distribution $P(X^A|q_{ik}^y < Y^A < q_{(i+1)k}^y)$ is equivalent to the distribution of the sum of $Y_i^A \frac{\sigma_x}{\sigma_y}R_{\text{ASap}}$ and $X_0$, where $X_0 \sim \mathcal{N}(0, \sigma_x^2(1 - R_{\text{ASap}}^2))$:

$$X_i^A = Y_i^A \frac{\sigma_x}{\sigma_y}R_{\text{ASap}} + X_0, \tag{48}$$

$$p(Y_i^A = y) = \begin{cases} \dfrac{1}{k\sqrt{2\pi}\sigma_y} \exp\left(-\dfrac{y^2}{2\sigma_y^2}\right) & \text{for } q_{ik}^y < y < q_{(i+1)k}^y, \\ 0 & \text{otherwise.} \end{cases} \tag{49}$$

The computation of $X_i^A$ involves two technical challenges: (1) The distribution of $Y_i^A$ is represented by a non-elementary function (**Equation 49**), which complicates the direct calculation of $X_i^A$, which is the sum of $Y_i^A R_{\text{ASap}}\sigma_x/\sigma_y$ and $X_0$. To facilitate approximation, we model $Y_i^A$ using a normal distribution with equivalent variance. (2) Calculating the variance of $Y_i^A$ presents direct challenges, and the variance of $Y_i^A$ differs across different neuron clusters $i$. Using a uniform distribution for $Y$ simplifies this task (this assumption is only used to calculate the variance of $Y_i^A$). Under this assumption, the variance of $Y_i^A$ can be straightforwardly calculated as $\text{Var}(Y_i^A) = k^2\sigma_y^2$. Consequently, we approximate $Y_i^A$ and $X_i^A$ as follows:

$$Y_i^A \sim \mathcal{N}\left(\frac{q_{ik}^y + q_{(i+1)k}^y}{2}, k^2\sigma_y^2\right), \tag{50}$$

$$X_i^A \sim \mathcal{N}\left(\frac{q_{ik}^y + q_{(i+1)k}^y}{2}\frac{\sigma_x}{\sigma_y}R_{\text{ASap}}, \sigma_x^2(1 - R_{\text{ASap}}^2 + k^2 R_{\text{ASap}}^2)\right). \tag{51}$$

Calculating the PR dimension directly from the distribution of $X_i^A$ is difficult; thus, we approximate anatomical sampling with fraction $k$ as functional sampling with fraction $k_f$, leading to:

$$k_f = \sqrt{1 + k^2 R_{\text{ASap}}^2 - R_{\text{ASap}}^2}. \tag{52}$$

Using the equation for functional sampling $\text{E}_{i\neq j}^k(C_{ij}^2) \approx k^{-2\mu/d}\text{E}_{i\neq j}(C_{ij}^2)$ (**Equation 30**):

$$\text{E}_{i\neq j}^k(C_{ij}^2) \approx (1 + k^2 R_{\text{ASap}}^2 - R_{\text{ASap}}^2)^{-\mu/d}\text{E}_{i\neq j}(C_{ij}^2). \tag{53}$$

$$D_{\text{PR}}^{\text{ASap}} \approx \frac{kN_0\text{E}(\sigma^2)^2}{\text{E}(\sigma^4) + (kN_0 - 1)(1 + k^2 R_{\text{ASap}}^2 - R_{\text{ASap}}^2)^{-\mu/d}\text{E}_{i\neq j}(C_{ij}^2)} \tag{54}$$

## Acknowledgements
We are grateful to Liqun Luo and Changsong Zhou for their helpful suggestions on our manuscript. QW thanks Hideaki Shimazaki for the suggestion that the functional space could be the feature space for sensory coding. QW thanks Jia Lou for improving the illustration in **Figures 1 and 3**. QW was supported by the STI2030-Major Projects 2022ZD0211900 and the NSFC-32071008 from the National Science Foundation of China. YH was supported by ECS-26303921 from the Research Grants Council of Hong Kong.

## Additional information

### Funding

| Funder | Grant reference number | Author |
| --- | --- | --- |
| Ministry of Science and Technology of the People's Republic of China | 2022ZD0211900 | Zezhen Wang Yuming Chai Kexin Qi Guodong Tan Quan Wen |
| Research Grants Council, University Grants Committee | ECS-26303921 | Weihao Mai Yu Hu |

| Funder | Grant reference number | Author |
| --- | --- | --- |
| National Science Foundation of China | NSFC-32071008 | Zezhen Wang<br>Yuming Chai<br>Kexin Qi<br>Guodong Tan |

The funders had no role in study design, data collection, and interpretation, or the decision to submit the work for publication.

## Author contributions

Zezhen Wang, Weihao Mai, Conceptualization, Data curation, Software, Formal analysis, Validation, Investigation, Visualization, Methodology, Writing – original draft, Writing – review and editing; Yuming Chai, Resources, Data curation, Formal analysis, Investigation, Methodology, Writing – original draft; Kexin Qi, Data curation, Formal analysis, Investigation, Methodology; Hongtai Ren, Chen Shen, Guodong Tan, Methodology; Shiwu Zhang, Project administration; Yu Hu, Conceptualization, Software, Formal analysis, Supervision, Funding acquisition, Validation, Investigation, Methodology, Writing – original draft, Project administration, Writing – review and editing; Quan Wen, Conceptualization, Resources, Software, Formal analysis, Supervision, Funding acquisition, Validation, Investigation, Visualization, Methodology, Writing – original draft, Project administration, Writing – review and editing

## Author ORCIDs

Zezhen Wang ⓘ https://orcid.org/0009-0006-1401-9871
Weihao Mai ⓘ https://orcid.org/0000-0002-2320-2276
Yuming Chai ⓘ https://orcid.org/0000-0003-0184-1824
Yu Hu ⓘ https://orcid.org/0000-0003-0790-1605
Quan Wen ⓘ https://orcid.org/0000-0003-0268-8403

Joint public review https://doi.org/10.7554/eLife.100666.3.sa1
Author response https://doi.org/10.7554/eLife.100666.3.sa2

# Additional files

## Supplementary files

MDAR checklist

## Data availability

The source code in this work can be found at https://github.com/wzz1999/ERM-scale (copy archived at *Mak and Wang, 2025*). The fish data collected and analyzed in this work can be found at https://doi.org/10.6084/m9.figshare.28721477.

The following dataset was generated:

| Author(s) | Year | Dataset title | Dataset URL | Database and Identifier |
| --- | --- | --- | --- | --- |
| Wang Z, Mai W, Chai Y, Hongtai R, Chen S, Shiwu Z, Guodong T, Yu H, Quan W, Kexin Q | 2025 | Whole-brain light-field imaging data | https://doi.org/10.6084/m9.figshare.28721477 | figshare, 10.6084/m9.figshare.28721477 |

The following previously published datasets were used:

| Author(s) | Year | Dataset title | Dataset URL | Database and Identifier |
|---|---|---|---|---|
| Chen X, Mu Y, Kuan A, Nikitchenko M, Randlett O, Chen A, Gavornik J, Sompolinsky H, Engert F, Ahrens MB | 2018 | Whole-brain light-sheet imaging data | https://doi.org/10.25378/janelia.7272617 | figshare, 10.25378/janelia.7272617 |
| Steinmetz N, Pachitariu M, Stringer C, Carandini M, Harris K | 2019 | Eight-probe Neuropixels recordings during spontaneous behaviors | https://doi.org/10.25378/janelia.7739750 | figshare, 10.25378/janelia.7739750 |
| Stringer C, Pachitariu M, Reddy C, Carandini M, Harris KD | 2018 | Recordings of ten thousand neurons in visual cortex during spontaneous behaviors | https://doi.org/10.25378/janelia.6163622 | figshare, 10.25378/janelia.6163622 |

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

## Appendix 1

## Extensions of ERM and factors not affecting the scale invariance

In *Appendix 1—figure 1*, we considered five additional types of spatial density distributions (coordinate distributions) in functional space and two additional functional space geometries. We examined the points distributed according to the uniform distribution ($\vec{x} \sim 1/L^d$), the normal distribution ($\vec{x} \sim \mathcal{N}(\mu_p, \sigma_p^2 \mathbf{I})$), and the log-normal distribution ($\log \vec{x} \sim \mathcal{N}(\mu_p, \sigma_p^2 \mathbf{I})$). We used the method described in Methods to adjust the parameters of the coordinate distributions based on the uniform distribution case, so that they all generate similar pairwise correlation distributions. The relationships between these parameters are described in Methods. In *Appendix 1—figure 1B*, we used the following parameters: $d = 2$ for the uniform distribution; $\mu_p = 0$, $\sigma_p = 2.82$ for the normal distribution; and $\mu_p = 2$, $\sigma_p = 0.39$ for the log-normal distribution.

Second, we introduced multiple clusters of neurons in the functional space, with each cluster uniformly distributed in a box. We considered three arrangements: (1) two closely situated clusters (with a box size of $L = 5\sqrt{2}$, the distance between two cluster centers being $L_c = L$), and (2) two distantly situated clusters (with a box size of $L = 5\sqrt{2}$ and the distance between clusters $L_c = 4L$), and three clusters arranged symmetrically in an equilateral triangle (with a box size of $L = 10/\sqrt{3}$ and the distance between clusters $L_c = L$).

Finally, we examined the scenario in which the points were uniformly distributed on the surface of a sphere ($4\pi l^2 = L^2$, $l$ being the radius of the sphere) or a hemisphere ($2\pi l^2 = L^2$) embedded in $\mathbb{R}^3$ (the pairwise distance is that in $\mathbb{R}^3$). It should be noted that both cases have the same surface area as the 2D box.

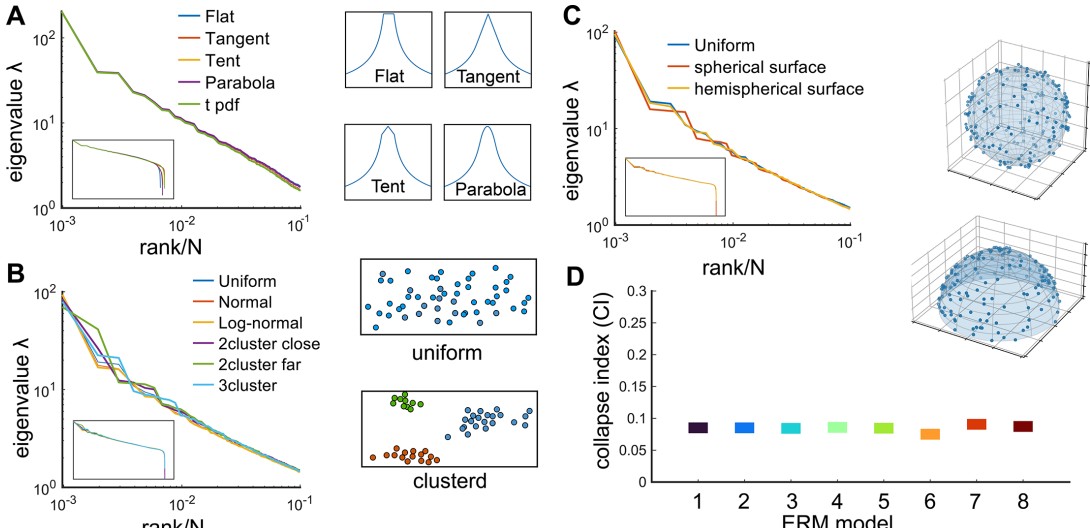

**Appendix 1—figure 1.** Factors that do not affect the scale invariance. (**A**) Rank plot of the covariance eigenspectrum for ERMs with different $f(\vec{x})$ (see *Table 3*). Diagrams show different slow-decaying kernel functions $f(\vec{x})$ along a 1D slice. (**B**) Same as A but for different coordinate distributions in the functional space (see text). The diagrams on the right illustrate uniform and clustered coordinate distributions. (**C**) Same as A but for different geometries of the functional space (see text). Diagrams illustrate spherical and hemispherical surfaces. (**D**) CI of the different ERMs considered in (**A–C**). The range on the y-axis is identical to *Figure 4C*. On the x-axis, 1: uniform distribution, 2: normal distribution, 3: log-normal distribution, 4: uniform two nearby clusters, 5: uniform two faraway clusters, 6: uniform 3-cluster, 7: spherical surface in $\mathbb{R}^3$, 8: hemispherical surface in $\mathbb{R}^3$. All ERM models in (**B, C**) are adjusted to have a similar distribution of pairwise correlations (Appendix 1).

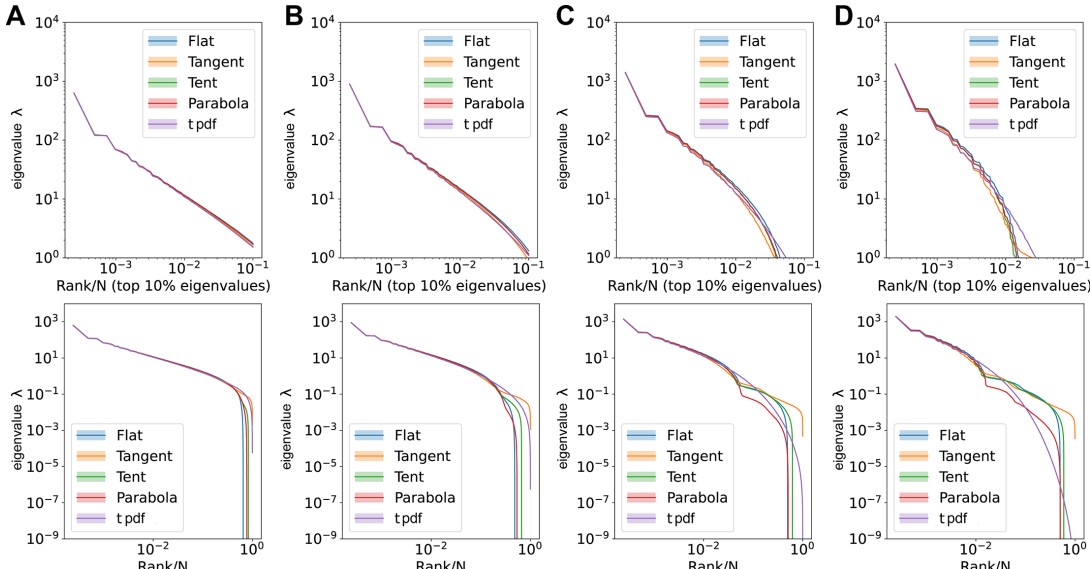

**Appendix 1—figure 2.** Comparisons of large eigenvalues across different smoothing interval sizes, $\varepsilon$. Rank plot (upper row) and pdf (lower row) of the covariance eigenspectrum for ERMs with different $f(\vec{x})$. (**A**) $\epsilon = 0.06$. (**B**) $\epsilon = 0.12$. (**C**) $\epsilon = 0.3$. (**D**) $\epsilon = 0.6$. Other ERM simulation parameters: $N = 4096$, $\rho = 100$, $\mu = 0.5$, $d = 2$, $L = 6.4$, $\sigma_i^2 = 1$. The formulas for different $f(\vec{x})$'s are listed in **Table 3** in Methods.

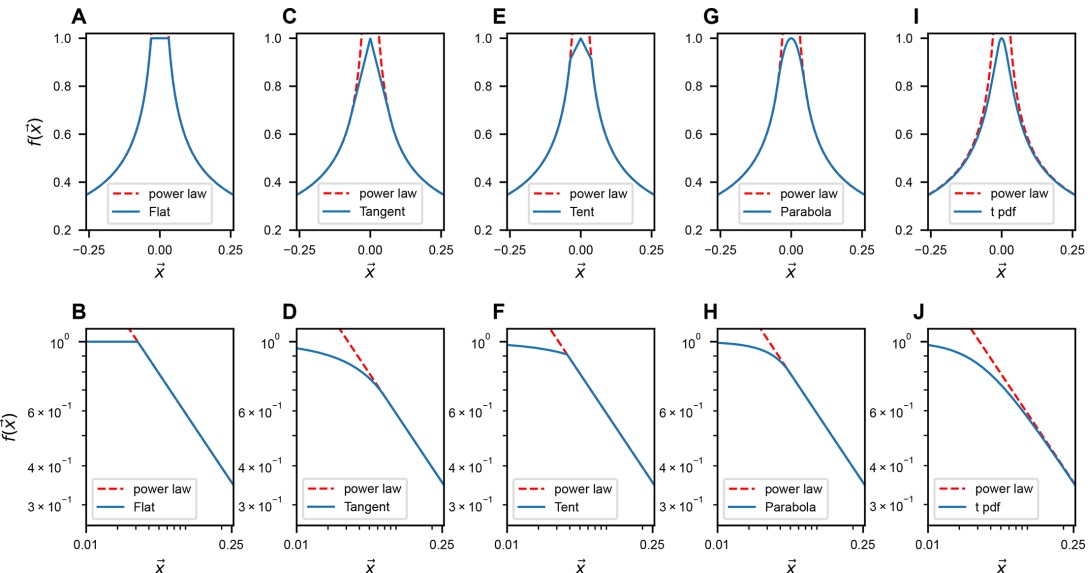

**Appendix 1—figure 3.** Modifications of $f(\vec{x})$ near $x = 0$. The upper row illustrates the slow-decaying kernel function $f(\vec{x})$ (blue solid line) and its power-law asymptote (red dashed line) along a 1D slice at various $f(\vec{x})$. The lower row is similar to **A**, but on the log–log scale. The formulas for different $f(\vec{x})$'s are listed in **Table 3** in Methods.

## RG approach

Here we briefly summarize the RG approach used in *Meshulam et al., 2019* and elucidate the adjustments required when applying the RG approach to ERM. The method consists of two stages: (i) iterative agglomerate clustering of neurons, and (ii) computing the spectrum of a block of the *original* covariance matrix corresponding to a cluster of the desired size based on the previous clustering result.

## Stage (i): iterative clustering

We begin with $N_0$ neurons, where $N_0$ is assumed to be a power of 2. In the first iteration, we compute Pearson's correlation coefficients for all neuron pairs. We then search greedily for the most correlated pairs and group the half pairs with the highest correlation into the first cluster; the remaining neurons form the second cluster. For each pair $(a, b)$, we define a coarse-grained variable according to:

$$x_i^k = Z_{ab}^{k-1}(x_a^{k-1} + x_b^{k-1}), \tag{S1}$$

where $Z_{ab}^{k-1}$ normalizes the average to ensure unit nonzero activity. This process reduces the number of neurons to $N_1 = N_0/2$. In subsequent iterations, we continue grouping the most correlated pairs of the coarse-grained neurons, iteratively reducing the number of neurons by half at each step. This process continues until the desired level of coarse-graining is achieved.

When applying the RG approach to ERM, instead of combining neural activity, we merge correlation matrices to traverse different scales. During the $k$th iteration, we compute the coarse-grained covariance as:

$$c_{ij}^k = c_{ab}^{k-1} + c_{ac}^{k-1} + c_{bc}^{k-1} + c_{bd}^{k-1} \tag{S2}$$

and the variance as:

$$c_{ii}^k = c_{aa}^{k-1} + c_{bb}^{k-1} + 2c_{ab}^{k-1} \tag{S3}$$

Following these calculations, we normalize the coarse-grained covariance matrix to ensure that all variances are equal to one. Note that these coarse-grained covariances are only used in stage (i) and not used to calculate the spectrum.

## Stage (ii): eigenspectrum calculation

The calculation of eigenspectra at different scales proceeds through three sequential steps. First, for each cluster identified in stage (i), we compute the covariance matrix using the original firing rates of neurons within that cluster (not the coarse-grained activities). Second, we calculate the eigenspectrum for each cluster. Finally, we average these eigenspectra across all clusters at a given iteration level to obtain the representative eigenspectrum for that scale.

In stage (ii), we calculate the eigenspectra of the sub-covariance matrices across different cluster sizes as described in **_Meshulam et al., 2019_**. Let $N_0 = 2^n$ be the original number of neurons. To reduce it to size $N = N_0/2^k = 2^{n-k}$, where $k$ is the $k$th reduction step, consider the coarse-grained neurons in step $n - k$ in stage (i). Each coarse-grained neuron is a cluster of $2^{n-k}$ neurons. We then calculate spectrum of the block of the original covariance matrix corresponding to neurons of each cluster (there are $2^k$ such blocks). Lastly, an average of these $2^k$ spectra is computed.

For example, when reducing from $N_0 = 2^3 = 8$ to $N = 2^{3-1} = 4$ neurons ($k = 1$), we would have two clusters of four neurons each. We calculate the eigenspectrum for each $4 \times 4$ block of the original covariance matrix, then average these two spectra together. To better understand this process through a concrete example, consider a hypothetical scenario where a set of eight neurons, labeled 1,2,3,...,7,8, are subjected to a two-step clustering procedure. In the first step, neurons are grouped based on their maximum correlation pairs, for example, resulting in the formation of four pairs: $\{1, 2\}, \{3, 4\}, \{5, 6\}$, and $\{7, 8\}$ (see **_Appendix 1—figure 4_**). Subsequently, the neurons are further grouped into two clusters based on the results of the RG step mentioned above. Specifically, if the correlation between the coarse-grained variables of the pair $\{1, 2\}$ and the pair $\{3, 4\}$ is found to be the largest among all other pairs of coarse-grained variables, the first group consists of neurons $\{1, 2, 3, 4\}$, while the second group contains neurons $\{5, 6, 7, 8\}$. Next, take the size of the cluster $N = 4$ for example. The eigenspectra of the covariance matrices of the four neurons within each cluster are computed. This results in two eigenspectra, one for each cluster. The correlation matrices used to compute the eigenspectra of different sizes do not involve coarse-grained neurons. It is the real neurons 1,2,3,...,7,8, but with expanding cluster sizes. Finally, the average of the eigenspectra of the two clusters is calculated.

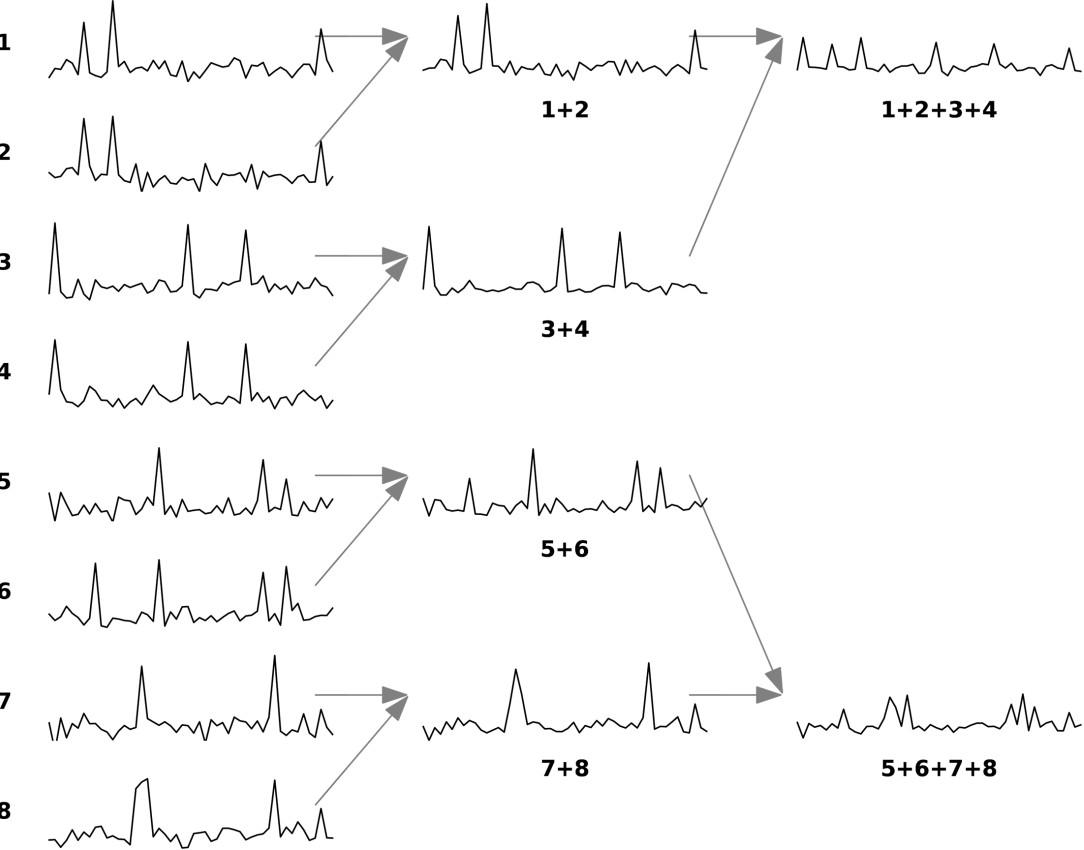

**Appendix 1—figure 4.** Example of renormalization group (RG) approach for a set of eight neurons. The figure is adapted from *Meshulam et al., 2019*. The diagram illustrates the iterative clustering process for eight neurons. In each iteration, neurons are paired based on maximum correlation, with their activities combined through summation and normalized to maintain unit mean for nonzero values. Each neuron can only be paired once per iteration, ensuring all neurons are grouped by the iteration's end.

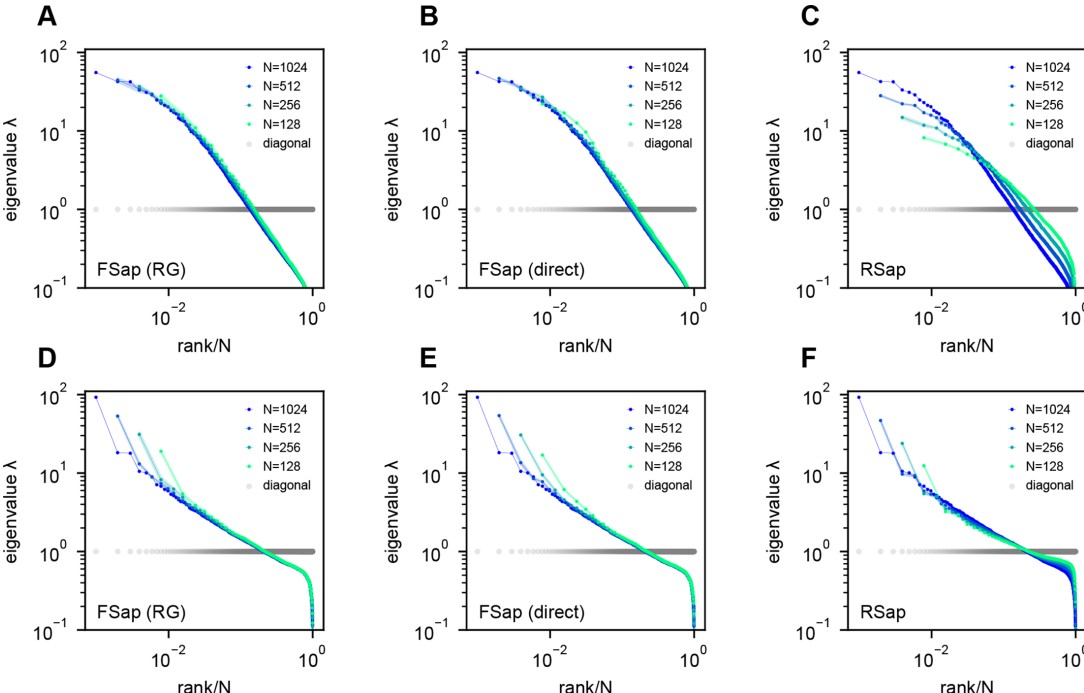

**Appendix 1—figure 5.** Eigenspectra of renormalization group (RG)-inspired clustering, direct functional region sampling (FSap), and random sampling (RSap) in ERM. (**A, D**) RG clustered eigenspectra of ERM. The size of the cluster is denoted by $N$, which is the number of neurons in each cluster. We adopt the RG approach (*Meshulam et al., 2018*; *Meshulam et al., 2019*), but with a specific modification (Appendix 1). (**B, E**) Direct spatial sampling in the functional space (FSap) and the corresponding ERM eigenspectra. We began our analysis with a set of $N_0$ neurons distributed in the functional space. Initially, we chose $N = N_0/2$ neurons that were located exclusively on one side of the x-axis of this space. We then proceeded to select $N = N_0/4$ neurons from 4 quadrants. This sampling process was repeated iteratively, generating successively smaller subsets of neurons. (**C, F**) Random sampled (RSap) eigenspectra of ERM. ERM parameters: (**A–C**) Exponential function $f(\vec{x}) = e^{-\|\vec{x}\|/b}$ where $b = 1$, $\rho = 10.24$ and dimension $d = 2$. (**D–F**) Approximate power law *Equation 11* with $\mu = 0.5$, $\rho = 10.24$ and dimension $d = 2$. Other parameters are the same as *Figure 3*. The standard error of the mean (SEM) across the clusters is represented by the shaded area of each line.

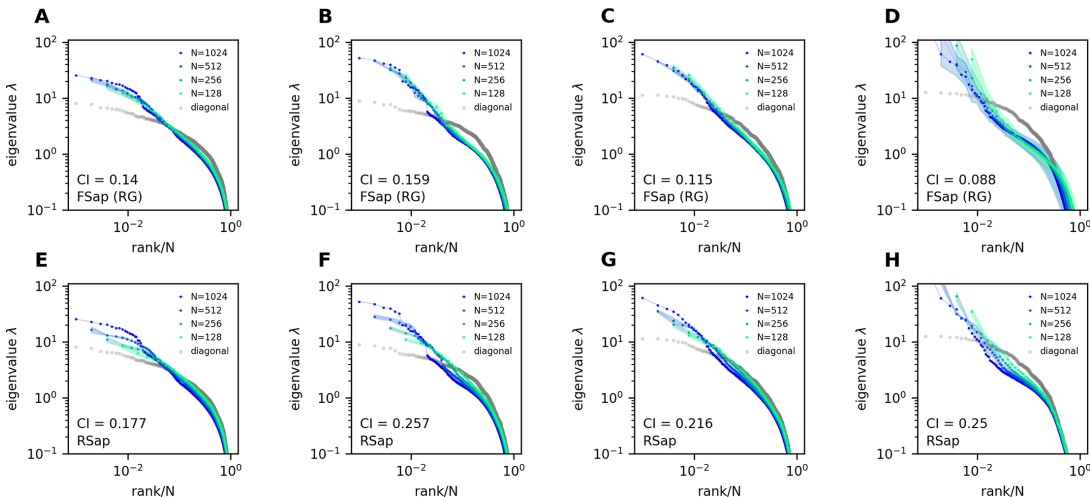

**Appendix 1—figure 6.** Morrell et al.'s latent variable model. (**A–D**) Functional sampled (FSap) eigenspectra of the Morrell et al. model. (**E–H**) Random sampled (RSap) eigenspectra of the same model. Briefly, in Morrell et al.'s latent variable model (*Morrell et al., 2024*; *Morrell et al., 2021*), neural activity is driven by $N_f$ latent fields and a place field. The latent fields are modeled as Ornstein–Uhlenbeck processes with a time constant $\tau$. The parameters

*Appendix 1—figure 6 continued on next page*

*Appendix 1—figure 6 continued*

$\varepsilon$ and $\eta$ control the mean and variance of individual neurons' firing rates, respectively. The following are the parameter values used. (**A, E**) Using the same parameters as in ***Morrell et al., 2021***: $N_f = 10$, $\epsilon = -2.67$, $\eta = 6$, $\tau = 0.1$. Half of the cells are also coupled to the place field. (**B–D, F–H**) Using parameters from ***Morrell et al., 2024***: $N_f = 5$, $\epsilon = -3$, $\eta = 4$. There is no place field. The time constant $\tau = 0.1, 1, 10$ for (**B, F, C, G**) and (**D, H**), respectively.

## Analyzing the effects of removing neural activity data during hunting

To identify and remove the time frames corresponding to putative hunting behaviors, the following procedure was used. The hunting interval was defined as 10 frames (1 s) preceding the onset of an eye convergence (see Methods) to 10 frames after the offset of this eye convergence. These frames were then excluded from the data before recalculating the covariance matrix (see Methods) and subsequently the sampled eigenspectra (***Appendix 1—figure 7B***, ***Appendix 1—figure 8B, D, F, H***). As a control to the removal of the hunting frame, an equal number of time frames that are not within those hunting intervals were randomly selected and then removed and analyzed (***Appendix 1—figure 7C***, ***Appendix 1—figure 8A, C, E, G***). The number of hunting interval frames and total recording frames for five fish exhibiting hunting behaviors are as follows: fish 1 – 268/7495, fish 2 – 565/9774, fish 3 – 2734/13,904, fish 4 – 843/7318, and fish 5 – 1066/7200. Fish 6 (number of time frames: 9388) was not exposed to a prey stimulus and, therefore, was excluded from the analysis.

To assess the impact of hunting removal on CI, we calculated the CI of the covariance matrix using all neurons recorded in each fish (without sampling to 1024 neurons). For the control case, we repeated the removal of the nonhunting frame 10 times to generate 10 covariance matrices and computed their CIs. We used a one-sample *t*-test to determine the level of statistical significance between the control CIs and the CI obtained after removal of the hunting frame.

Using fitted ERM parameters by full data, we performed a MDS on the control data and hunting-removed data to infer the functional coordinates. Note that the functional coordinates inferred by MDS are not unique: rotations and translations give equivalent solutions. For visualization purposes (not needed for analysis), we first used the Umeyama algorithm to optimally align the functional coordinates of control and hunting-removed data.

To identify distinct clusters within the functional coordinates, we fit Gaussian Mixture Models (GMMs) using the 'GaussianMixtures' package in Julia. We chose the number of clusters $K$ based on giving the smallest Bayesian information criterion score. After fitting the GMMs, a list of probabilities $p_{ik}$, $k = 1, 2, \ldots, K$ was given for each neuron $i$ specifying the probability of the neuron belonging to the cluster $k$. The mean and covariance parameters were estimated for each Gaussian distributed cluster. For visualization (but not for analysis), a neuron was colored according to cluster $k^*$ where $k^* = \arg\max_{1 \leq k \leq K} p_{ik}$.

We used the following method to measure the size of the cluster and its fold change. For a 2D (recall $d = 2$ in our ERM) Gaussian distributed cluster, let us consider an ellipse centered on its mean, and its axes are aligned with the eigenvectors of its covariance matrix $C_{2 \times 2}$. Let the eigenvalues of $C$ be $\lambda_1, \lambda_2$. Then we set the length of the half-axis of the ellipse to be $c\sqrt{\lambda_i}$, respectively. Here, $c > 0$ is a constant determined below. Note that the ellipse axes correspond to linear combinations of 2D Gaussian random variables that are independent and $\lambda_i$'s are the variance of these linear combinations. From this fact, it is straightforward to show that the probability that a sample from the Gaussian cluster lies in the above ellipse depends only on $c$, that is, $1 - e^{-\frac{c^2}{2}}$, and not on the shape of the cluster. So, the ellipse represents a region that covers a fixed proportion of neurons for any cluster, and its area can be used as a measure for the size of the Gaussian cluster. Note that the area of the ellipse is $\pi c^2 \sqrt{\lambda_1 \lambda_2} = \pi c^2 \sqrt{\det(C)}$. In ***Appendix 1—figure 9***, we plot the ellipses to help visualize the clusters and their changes. We choose $c$ such that the ellipse covers 95% of the probability (i.e., the fraction of neurons belonging to the cluster).

In the control functional map where we fit the GMMs, we directly calculated the size measure $\pi c^2 \sqrt{\det(C)}$ from the estimated covariance $C$ for each Gaussian cluster. In the hunting-removed functional map, we needed to estimate the covariance $C'$ for neurons belonging to a cluster $k$ under the new coordinates (we assume that the new distribution can still be approximated by a Gaussian distribution). We performed this estimation in a probabilistic manner to avoid issues of highly overlapping clusters where the cluster membership could be ambiguous for some neurons. First, we estimated the center/mean of the new Gaussian distribution by

$$(\bar{x}, \bar{y}) := \left( \frac{\sum_{i=1}^{N} p_{ik}x_i}{\sum_{i=1}^{N} p_{ik}}, \frac{\sum_{i=1}^{N} p_{ik}y_i}{\sum_{i=1}^{N} p_{ik}} \right).$$

Here, the summation goes over all the $N$ neurons in the functional space and $p_{ik}$ is the membership probability defined above, and $(x_i, y_i)$ is the coordinate of neuron $i$ in the hunting-removed map. Similarly, we can use a weighted average to estimate the entries in the covariance matrix $C' = \begin{bmatrix} C'_{xx} & C'_{xy} \\ C'_{yx} & C'_{yy} \end{bmatrix}$. For example,

$$\hat{C}'_{xy} := \frac{\sum_{i=1}^{N} p_{ik} (x_i - \bar{x}) (y_i - \bar{y})}{\sum_{i=1}^{N} p_{ik}}.$$

Then we calculated the size of the cluster on the new map as $\pi c^2 \sqrt{\det(\hat{C}')}$. Finally, we computed the fold change in size as $\sqrt{\frac{\det(\hat{C}')}{\det(C)}}$.

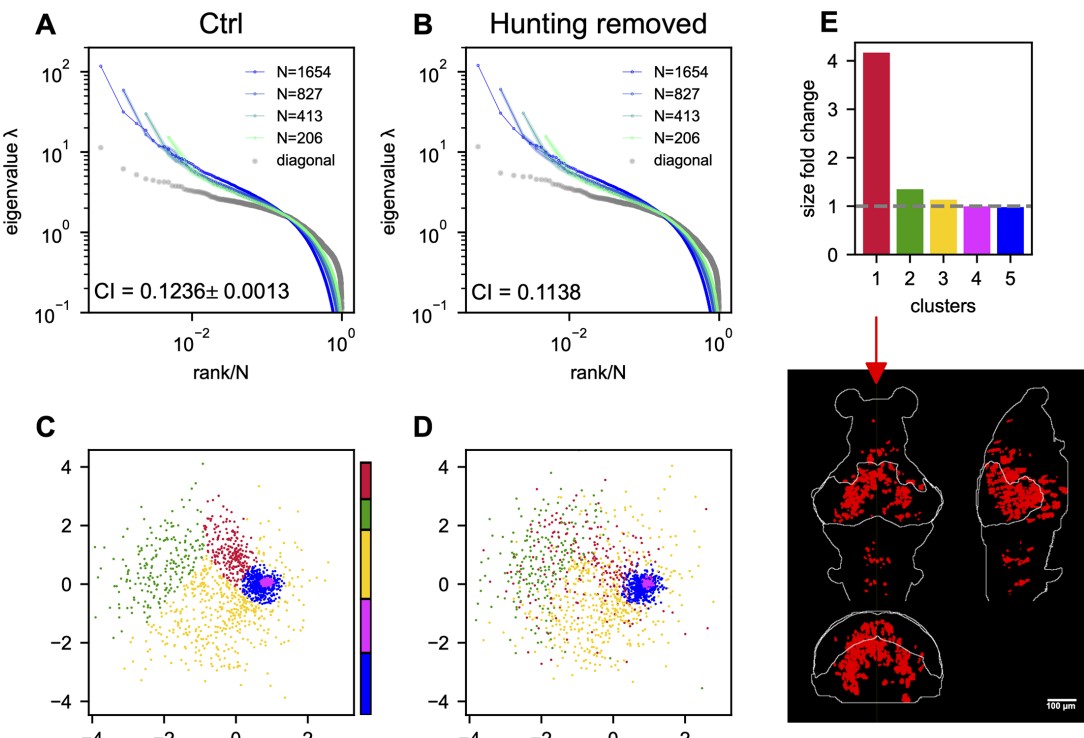

**Appendix 1—figure 7.** The effects of hunting behavior on scale invariance and functional space organization. Sampled covariance eigenspectra of the data from fish 1 calculated from control (**A**) and hunting removed (**B**) data. Ctrl: We randomly remove the same number of non-hunting frames. This process is repeated 10 times, and the mean ± SD of the CI is shown in the plot. Hunting removed: The time frames corresponding to the eye-converged intervals (putative hunting state) are removed when calculating the covariance (Appendix 1). The CI for the hunting-removed data appears to be statistically smaller than in the control case (p-value = $1.5 \times 10^{-9}$). (**C**) Functional space organization of control data. The neurons are clustered using the Gaussian Mixture Models (GMMs) and their cluster memberships are shown by the color. The color bar represents the proportion of neurons that belong to each cluster. (**D**) Similar to (**C**) but the functional coordinates are inferred from the hunting-removed data. The color code of each neuron is the same as that of the control data (**C**), which allows for a comparison of the changes to the clusters under the hunting-removed condition. See also the Movie S1. (**E**) Fold change in size/area (Appendix 1) for each cluster (top; the gray dashed line represents a fold change of 1, i.e., no change in size) and the anatomical distribution of the most dispersed cluster (bottom).

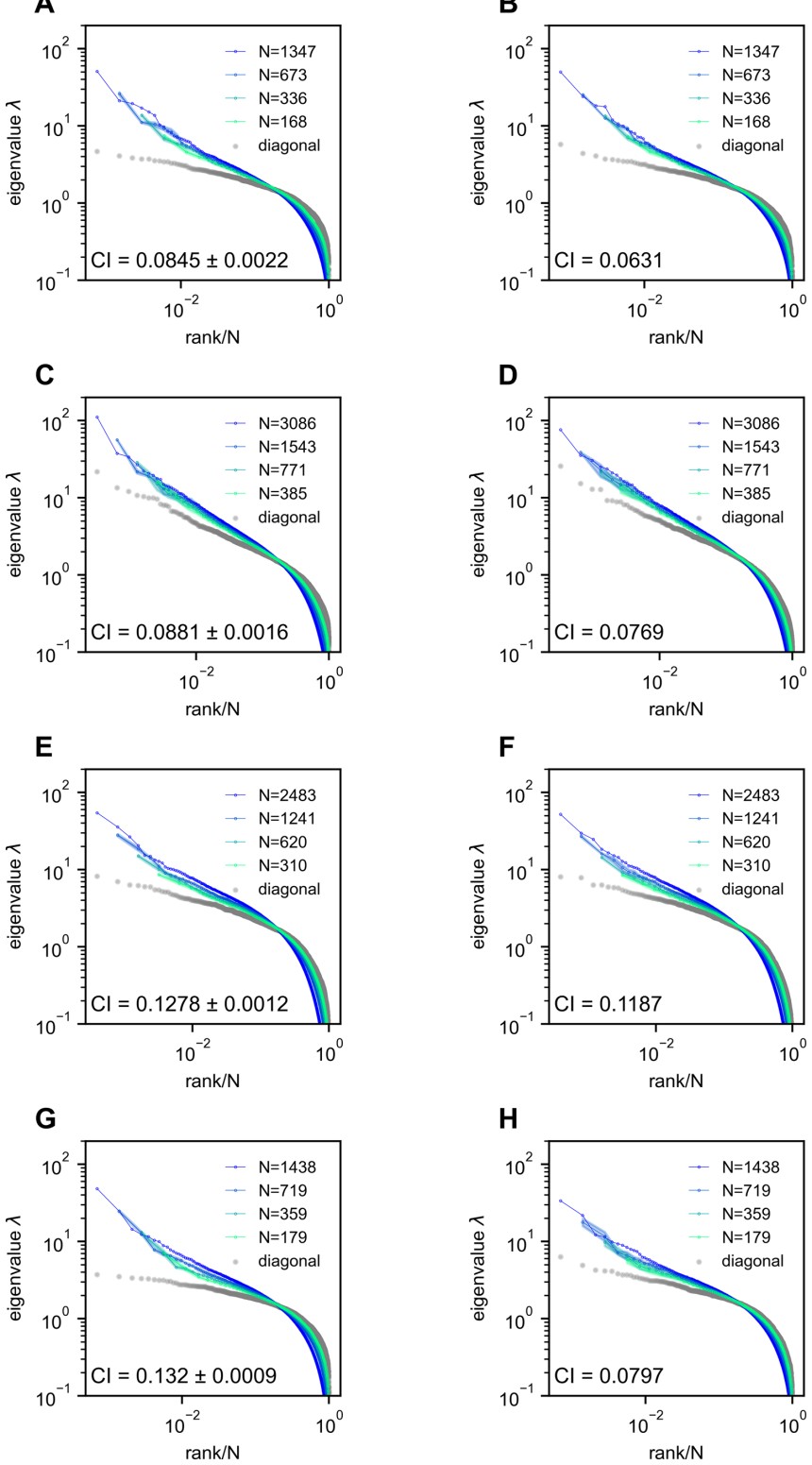

**Appendix 1—figure 8.** Removing the time segment of hunting behavior does not obliterate the scale-invariant eigenspectra. Rows correspond to four light-field zebrafish data: fish 2 to fish 5 (results for fish 1 have been shown in *Appendix 1—figure 7*). (**A, C, E, G**) Ctrl: we randomly remove the same number of time frames that are *not* the putative hunting frames. We repeat this process 10 times to generate 10 control covariance matrices and the CI is represented by mean ± SD. (**B, D, F, H**) Hunting removed: data obtained by removing hunting frames from the full data (Appendix 1). The CI for the hunting removed data appears to be significantly smaller than that of the

*Appendix 1—figure 8 continued on next page*

*Appendix 1—figure 8 continued*

control case (one-sample *t*-test $p = 2.2 \times 10^{-10}$ in fish 2, $p = 4.6 \times 10^{-9}$ in fish 3, $p = 1.7 \times 10^{-9}$ in fish 4, and $p = 3.4 \times 10^{-17}$ in fish 5).

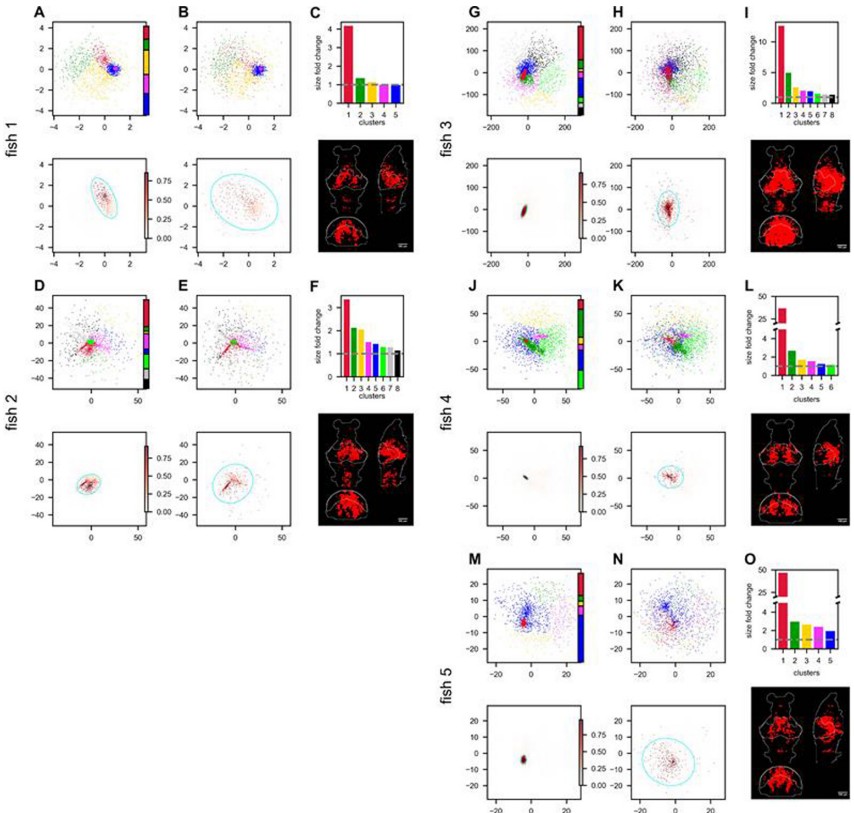

**Appendix 1—figure 9.** Hunting behavior reorganizes neurons in the functional space (continued on next page). Rows correspond to five light-field recordings of zebrafish engaged in hunting behavior: fish 1 to fish 5. (**A, D, G, J, M**) (top) Functional space organization of the control data inferred by fitting the ERM and MDS (Result). Neurons are clustered using the Gaussian Mixture Models (GMMs) and their cluster memberships are shown by the color. The colorbar represents the proportion of neurons belonging to each cluster. (**A, D, G, J, M**) (bottom) The coordinate distribution of the cluster in control data which is most dispersed (i.e., largest fold change in size, see below) after hunting-removal. The transparency of the dots (colorbar) is proportional to the probability of the neurons belonging to this cluster (Appendix 1). The cyan ellipse serves as a visual aid for the cluster size: it encloses 95% of the neurons belonging to that cluster (Appendix 1). (**B, E, H, K, N**) (top) Similar to (**A, D, G, J, M**) (top) but the functional coordinates are inferred from the hunting-removed data. The color code of each neuron is the same as that in the control data, which allows for a comparison of the changes to the clusters under the hunting-removed condition. (**B, E, H, K, N**) (bottom) Similar to (**A, D, G, J, M**) (bottom) but the functional coordinates are inferred from the hunting-removed data. The transparency of each neuron is the same as in (**A, D, G, J, M**). (bottom), and it represents the probability $p_{ik}$ (Appendix 1) of neurons belonging to the most dispersed cluster $k$ in the control data. Likewise, the cyan ellipse encloses 95% of the neurons belonging to that cluster (Appendix 1). (**C, F, I, L, O**) Top, size/area fold change (Appendix 1) for each cluster (the gray dashed line represents a fold change of 1, i.e., no change in size); bottom, the anatomical distribution of the neurons in the most dispersed cluster.

# Fish 1 example trial

**Appendix 1—video 1.** Neural activity patterns in anatomical and functional space during hunting (click here). Single-trial examples of fish 1 and fish 3. (A) Inferred firing rate activity in anatomical space. Scale bar, 100 μm. (B) Inferred firing rate activity in functional space. Functional space organization of the control data inferred by fitting the ERM and MDS in Result. The cyan ellipse serves as a visual aid for the cluster size: it encloses 95% of the neurons belonging to that cluster (Appendix 1). The inset illustrates the functional space organization, similar to that shown in *Appendix 1—figure 7C*. The colorbars in panels A and B depict the inferred activity magnitude of individual neurons. (C) Simultaneous behavior recording alongside the neural activity. Time, seconds.
https://elifesciences.org/articles/100666/figures#video1

# Appendix 2

In this appendix, we elaborate upon the sketch introduced in the Methods, and present a full derivation of the covariance eigenspectrum of our ERM model. This section is organized as follows. First, we will briefly introduce the relationship between the eigenvalue probability density distribution and the resolvent. Second, we will turn the problem of calculating the resolvent to a calculation of the partition function using a field-theoretic representation and proceed to manipulate the partition function using the replica method. Third, we will introduce two approximate methods for calculating the partition function, leading to the high-density theory and the Gaussian variational method. We will discuss the implications and predictions of each method. Finally, we will discuss the relationship between the two methods and identify the parameter regime where the high-density theory agrees with the numerical simulation. Notation table: *Appendix 2—table 1*.

## Resolvent

The eigenvalues $\lambda_n$ of a Hermitian matrix $C$ are real. Their probability density function or eigendensity is formally given by

$$p(\lambda) = \frac{1}{N} \left\langle \sum_{n=1}^{N} \delta(\lambda - \lambda_n) \right\rangle, \tag{S4}$$

where $\langle \ldots \rangle$ represents an average across different realizations of $C$. The eigendensity is connected with the resolvent (*Mézard et al., 1999*; *Goetschy and Skipetrov, 2013*),

$$g(z) = \frac{1}{N} \left\langle \text{Tr} \frac{1}{z - C} \right\rangle = \frac{1}{N} \left\langle \sum_{n=1}^{N} \frac{1}{z - \lambda_n} \right\rangle, \tag{S5}$$

we therefore compute the eigendensity using the standard inverse formula of Stieltjes tranform:

$$p(\lambda) = -\frac{1}{\pi} \lim_{\eta \to 0^+} \text{Im } g(\lambda + i\eta) \tag{S6}$$

## Field representation

In this section, we discuss a field-theoretical representation of the resolvent $g(z)$. First, we rewrite *Equation S5* as

$$g(z) = -\frac{2}{N} \partial_z \left\langle \ln \left[ \left( \det(z - C) \right)^{-1/2} \right] \right\rangle \tag{S7}$$

The determinant $\left( \det(z - C) \right)^{-1/2}$ can be represented as a Gaussian integral

$$\Xi(z) = \left( \det(z - C) \right)^{-1/2} = i^{-N/2} \int_{-\infty}^{+\infty} \frac{d\phi_1}{\sqrt{2\pi}} \ldots \frac{d\phi_N}{\sqrt{2\pi}} \exp \left[ -\frac{i}{2} \Phi^T (z - C) \Phi \right], \tag{S8}$$

where $\Phi = [\phi_1, \ldots, \phi_N]^T$, and $i \equiv \sqrt{-1}$.

$$\ln \Xi(z) = \ln \int_{-\infty}^{+\infty} \frac{d\phi_1}{\sqrt{2\pi}} \ldots \frac{d\phi_N}{\sqrt{2\pi}} \exp \left[ -\frac{i}{2} \Phi^T (z - C) \Phi \right] - \frac{i\pi N}{4} \tag{S9}$$

We thus establish a relationship between the resolvent and $\Xi$

$$g(z) = -\frac{2}{N} \partial_z \left\langle \ln \Xi(z) \right\rangle \tag{S10}$$

Note that the constant term in *Equation S9* can be killed by $\partial_z$ and we will ignore it in the sequel. *Equation S10* is the central formula in this note. $\Xi(z)$ is also called the partition function in statistical physics. We endeavor to find a way to compute the average of $\ln \Xi(z)$.

Recall that in our ERM model (Result *Equation S2* and *Figure 3A*), the covariance between neuron $i$ and neuron $j$ is determined by the distance kernel function and their neural activity variances:

$$C_{ij} = f(\vec{x}_i - \vec{x}_j)\sigma_i\sigma_j, \tag{S11}$$

where $\vec{x}_i$ are sampled from a uniform coordinate distribution $p(\vec{x}_i) = 1/V$; $\sigma_i$ are i.i.d. chosen from a probability density distribution $p(\sigma)$ and are independent of the neuron coordinates $\vec{x}_i$. The $\langle \dots \rangle$ in **Equation S10** is therefore an average over all possible $\vec{x}_i$ and $\sigma_i$. In order to compute $\langle \ln \Xi(z) \rangle$, we apply the replica method based on a smart use of the identity

$$\ln x = \lim_{n \to 0} \frac{x^n - 1}{n}$$

**Equation S10** now becomes

$$g(z) = -\frac{2}{N} \partial_z \left[ \lim_{n \to 0} \frac{1}{n} \left\langle \Xi^n(z) - 1 \right\rangle \right] = -\frac{2}{N} \partial_z \left[ \lim_{n \to 0} \frac{1}{n} \ln \left\langle \Xi^n(z) \right\rangle \right] \tag{S12}$$

The idea is to compute the right-hand side for finite and integer $n$ and then perform the analytic continuation to $n \to 0$.

Now we seek to determine the value of $\langle \Xi^n(z) \rangle$. It contains $n$ copies (replicas) of the original system

$$\left\langle \Xi^n(z) \right\rangle = \left(\frac{1}{2\pi}\right)^{\frac{Nn}{2}} \int_{-\infty}^{+\infty} (d\phi_1^1 \dots d\phi_1^n) \dots (d\phi_N^1 \dots d\phi_N^n) \left\langle \exp\left[ -\frac{i}{2} \sum_{\alpha=1}^{n} \Phi^{\alpha T}(z - C)\Phi^\alpha \right] \right\rangle. \tag{S13}$$

Writing it down explicitly, we have

$$\left\langle \Xi^n(z) \right\rangle = \left(\frac{1}{2\pi}\right)^{\frac{Nn}{2}} \int_{-\infty}^{+\infty} (d\phi_1^1 \dots d\phi_1^n) \dots (d\phi_N^1 \dots d\phi_N^n) \int_{-L}^{L} \frac{d^d\vec{x}_1}{V} \dots \frac{d^d\vec{x}_N}{V} \int p(\sigma_1)d\sigma_1 \dots p(\sigma_N)d\sigma_N$$
$$\exp\left[ -\frac{zi}{2} \sum_{\alpha=1}^{n} \sum_{j=1}^{N} (\phi_j^\alpha)^2 + \frac{i}{2} \sum_{\alpha=1}^{n} \sum_{j,k=1}^{N} \phi_j^\alpha \phi_k^\alpha f(\vec{x}_j - \vec{x}_k)\sigma_j\sigma_k \right] \tag{S14}$$

In order to proceed further, we introduce the following auxiliary fields:

$$\psi^\alpha(\vec{x}) = \sum_{j=1}^{N} \phi_j^\alpha \delta(\vec{x} - \vec{x}_j) \tag{S15}$$

**Equation S15** can be represented as a following functional integral

$$1 = \int_{-\infty}^{+\infty} \prod_{\alpha=1}^{n} D[\psi^\alpha] \delta_F[\psi^\alpha(\vec{x}) - \sum_{j=1}^{N} \phi_j^\alpha \delta(\vec{x} - \vec{x}_j)] \tag{S16}$$

$$\delta_F[\psi] = \int_{-\infty}^{+\infty} D[\hat{\psi}] \exp[i \int_{-\infty}^{+\infty} d^d\vec{x} \psi(\vec{x})\hat{\psi}(\vec{x})] \tag{S17}$$

or we can combine **Equations S16 and S17** as

$$1 = \int_{-\infty}^{+\infty} \int_{-\infty}^{+\infty} \prod_{\alpha=1}^{n} D[\hat{\psi}^\alpha] D[\psi^\alpha] \exp\left[ i \int_{-\infty}^{+\infty} d^d\vec{x}[\psi^\alpha(\vec{x}) - \sum_{j=1}^{N} \phi_j^\alpha \delta(\vec{x} - \vec{x}_j)]\hat{\psi}^\alpha(\vec{x}) \right] \tag{S18}$$

Using **Equation S15**, we can write the term $\frac{1}{2} \sum_{j,k=1}^{N} \phi_j^\alpha \phi_k^\alpha f(\vec{x}_j - \vec{x}_k)$ in **Equation S14** as

$$\frac{1}{2} \sum_{j,k=1}^{N} \phi_j^\alpha \phi_k^\alpha f(\vec{x}_j - \vec{x}_k) = \frac{1}{2} \int_{-\infty}^{+\infty} d\vec{x} d\vec{x}' f(\vec{x} - \vec{x}') \psi^\alpha(\vec{x}) \psi^\alpha(\vec{x}') \tag{S19}$$

We insert the relation **Equations S18 and S19** into **Equation S14**,

$$\langle \Xi^n(z) \rangle = (\frac{1}{2\pi})^{\frac{Nn}{2}} \int_{-\infty}^{+\infty} (d\phi_1^1...d\phi_1^n)...(d\phi_N^1...d\phi_N^n) \int_{-L}^{L} \frac{d^d\vec{x}_1}{V}...\frac{d^d\vec{x}_N}{V} \int p(\sigma_1)d\sigma_1...p(\sigma_N)d\sigma_N$$

$$\exp\left[-\frac{zi}{2}\sum_{\alpha=1}^{n}\sum_{j=1}^{N}(\phi_j^\alpha)^2 + \frac{i}{2}\sum_{\alpha=1}^{n}\sum_{j,k=1}^{N}\phi_j^\alpha\phi_k^\alpha f(\vec{x}_j - \vec{x}_k)\sigma_j\sigma_k\right]$$

$$\int_{-\infty}^{+\infty}\int_{-\infty}^{+\infty}\prod_{\alpha=1}^{n}D[\psi^\alpha]D[\hat\psi^\alpha]\exp\left[i\int_{-\infty}^{+\infty}d^d\vec{x}(\psi^\alpha(\vec{x}) - \sum_{j=1}^{N}\phi_j^\alpha\delta(\vec{x}-\vec{x}_j)\sigma_j)\hat\psi^\alpha(\vec{x})\right]$$

$$= (\frac{1}{2\pi})^{\frac{Nn}{2}}\int_{-\infty}^{+\infty}\prod_{\alpha=1}^{n}D[\psi^\alpha]D[\hat\psi^\alpha]\int_{-\infty}^{+\infty}(d\phi_1^1...d\phi_1^n)...(d\phi_N^1...d\phi_N^n)$$

$$\int_{-L}^{L}\frac{d^d\vec{x}_1}{V}...\frac{d^d\vec{x}_N}{V}\int p(\sigma_1)d\sigma_1...p(\sigma_N)d\sigma_N$$

$$\exp\left[-\frac{zi}{2}\sum_{\alpha=1}^{n}\sum_{j=1}^{N}(\phi_j^\alpha)^2 + \frac{i}{2}\sum_{\alpha=1}^{n}\sum_{j,k=1}^{N}\phi_j^\alpha\phi_k^\alpha f(\vec{x}_j - \vec{x}_k)\sigma_j\sigma_k\right]$$

$$\exp\left[i\sum_{\alpha=1}^{n}\int_{-\infty}^{+\infty}d^d\vec{x}(\psi^\alpha(\vec{x}) - \sum_{j=1}^{N}\phi_j^\alpha\delta(\vec{x}-\vec{x}_j)\sigma_j)\hat\psi^\alpha(\vec{x})\right]$$

(S20)

$$= (\frac{1}{2\pi})^{\frac{Nn}{2}}\int_{-\infty}^{+\infty}\prod_{\alpha=1}^{n}D[\psi^\alpha]D[\hat\psi^\alpha]\exp\left[\frac{i}{2}\sum_{\alpha=1}^{n}\int_{-\infty}^{+\infty}d\vec{x}d\vec{x}'f(\vec{x}-\vec{x}')\psi^\alpha(\vec{x})\psi^\alpha(\vec{x}')\right]$$

$$\int_{-\infty}^{+\infty}(d\phi_1^1...d\phi_1^n)...(d\phi_N^1...d\phi_N^n)\int_{-L}^{L}\frac{d^d\vec{x}_1}{V}...\frac{d^d\vec{x}_N}{V}\int p(\sigma_1)d\sigma_1...p(\sigma_N)d\sigma_N$$

$$\exp\left[-\frac{zi}{2}\sum_{\alpha=1}^{n}\sum_{j=1}^{N}(\phi_j^\alpha)^2 + i\sum_{\alpha=1}^{n}\int_{-\infty}^{+\infty}d^d\vec{x}(\psi^\alpha(\vec{x}) - \sum_{j=1}^{N}\phi_j^\alpha\delta(\vec{x}-\vec{x}_j)\sigma_j)\hat\psi^\alpha(\vec{x})\right]$$

$$= (\frac{1}{2\pi})^{\frac{Nn}{2}}\int_{-\infty}^{+\infty}\prod_{\alpha=1}^{n}D[\psi^\alpha]D[\hat\psi^\alpha]\exp\left[\frac{i}{2}\sum_{\alpha=1}^{n}\int_{-\infty}^{+\infty}d\vec{x}d\vec{x}'f(\vec{x}-\vec{x}')\psi^\alpha(\vec{x})\psi^\alpha(\vec{x}')\right]$$

$$\exp\left[i\sum_{\alpha=1}^{n}\int_{-\infty}^{+\infty}d^d\vec{x}\psi^\alpha(\vec{x})\hat\psi^\alpha(\vec{x})\right]$$

$$\int_{-\infty}^{+\infty}(d\phi_1^1...d\phi_1^n)...(d\phi_N^1...d\phi_N^n)\int_{-L}^{L}\frac{d^d\vec{x}_1}{V}...\frac{d^d\vec{x}_N}{V}\int p(\sigma_1)d\sigma_1...p(\sigma_N)d\sigma_N$$

$$\exp\left[-\frac{zi}{2}\sum_{\alpha=1}^{n}\sum_{j=1}^{N}(\phi_j^\alpha)^2 - i\sum_{\alpha=1}^{n}\int_{-\infty}^{+\infty}d^d\vec{x}\sum_{j=1}^{N}\phi_j^\alpha\delta(\vec{x}-\vec{x}_j)\sigma_j\hat\psi^\alpha(\vec{x})\right]$$

Integrating the last term in *Equation S20*

$$\int_{-\infty}^{+\infty}d\phi_i^1...d\phi_i^n\int_{-L}^{L}\frac{d^d r_i}{V}\int d\sigma_i p(\sigma_i)\exp\left[-\frac{zi}{2}\sum_{\alpha=1}^{n}(\phi_i^\alpha)^2 - i\sum_{\alpha=1}^{n}\int_{-\infty}^{+\infty}d^d r\phi_i^\alpha\delta(r-r_i)\sigma_i\hat\psi^\alpha(r)\right]$$

$$= \int_{-L}^{L}\frac{d^d r_i}{V}\int_{-\infty}^{+\infty}d\phi_i^1...d\phi_i^n\int d\sigma_i p(\sigma_i)\exp\left[-\frac{zi}{2}\sum_{\alpha=1}^{n}(\phi_i^\alpha)^2 - i\sum_{\alpha=1}^{n}\phi_i^\alpha\sigma_i\hat\psi^\alpha(r_i)\right]$$

(S21)

$$= (\frac{2\pi}{zi})^{\frac{n}{2}}\int_{-L}^{L}\frac{d^d r_i}{V}\int d\sigma_i p(\sigma_i)\exp\left[\frac{i}{2z}\sum_{\alpha=1}^{n}\hat\psi^\alpha(r_i)^2\sigma_i^2\right]$$

$$= (\frac{2\pi}{zi})^{\frac{n}{2}}\int_{-L}^{L}\frac{d^d r}{V}\int d\sigma p(\sigma)\exp\left[\frac{i}{2z}\sum_{\alpha=1}^{n}\hat\psi^\alpha(r)^2\sigma^2\right]$$

so that $\langle \Xi^n(z) \rangle$ from *Equation S14* can be written as

$$\langle \Xi^n(z) \rangle = \int_{-\infty}^{+\infty}\prod_{\alpha=1}^{n}D[\psi^\alpha]D[\hat\psi^\alpha]A^N e^{S_0}$$

(S22)

$$\text{where} \quad A = \int_{-L}^{L} \frac{\mathrm{d}^d \vec{x}}{V} (zi)^{-\frac{n}{2}} \int \mathrm{d}\sigma p(\sigma) \exp\left[ \frac{i}{2z} \sum_{\alpha=1}^{n} \hat{\psi}^{\alpha}(\vec{x})^2 \sigma^2 \right] \tag{S23}$$

$$\text{and} \quad S_0 = \frac{i}{2} \sum_{\alpha=1}^{n} \int_{-\infty}^{+\infty} \mathrm{d}\vec{x}\mathrm{d}\vec{x}' f(\vec{x} - \vec{x}') \psi^{\alpha}(\vec{x})\psi^{\alpha}(\vec{x}') + i \sum_{\alpha=1}^{n} \int_{-\infty}^{+\infty} \mathrm{d}^d \vec{x} \psi^{\alpha}(\vec{x})\hat{\psi}^{\alpha}(\vec{x}) \tag{S24}$$

Integrating out the $\psi^{\alpha}$ in $\langle \Xi^n(z) \rangle$ *Equations S22 and S24*

$$\int_{-\infty}^{+\infty} D[\psi^{\alpha}] \exp\left[ \frac{i}{2} \int_{-\infty}^{+\infty} \mathrm{d}\vec{x}\mathrm{d}\vec{x}' f(\vec{x} - \vec{x}') \psi^{\alpha}(\vec{x})\psi^{\alpha}(\vec{x}') + i \int_{-\infty}^{+\infty} \mathrm{d}^d \vec{x} \psi^{\alpha}(\vec{x})\hat{\psi}^{\alpha}(\vec{x}) \right]$$
$$= (2\pi i)^{N/2} (\det f)^{-1/2} \exp\left[ -\frac{i}{2} \int_{-\infty}^{+\infty} \mathrm{d}\vec{x}\mathrm{d}\vec{x}' f^{-1}(\vec{x} - \vec{x}') \hat{\psi}^{\alpha}(\vec{x})\hat{\psi}^{\alpha}(\vec{x}') \right] \tag{S25}$$

Here, $f^{-1}$ is the inverse kernel satisfying:

$$\int_{-\infty}^{+\infty} \mathrm{d}\vec{x}'' f(\vec{x} - \vec{x}'') f^{-1}(\vec{x}'' - \vec{x}') = \delta(\vec{x} - \vec{x}') \tag{S26}$$

so that $\langle \Xi^n(z) \rangle$ can be written as

$$\langle \Xi^n(z) \rangle = (2\pi i)^{\frac{Nn}{2}} (\det f)^{-n/2} \int_{-\infty}^{+\infty} D[\hat{\psi}^{\alpha}] e^{S_1} \tag{S27}$$

$$S_1 = N \ln A - \frac{i}{2} \sum_{\alpha=1}^{n} \int_{-\infty}^{+\infty} \mathrm{d}\vec{x}\mathrm{d}\vec{x}' f^{-1}(\vec{x} - \vec{x}') \hat{\psi}^{\alpha}(\vec{x})\hat{\psi}^{\alpha}(\vec{x}') \tag{S28}$$

The constant term $(2\pi i)^{\frac{Nn}{2}}$ of $\langle \Xi^n(z) \rangle$ can be ignored because we should compute $\partial_z \langle \ln \Xi(z) \rangle$ *Equation S10* in the end.

To ensure the mathematical rigor in *Equation S45*, we next apply the Wick rotation $\psi^{\alpha}(\vec{x}) \to \psi^{\alpha}(\vec{x}) e^{-i\frac{\pi}{4}}$ (Appendix 2).

$$\langle \Xi^n(z) \rangle = (\det f)^{-n/2} \int_{-\infty}^{+\infty} D[\hat{\psi}^{\alpha}] e^{S_1} \tag{S29}$$

$$S_1 = N \ln A - \frac{1}{2} \sum_{\alpha=1}^{n} \int_{-\infty}^{+\infty} \mathrm{d}\vec{x}\mathrm{d}\vec{x}' f^{-1}(\vec{x} - \vec{x}') \hat{\psi}^{\alpha}(\vec{x})\hat{\psi}^{\alpha}(\vec{x}') \tag{S30}$$

$$A = \int_{-L}^{L} \frac{\mathrm{d}^d \vec{x}}{V} (z)^{-\frac{n}{2}} \int \mathrm{d}\sigma p(\sigma) \exp\left[ \frac{1}{2z} \sum_{\alpha=1}^{n} \hat{\psi}^{\alpha}(\vec{x})^2 \sigma^2 \right] \tag{S31}$$

## High-density expansion

In this section, we directly calculate the canonical partition function $\langle \Xi^n(z) \rangle$ in the $z \to \infty$ limit by approximating the term $N \ln A$ (*Equation S30*) to a quadratic action, from which the partition function (*Equation S29*) would become a Gaussian integral.

Let us first calculate the $A^N$ in $z \to \infty$ limit

$$\lim_{z \to \infty} A \approx (z)^{-\frac{n}{2}} \int \mathrm{d}\sigma p(\sigma) \left[ 1 + \int_{-L}^{L} \frac{\mathrm{d}^d \vec{x}}{V} \frac{1}{2z} \sum_{\alpha=1}^{n} \hat{\psi}^{\alpha}(\vec{x})^2 \sigma^2 \right]$$
$$= (z)^{-\frac{n}{2}} \left[ 1 + \int \mathrm{d}\sigma p(\sigma) \sigma^2 \int_{-L}^{L} \frac{\mathrm{d}^d \vec{x}}{V} \frac{1}{2z} \sum_{\alpha=1}^{n} \hat{\psi}^{\alpha}(\vec{x})^2 \right]$$
$$= (z)^{-\frac{n}{2}} \left[ 1 + \mathrm{E}(\sigma^2) \int_{-L}^{L} \frac{\mathrm{d}^d \vec{x}}{V} \frac{1}{2z} \sum_{\alpha=1}^{n} \hat{\psi}^{\alpha}(\vec{x})^2 \right] \tag{S32}$$

$$\lim_{z\to\infty} A^N = \lim_{z\to\infty} (z)^{-\frac{Nn}{2}} \left[ 1 + N\mathrm{E}(\sigma^2) \int_{-L}^{L} \frac{\mathrm{d}^d \vec{x}}{V} \frac{1}{2z} \sum_{\alpha=1}^{n} \hat{\psi}^\alpha(\vec{x})^2 \right]$$

$$= \lim_{z\to\infty} (z)^{-\frac{Nn}{2}} \left[ 1 + N\mathrm{E}(\sigma^2) \sum_{\alpha=1}^{n} \int_{-L}^{L} \frac{\mathrm{d}^d \vec{x}}{V} \frac{1}{2z} \hat{\psi}^\alpha(\vec{x})^2 \right] \tag{S33}$$

$$\approx (z)^{-\frac{Nn}{2}} \exp\left[ \mathrm{E}(\sigma^2) \int_{-L}^{L} \frac{\mathrm{d}^d \vec{x}}{V} \frac{N}{2z} \sum_{\alpha=1}^{n} \hat{\psi}^\alpha(\vec{x})^2 \right]$$

Now let us calculate $\langle \Xi^n(z) \rangle$ (*Equation S29–S31*) by letting $L \to \infty$

$$\langle \Xi^n(z) \rangle = (\det f)^{-n/2} (z)^{-\frac{Nn}{2}} \int_{-\infty}^{+\infty} D[\hat{\psi}^\alpha] e^{S_h} \tag{S34}$$

where the high-density quadratic action

$$S_h = \mathrm{E}(\sigma^2) \int_{-\infty}^{\infty} \frac{\mathrm{d}^d \vec{x}}{V} \frac{N}{2z} \sum_{\alpha=1}^{n} \hat{\psi}^\alpha(\vec{x})^2 - \frac{1}{2} \sum_{\alpha=1}^{n} \int_{-\infty}^{+\infty} \mathrm{d}\vec{x}\mathrm{d}\vec{x}' f^{-1}(\vec{x} - \vec{x}') \hat{\psi}^\alpha(\vec{x}) \hat{\psi}^\alpha(\vec{x}')$$

$$= -\frac{1}{2} \sum_{\alpha=1}^{n} \int_{-\infty}^{+\infty} \mathrm{d}\vec{x}\mathrm{d}\vec{x}' G^{-1}(\vec{x} - \vec{x}') \hat{\psi}^\alpha(\vec{x}) \hat{\psi}^\alpha(\vec{x}') \tag{S35}$$

where $G^{-1}(\vec{x} - \vec{y}) = f^{-1}(\vec{x} - \vec{y}) - \frac{N\mathrm{E}(\sigma^2)}{Vz}\delta(\vec{x} - \vec{y})$. Next, by integrating out the $\hat{\psi}$ field, we find

$$\langle \Xi^n(z) \rangle = (\det f)^{-n/2} (z)^{-\frac{Nn}{2}} \int_{-\infty}^{+\infty} D[\hat{\psi}^\alpha] e^{S_h} \tag{S36}$$

$$= (z^N \det f \det(G^{-1}))^{-n/2}$$

Using *Equation S12* that connects the partition function with the resolvent, we have

$$g(z) = -\frac{2}{N}\partial_z \left[ \lim_{n\to0} \frac{1}{n} \ln\left( (\det(zfG^{-1}))^{-n/2} \right) \right]$$

$$= \frac{V}{N}\partial_z \int_{-\infty}^{+\infty} \frac{\mathrm{d}^d \vec{k}}{(2\pi)^d} \ln\left( z - \frac{N\mathrm{E}(\sigma^2)\tilde{f}(\vec{k})}{V} \right) \tag{S37}$$

$$= \frac{1}{\rho} \int_{-\infty}^{+\infty} \frac{\mathrm{d}^d \vec{k}}{(2\pi)^d} \frac{1}{z - \rho\mathrm{E}(\sigma^2)\tilde{f}(\vec{k})}$$

where $\tilde{f}(\vec{k})$ is the Fourier transform of $f(\vec{x})$.

Finally, the eigendensity $p(\lambda)$ (*Equation S6*) is given by

$$p(\lambda) = -\frac{1}{\pi} \lim_{\eta\to0^+} \mathbf{Im}(g(\lambda + i\eta))$$

$$= \frac{1}{\rho} \int_{-\infty}^{+\infty} \frac{\mathrm{d}^d \vec{k}}{(2\pi)^d} \delta(\lambda - \rho\mathrm{E}(\sigma^2)\tilde{f}(\vec{k})) \tag{S38}$$

$$= \frac{1}{\rho\mathrm{E}(\sigma^2)} \int_{-\infty}^{+\infty} \frac{\mathrm{d}^d \vec{k}}{(2\pi)^d} \delta\left( \frac{\lambda}{\mathrm{E}(\sigma^2)} - \rho\tilde{f}(\vec{k}) \right)$$

## Derivation of power-law eigenspectrum in high-density limit

Here we calculate the eigendensity of our model, with the kernel function $f(\vec{x})$ (*Table 3*). The *Equation S38* (set $\mathrm{E}(\sigma^2) = 1$ as in Result) can be written as:

$$p(\lambda) = \frac{S_{d-1}}{(2\pi)^d} \frac{\|\vec{k}_0\|^{d-1}}{\rho^2 |\tilde{f}'(\vec{k}_0)|}, \qquad \|\vec{k}_0\| = \tilde{f}^{-1}(\frac{\lambda}{\rho}) \tag{S39}$$

where $S_{d-1}$ is the surface area of $d-1$ dimensional sphere. Here, we consider the approximation $f(\vec{x}) \approx \epsilon^\mu \|\vec{x}\|^{-\mu}$, whose Fourier transform and its derivative are $\tilde{f}(\vec{k}) = c_0 \|\vec{k}\|^{-(d-\mu)}$, $\tilde{f}'(\vec{k}) = c_1 \|\vec{k}\|^{-(d-\mu+1)}$ and $\|k_0\| = \tilde{f}^{-1}(\frac{\lambda}{\rho}) = (\frac{\lambda}{c_0 \rho})^{-\frac{1}{d-\mu}}$. The constants are given by $c_0 = 2^{d-\mu} \pi^{\frac{d}{2}} \epsilon^\mu \frac{\Gamma(\frac{d-\mu}{2})}{\Gamma(\frac{\mu}{2})} = \epsilon^\mu c_2$, $c_1 = -(d-\mu)c_0$, $c_2 = 2^{d-\mu} \pi^{\frac{d}{2}} \frac{\Gamma(\frac{d-\mu}{2})}{\Gamma(\frac{\mu}{2})}$

$$p(\lambda) = \frac{S_{d-1}}{(2\pi)^d} \frac{\|\vec{k}_0\|^{d-1}}{\rho^2 |\tilde{f}'(\vec{k}_0)|} = \frac{S_{d-1}}{(2\pi)^d} \frac{\|\vec{k}_0\|^{2d-\mu}}{\rho^2 |c_1|}$$

$$= \frac{S_{d-1}}{(2\pi)^d} \frac{c_0^{\frac{d}{d-\mu}}}{\rho^2 (d-\mu)} (\frac{\lambda}{\rho})^{-\frac{2d-\mu}{d-\mu}} = \frac{S_{d-1}}{(2\pi)^d} \frac{c_2^{\frac{d}{d-\mu}}}{d-\mu} \lambda^{-\frac{2d-\mu}{d-\mu}} (\rho \epsilon^d)^{\frac{\mu}{d-\mu}}$$

(S40)

## Derivation of eigenspectrum with exponential kernel function in high-density limit

Here we consider the exponential kernel function $f(\vec{x}) = e^{-b\|\vec{x}\|}$, whose Fourier transform and its derivative are $\tilde{f}(\vec{k}) = \frac{c_1}{(b^2+\|\vec{k}\|^2)^{\frac{d+1}{2}}}$, $\tilde{f}'(\vec{k}) = -\frac{(d+1)\vec{k}c_1}{(b^2+\|\vec{k}\|^2)^{-\frac{d+3}{2}}}$ and $\|k_0\| = \tilde{f}^{-1}(\frac{\lambda}{\rho}) = \sqrt{(\frac{c_1\rho}{\lambda})^{\frac{2}{d+1}} - b^2}$, $\|k_0\|^2 + b^2 = (\frac{c_1\rho}{\lambda})^{\frac{2}{d+1}}$, where $c_1 = 2^d \pi^{\frac{d-1}{2}} b \Gamma(\frac{d+1}{2})$

$$p(\lambda) = \frac{S_{d-1}}{(2\pi)^d} \frac{\|\vec{k}_0\|^{d-1}}{\rho^2 |\tilde{f}'(\vec{k}_0)|} = \frac{S_{d-1}}{(2\pi)^d} \frac{\left(b^2 + \|\vec{k}_0\|^2\right)^{-\frac{d+3}{2}} \|\vec{k}_0\|^{d-1}}{\rho^2 |(d+1)\vec{k}_0 c_1|}$$

$$= \frac{S_{d-1}}{(2\pi)^d} \frac{\left(\frac{c_1\rho}{\lambda}\right)^{-\frac{d+3}{d+1}} \|\vec{k}_0\|^{d-2}}{(d+1)\rho^2 |c_1|} = \frac{S_{d-1}}{(d+1)(2\pi)^d} c_1^{\frac{2}{d+1}} \rho^{\frac{-d+1}{d+1}} \lambda^{-\frac{d+3}{d+1}} \|\vec{k}_0\|^{d-2}$$

$$= \frac{S_{d-1}}{(d+1)(2\pi)^d} 2^{\frac{2d}{d+1}} \pi^{\frac{d-1}{d+1}} \Gamma\left(\frac{d+1}{2}\right)^{\frac{2}{d+1}} (\rho b^{-d})^{\frac{-d+1}{d+1}}$$

(S41)

$$\lambda^{-\frac{d+3}{d+1}} \left(\left(\frac{2^d \pi^{\frac{d-1}{2}} \Gamma\left(\frac{d+1}{2}\right) \rho b^{-d}}{\lambda}\right)^{\frac{2}{d+1}} - 1\right)^{\frac{d-2}{2}}$$

It is straightforward to see that this spectrum is not scale invariant. For example, when $d = 2$, the above expression reduces to a perfect power-law spectrum $p(\lambda) \sim \rho^{\frac{-d+1}{d+1}} \lambda^{-\frac{d+3}{d+1}}$, which changes with scale over sampling.

## Variational approximation

To find a general approximation for the eigenspectrum that goes beyond the high-density limit, we use Gaussian variational approximation in the field representation, namely by looking for the best quadratic action $S_v$,

$$S_v = -\frac{1}{2} \sum_{\alpha\beta}^{n} \int_{-\infty}^{+\infty} d\vec{x} d\vec{x}' G_{\alpha\beta}^{-1}(\vec{x} - \vec{x}') \hat{\psi}^\alpha(\vec{x}) \hat{\psi}^\beta(\vec{x}'),$$

(S42)

to approximate the action $S_1$ in the partition function (*Equations S29–S31*). This enables us to represent the partition function by a Gaussian integral, which can be evaluated analytically. We find the best quadratic action $S_v$ by minimizing the difference between $S_1$ and $S_v$, which is defined as KL divergence between two distributions that are proportional to $e^{S_1}$ and $e^{S_v}$.

In this section, we will proceed by using the grand canonical ensemble formulation, namely the average in *Equation S4*, instead of using a fixed covariance matrix size $N$, which is now carried out

across all different sizes. If $N$ follows a Poisson distribution, it is easy to show (Appendix 2) that the grand canonical partition function is given by **Equation S116**:

$$\mathcal{Z} = \sum_N \langle \Xi_N^n(z) \rangle \frac{a^N}{N!},$$

where $a = \langle N \rangle$. As a result, the new action $S_1$ becomes

$$S_1 = NA - \frac{1}{2} \sum_{\alpha=1}^{n} \int_{-\infty}^{+\infty} d\vec{x} d\vec{x}' f^{-1}(\vec{x} - \vec{x}') \hat{\psi}^\alpha(\vec{x}) \hat{\psi}^\alpha(\vec{x}'). \tag{S43}$$

Here and below, $N$ should be viewed as the average matrix size. The resolvent $g(z)$ in **Equation S12** can be similarly generalized to **Equation S117**,

$$g(z) = \lim_{n \to 0} -\frac{2}{Nn} \partial_z \ln \mathcal{Z}$$

As in statistical physics, we define the free energy as

$$F_1 = -\ln \mathcal{Z} = -\ln \int_{-\infty}^{+\infty} D[\hat{\psi}^\alpha] e^{S_1} \tag{S44}$$

We shall define the variational free energy $F_v$ such that it would approximate the true free energy $F_1$ by minimizing $D_{KL}(P_v \| P_1)$,

$$F_v = D_{KL}(P_v \| P_1) + F_1 \tag{S45}$$

where

$$P_1 = \frac{e^{S_1}}{\int_{-\infty}^{+\infty} D[\hat{\psi}^\alpha] e^{S_1}} \tag{S46}$$

$$P_v = \frac{e^{S_v}}{\int_{-\infty}^{+\infty} D[\hat{\psi}^\alpha] e^{S_v}} \tag{S47}$$

The KL divergence $D_{KL}(P_v \| P_1)$ is always nonnegative and the free energy $F_1$ is independent of the quadratic action $S_v$. Therefore, we need to minimize the variational free energy $F_v$. Let us now examine the variational free energy $F_v$

$$
\begin{aligned}
F_v &= D_{KL}(P_v \| P_1) + F_1 \\
&= \frac{1}{Z_v} \int_{-\infty}^{+\infty} D[\hat{\psi}^\alpha] e^{S_v} \ln \frac{P_v}{P_1} - \ln \mathcal{Z} \\
&= \frac{1}{Z_v} \int_{-\infty}^{+\infty} D[\hat{\psi}^\alpha] e^{S_v} (S_v - S_1 - \ln \int_{-\infty}^{+\infty} D[\hat{\psi}^\alpha] e^{S_v} + \ln \int_{-\infty}^{+\infty} D[\hat{\psi}^\alpha] e^{S_1}) - \ln \mathcal{Z} \\
&= \frac{1}{Z_v} \int_{-\infty}^{+\infty} D[\hat{\psi}^\alpha] e^{S_v} (S_v - S_1) - \ln Z_v
\end{aligned}
\tag{S48}
$$

Here $Z_v$ is the normalization factor

$$Z_v = \int_{-\infty}^{+\infty} D[\hat{\psi}^\alpha] e^{S_v} \tag{S49}$$

Since we want to minimize $F_v$, the constant term

$$\frac{1}{Z_v} \int_{-\infty}^{+\infty} D[\hat{\psi}^\alpha] e^{S_v} S_v = \text{const} \tag{S50}$$

can beignored, and *Equation S48* is reduced to

$$F_v = -\frac{1}{Z_v} \int_{-\infty}^{+\infty} D[\hat{\psi}^\alpha] e^{S_v} S_1 - \ln Z_v \tag{S51}$$

To simplify the formula, let us introduce $S_2$

$$S_2 = -\frac{1}{2} \sum_{\alpha=1}^{n} \int_{-\infty}^{+\infty} d\vec{x} d\vec{x}' f^{-1}(\vec{x} - \vec{x}') \hat{\psi}^\alpha(\vec{x}) \hat{\psi}^\alpha(\vec{x}') \tag{S52}$$

and rewrite *Equation S51* as

$$F_v = -\frac{1}{Z_v} \int_{-\infty}^{+\infty} D[\hat{\psi}^\alpha] e^{S_v} S_2 - \frac{1}{Z_v} \int_{-\infty}^{+\infty} D[\hat{\psi}^\alpha] e^{S_v} NA - \ln Z_v \tag{S53}$$

Next, we will compute each term in the variational free energy $F_v$ First, we calculate the third term $\ln Z_v$ in *Equation S53* by *Equations S42 and S49*

$$\ln Z_v = \ln \left( \prod_{\alpha,\beta} (2\pi)^{N/2} (\det(G_{\alpha\beta}^{-1}))^{-\frac{1}{2}} \right)$$
$$= \sum_{\alpha,\beta} \frac{1}{2} \ln \det(G_{\alpha\beta}) + \frac{n^2 N}{2} \ln(2\pi) \tag{S54}$$

Second, we calculate the first term $\frac{1}{Z_v} \int_{-\infty}^{+\infty} D[\hat{\psi}^\alpha] e^{S_v} S_2$ in *Equation S53*

$$\frac{1}{Z_v} \int_{-\infty}^{\infty} D[\hat{\psi}^\alpha] e^{S_v} S_2 = \frac{1}{Z_v} \lim_{h \to 0} \frac{\partial}{\partial h} \int_{-\infty}^{\infty} D[\hat{\psi}^\alpha] e^{S_v + hS_2}$$
$$= \frac{1}{Z_v} \lim_{h \to 0} \frac{\partial}{\partial h} \prod_{\alpha=\beta} \left[ \det(G_{\alpha\beta}^{-1} + hf^{-1}) \right]^{-\frac{1}{2}} \prod_{\alpha \neq \beta} \left[ \det(G_{\alpha\beta}^{-1}) \right]^{-\frac{1}{2}}$$
$$= \lim_{h \to 0} \frac{\partial}{\partial h} \prod_{\alpha} \left[ \det(I + hf^{-1} G_{\alpha\alpha}) \right]^{-\frac{1}{2}} \tag{S55}$$
$$= \sum_{\alpha}^{n} \frac{\partial}{\partial h} \lim_{h \to 0} \left( 1 - \frac{h}{2} \mathrm{Tr}(f^{-1} G_{\alpha\alpha}) \right)$$
$$= -\sum_{\alpha}^{n} \frac{1}{2} \mathrm{Tr}(f^{-1} G_{\alpha\alpha})$$

Third, we calculate the second term $\frac{1}{Z_v} \int_{-\infty}^{+\infty} D[\hat{\psi}^\alpha] e^{S_v} NA$ in *Equation S53*, recall the term $A$ (*Equation S31*)

$$A = \int_{-L}^{L} \frac{d^d \vec{x}}{V} (z)^{-\frac{n}{2}} \int d\sigma p(\sigma) \exp \left[ \frac{1}{2z} \sum_{\alpha=1}^{n} \hat{\psi}^\alpha(\vec{x})^2 \sigma^2 \right]$$

$$\frac{1}{Z_v} \int_{-\infty}^{+\infty} D[\hat{\psi}^\alpha] e^{S_v} NA$$

$$= \frac{N(z)^{-\frac{n}{2}}}{Z_v} \int d\sigma p(\sigma) \int_{-\infty}^{+\infty} D[\hat{\psi}^\alpha] e^{S_v} \int_{-L}^{L} \frac{d^d \vec{x}}{V} \exp\left[\frac{1}{2z} \sum_{\alpha=1}^{n} \hat{\psi}^\alpha(\vec{x})^2 \sigma^2\right]$$

$$= \frac{N(z)^{-\frac{n}{2}}}{Z_v} \int d\sigma p(\sigma) \int_{-L}^{L} \frac{d^d \vec{x}_0}{V} \int_{-\infty}^{+\infty} D[\hat{\psi}^\alpha] \exp\left[S_v + \frac{1}{2z} \sum_{\alpha=1}^{n} \hat{\psi}^\alpha(\vec{x})^2 \sigma^2\right] \qquad \text{(S56)}$$

$$= \frac{N(z)^{-\frac{n}{2}}}{Z_v} \int d\sigma p(\sigma) \int_{-L}^{L} \frac{d^d \vec{x}_0}{V} \prod_{\alpha,\beta} \left[\det(K_{\alpha\beta})\right]^{\frac{1}{2}}$$

$$= N(z)^{-\frac{n}{2}} \int d\sigma p(\sigma) \int_{-L}^{L} \frac{d^d \vec{x}_0}{V} \prod_{\alpha,\beta} \left[\det(K_{\alpha\beta} G_{\alpha\beta}^{-1})\right]^{\frac{1}{2}}$$

where

$$S_v + \frac{1}{2z} \sum_{\alpha=1}^{n} \hat{\psi}^\alpha(\vec{x})^2 \sigma^2 = -\frac{1}{2} \sum_{\alpha\beta}^{n} \int_{-\infty}^{+\infty} d\vec{x} d\vec{x}' G_{\alpha\beta}^{-1}(\vec{x}-\vec{x}') \hat{\psi}^\alpha(\vec{x}) \hat{\psi}^\beta(\vec{x}') + \frac{1}{2z} \sum_{\alpha=1}^{n} \hat{\psi}^\alpha(\vec{x})^2 \sigma^2$$

$$= -\frac{1}{2} \sum_{\alpha\beta}^{n} \int_{-\infty}^{+\infty} d\vec{x} d\vec{x}' K_{\alpha\beta}^{-1}(\vec{x}-\vec{x}') \hat{\psi}^\alpha(\vec{x}) \hat{\psi}^\beta(\vec{x}') \qquad \text{(S57)}$$

$$K_{\alpha\beta}^{-1}(\vec{x}, \vec{y}) = G_{\alpha\beta}^{-1}(\vec{x}, \vec{y}) - \frac{\sigma^2}{z} \delta_{\alpha\beta} \delta(\vec{x}-\vec{x}_o) \delta(\vec{y}-\vec{x}_0)$$

$$\det(K_{\alpha\beta}^{-1} G_{\alpha\beta}) = 1 - \frac{\sigma^2}{z} \delta_{\alpha\beta} G(\vec{x}_0, \vec{x}_0) \qquad \text{(S58)}$$

$$\frac{1}{Z_v} \int_{-\infty}^{+\infty} D[\hat{\psi}^\alpha] e^{S_v} NA = N(z)^{-\frac{n}{2}} \int d\sigma p(\sigma) \int_{-L}^{L} \frac{d^d \vec{x}_0}{V} \prod_{\alpha,\beta} \left[\det(K_{\alpha\beta}^{-1} G_{\alpha\beta})\right]^{-\frac{1}{2}}$$

$$= N(z)^{-\frac{n}{2}} \int d\sigma p(\sigma) \int_{-L}^{L} \frac{d^d \vec{x}_0}{V} \prod_{\alpha} \left[\det(K_{\alpha\alpha}^{-1} G_{\alpha\alpha})\right]^{-\frac{1}{2}}$$

$$= N(z)^{-\frac{n}{2}} \int d\sigma p(\sigma) \int_{-L}^{L} \frac{d^d \vec{x}_0}{V} \prod_{\alpha} (1 - \frac{\sigma^2}{z} G_{\alpha\alpha}(\vec{x}_0, \vec{x}_0))^{-\frac{1}{2}} \qquad \text{(S59)}$$

$$= N(z)^{-\frac{n}{2}} \int d\sigma p(\sigma) \prod_{\alpha} (1 - \frac{\sigma^2}{z} G_{\alpha\alpha}(0))^{-\frac{1}{2}}$$

$$= N(z)^{-\frac{n}{2}} \int d\sigma p(\sigma) \exp(-\frac{1}{2} \mathrm{Tr}_n \ln(1 - \frac{\sigma^2}{z} \int \frac{d^d \vec{k}}{(2\pi)^d} \tilde{G}(\vec{k})))$$

In sum, the variational free energy $F_v$ is equal to

$$F_v = \sum_{\alpha} \frac{1}{2} \mathrm{Tr}(f^{-1} G_{\alpha\alpha}) - \sum_{\alpha,\beta} \frac{1}{2} \ln(\det(G_{\alpha\beta}))$$

$$- N(z)^{-\frac{n}{2}} \int d\sigma p(\sigma) \exp(-\frac{1}{2} \mathrm{Tr}_n \ln(1 - \frac{\sigma^2}{z} \int \frac{d^d \vec{k}}{(2\pi)^d} \tilde{G}(\vec{k})))$$

$$= \sum_{\alpha} \frac{V}{2} \int \frac{d^d \vec{k}}{(2\pi)^d} \frac{\tilde{G}(\vec{k})}{\tilde{f}(\vec{k})} - \frac{V}{2} \int \frac{d^d \vec{k}}{(2\pi)^d} \sum_{\alpha,\beta} \ln(\tilde{G}_{\alpha\beta}(\vec{k}))$$

$$- N(z)^{-\frac{n}{2}} \int d\sigma p(\sigma) \exp(-\frac{1}{2} \mathrm{Tr}_n \ln(1 - \frac{\sigma^2}{z} \int \frac{d^d \vec{k}}{(2\pi)^d} \tilde{G}(\vec{k}))) \qquad \text{(S60)}$$

Now let us find the best quadratic action $S_v$ that minimizes the variational free energy $F_v$

$$\frac{\delta F_v}{\delta \tilde{G}_{\alpha\beta}} = 0 \tag{S61}$$

The solution of *Equation S61* is given by

$$\tilde{G}^{-1}_{\alpha\beta}(\vec{k}) = \delta_{\alpha\beta}\tilde{G}^{-1}(\vec{k}) \tag{S62}$$

$$\frac{1}{\tilde{f}(\vec{k})} - \int \mathrm{d}\sigma p(\sigma)\frac{\rho\sigma^2}{z - \sigma^2 \int \mathrm{D}\vec{k}\,\tilde{G}(\vec{k})} - \frac{1}{\tilde{G}(\vec{k})} = 0 \tag{S63}$$

where $\int \mathrm{D}\vec{k} \equiv \int \frac{\mathrm{d}^d \vec{k}}{(2\pi)^d}$. By using *Equation S117*, we finally obtain

$$g(z) = \lim_{n\to 0}\frac{2}{nN}\frac{\partial}{\partial z}F_1 \approx \lim_{n\to 0}\frac{2}{nN}\frac{\partial}{\partial z}F_v = \int \mathrm{d}\sigma p(\sigma)\frac{1}{z - \sigma^2 \int \mathrm{D}\vec{k}\,\tilde{G}(\vec{k})} \tag{S64}$$

## Scale invariance of the covariance spectrum in the Gaussian variational Model

In Result, we point to two factors that contribute to the scale invariance of eigenspectrum using the high-density theory. In this section, we show that the same conclusion can be drawn by using the Gaussian variational method. Furthermore, we examine how the heterogeneity of neural activity influences the eigendensity calculated by the Gaussian variational model. We show that $\frac{\partial p(\lambda)}{\partial \rho}$, which characterizes the change of eigendensity due to sampling in the functional space, decreases with the heterogeneity of neural activity described by higher-order moment of neural activity variance, for example, $\mathrm{E}(\sigma^4)$.

Let us rewrite *Equation S63* as

$$\mathcal{G} = \int \mathrm{D}\vec{k}\,\tilde{G}(\vec{k}) = \int \mathrm{D}\vec{k}\,\frac{\tilde{f}(\vec{k})}{1 - M(z)\tilde{f}(\vec{k})}$$
$$M(z) = \int \mathrm{d}\sigma p(\sigma)\frac{\rho\sigma^2}{z - \sigma^2\mathcal{G}(z)} \tag{S65}$$

To present a formal expression for the eigendensity, let us define $\mathbf{Re}(\mathcal{G}) \equiv g_r$, $\mathbf{Im}(\mathcal{G}) \equiv g_i$. From *Equations S6 and S64*, we find

$$p(\lambda, \rho) = \frac{1}{\pi}\left\langle \frac{\sigma^2 g_i}{(\lambda - \sigma^2 g_r)^2 + \sigma^4 g_i^2} \right\rangle_\sigma, \tag{S66}$$

where $\langle ... \rangle_\sigma = \int ...p(\sigma)\mathrm{d}\sigma$.

A direct computation of *Equation S66*, however, remains difficult: the complication arises from the complex function $M(z)$ in *Equation S65*, which in turn is an integral function of $\mathcal{G}$. To streamline the calculation, let us further define $\mathbf{Re}(M) \equiv \rho a$, $\mathbf{Im}(M) \equiv \rho b$. Writing it down explicitly, we have

$$a = \left\langle \frac{\sigma^2(\lambda - \sigma^2 g_r)}{(\lambda - \sigma^2 g_r)^2 + \sigma^4 g_i^2} \right\rangle_\sigma \tag{S67}$$

$$b = \left\langle \frac{\sigma^4 g_i}{(\lambda - \sigma^2 g_r)^2 + \sigma^4 g_i^2} \right\rangle_\sigma \tag{S68}$$

The real and imaginary part of $\mathcal{G}$ can now be expressed as functions of $a$ and $b$. Integrating *Equation S65* in the spherical coordinates, we have

$$g_r(\rho) = \frac{S_{d-1}}{(2\pi)^d} \int_{\pi/L}^{\pi/\epsilon} \mathrm{d}k k^{d-1} \frac{\tilde{f}(k)[1 - \rho a \tilde{f}(k)]}{[1 - \rho a \tilde{f}(k)]^2 + \rho^2 b^2 \tilde{f}^2(k)}$$

$$g_i(\rho) = \frac{S_{d-1}}{(2\pi)^d} \int_{\pi/L}^{\pi/\epsilon} \mathrm{d}k k^{d-1} \frac{\rho b \tilde{f}^2(k)}{[1 - \rho a \tilde{f}(k)]^2 + \rho^2 b^2 \tilde{f}^2(k)}$$

(S69)

where for clarity, we have abused the notation a bit by defining $k = \|\vec{k}\|$; $S_{d-1}$ is the surface area of unit $d$-ball in the momentum space. In order to evaluate the integrals analytically, we introduce an ultraviolet cutoff $\pi/\epsilon$. Numerically, whether integrating up to $\pi/\epsilon$ or greater than this bound shows little difference.

## Numerical solution of the Gaussian variational method

With **Equations S66–S69**, we numerically calculate the eigendensity iteratively from the following steps:

- Step 1: set the initial values of $a$ and $b$ as $a_0 = 1$, $b_0 = 1$
- Step 2: solve for $a$ in **Equation S67** with fixed $b$
- Step 3: solve for $b$ in **Equation S68** with fixed $a$
- Step 4: iterate Steps 2 and 3 10 times
- Step 5: calculate $p(\lambda)$ using **Equation S66**

Note that we plug **Equation S69** into **Equations S67 and S68** in step 2–3.

## Two contributing factors on the scale invariance

We next derive an analytical expression for **Equation S69** by considering the approximate power law kernel function $f(\vec{x}) \approx \epsilon^\mu \|\vec{x}\|^{-\mu}$, $\mu > 0$, from which the high-density theory results on the scale invariance can be extended.

By a change of variable $x = \tilde{f}(k) \sim \epsilon^\mu k^{-(d-\mu)}$, and let $x_\epsilon \equiv \tilde{f}(\frac{\pi}{\epsilon})$, $x_L \equiv \tilde{f}(\frac{\pi}{L})$, we have

$$g_i(\rho) \sim \frac{\epsilon^{\frac{\mu d}{d-\mu}}}{d-\mu} \int_{x_\epsilon}^{x_L} \mathrm{d}x \frac{\rho b x^{-\frac{\mu}{d-\mu}}}{[1 - \rho a x]^2 + \rho^2 b^2 x^2},$$

(S70)

where $\sim$ indicates that all constant numerical factors (e.g., $\pi$ and $\Gamma(d/2)$) are ignored. To compute **Equation S70**, we perform a branch cut at $[0, \infty]$, and perform a contour integral on the complex plane following the path in , **Appendix 2—figure 1A**. When $0 < \beta = 1 - \frac{\mu}{d-\mu} < 2$, the integral on the large circle $\Gamma_R$ and the small circle $\Gamma_\epsilon$ goes to zero as $x_L \to \infty, x_\epsilon \to 0$, leaving only two simple poles (zeros of the function in the denominator) in the complex plane. By applying the residue theorem, we find an expression for $g_i$ in the limit $L \to \infty, \epsilon \to 0$

$$\cos\theta = -\frac{a}{\sqrt{a^2 + b^2}}$$

$$\beta = \frac{d - 2\mu}{d - \mu}$$

(S71)

$$g_i \sim -\frac{\epsilon^{\frac{\mu d}{d-\mu}}}{d-\mu} \frac{\sin(\beta - 1)\theta}{\sin\theta \sin\pi\beta} \frac{\pi b \rho^{1-\beta}}{(a^2 + b^2)^{\beta/2}}$$

The analytical expression for $g_r$ is a bit more involving.

$$g_r(\rho) \sim \frac{\epsilon^{\frac{\mu d}{d-\mu}}}{d-\mu} \int_{x_\epsilon}^{x_L} \mathrm{d}x \frac{x^{-\frac{d}{d-\mu}}}{[1 - \rho a x]^2 + \rho^2 b^2 x^2} - \frac{a}{b} g_i$$

(S72)

It has two terms, the second term is similar to **Equation S70**; the first term, however, diverges as $x_\epsilon \to 0$. Thus, the radius of the small circle $\Gamma_\epsilon$ in **Appendix 2—figure 1** cannot shrink to zero: this is precisely the requirement of an ultraviolet cutoff in the wave vector $\vec{k}$. The contour integral on the large circle $\Gamma_R$, on the other hand, goes to zero as $x_L \to \infty$. Thus, the integral on $\Gamma_\epsilon$ contributes to the final result. By considering leading order term of $x_\epsilon$ for finite but small $x_\epsilon$, we find

$$\cos\theta = -\frac{a}{\sqrt{a^2 + b^2}}$$

$$\gamma = \frac{-\mu}{d - \mu}$$

(S73)

$$g_r \sim -\frac{\epsilon^{\frac{\mu d}{d - \mu}}}{d - \mu} \frac{x_\epsilon^\gamma}{\gamma} - \frac{\epsilon^{\frac{\mu d}{d - \mu}}}{d - \mu} \frac{\sin(\gamma - 1)\theta}{\sin\theta \sin\pi\gamma} \frac{\pi\rho^{-\gamma}}{(a^2 + b^2)^{\gamma/2}} - \frac{a}{b} g_i$$

Recall $x_\epsilon \sim \frac{\epsilon^\mu}{(\pi/\epsilon)^{d-\mu}}$, and we find that the first term in $g_r$ is proportional to $\pi^\mu/\mu$, independent of $\epsilon$.

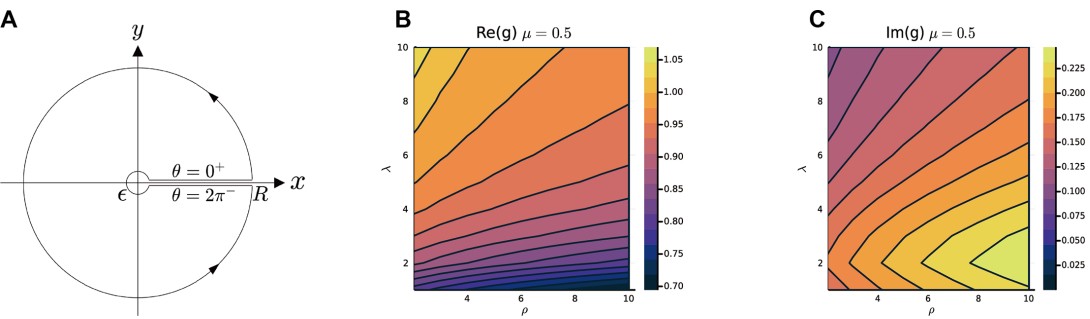

**Appendix 2—figure 1.** Calculate $g_i$ and $g_r$. (**A**) The path of the contour integral for $g_i$, $g_r$ (**Equation S70**). The heatmap of $g_r$ and $g_i$ with respect to $\lambda$ and $\rho$, $g_r$ in (**B, C**) are calculated by the numerical method (Methods). The parameters are $N = 1024$, $\rho = 10.24$, $d = 2$, $L = 10$, $\mu = 0.5$, $\epsilon = 0.03125$ is i.i.d. sampled from a log-normal distribution with zero mean and a standard deviation of 0.5 in the natural logarithm of the $\sigma_i^2$ values; we also normalize $E(\sigma_i^2) = 1$.

According to **Equation S71 and S73**, one can immediately see that as $\mu/d \to 0$, the $\rho$-dependence relationship vanishes for $g_r$ and $g_i$. We therefore conclude that a slower power-law decay in the kernel function and/or a higher dimension of the functional space are two contributing factors for the scale invariance of the covariance spectrum.

## Heterogeneity of neural activity across neurons enhances scale invariance

Next, we take a more close look at how the eigendensity changes with $\rho$ for finite but small $\mu/d$ and when $\lambda \gg 1$. Using **Equation S66**, we have

$$\frac{\partial p}{\partial \rho} = \frac{1}{\pi} \left\langle \frac{\partial g_i}{\partial \rho} \frac{\sigma^2 \left[(\lambda - \sigma^2 g_r)^2 + \sigma^4 g_i^2\right] - 2\sigma^6 g_i^2}{\left[(\lambda - \sigma^2 g_r)^2 + \sigma^4 g_i^2\right]^2} + \frac{\partial g_r}{\partial \rho} \frac{2\sigma^4 g_i(\lambda - \sigma^2 g_r)}{\left[(\lambda - \sigma^2 g_r)^2 + \sigma^4 g_i^2\right]^2} \right\rangle_\sigma$$

(S74)

From numerical calculation, we find that typically $g_r \gg g_i$, so one can use the approximation

$$\frac{\partial p}{\partial \rho} \approx \frac{1}{\pi} \left\langle \frac{\partial g_i}{\partial \rho} \frac{\sigma^2}{(\lambda - \sigma^2 g_r)^2 + \sigma^4 g_i^2} \right\rangle_\sigma + \frac{1}{\pi} \left\langle \frac{\partial g_r}{\partial \rho} \frac{2\sigma^4 g_i}{(\lambda - \sigma^2 g_r)^3} \right\rangle_\sigma$$

(S75)

Recall **Equation S66**

$$p(\lambda, \rho) = \frac{1}{\pi} \left\langle \frac{\sigma^2 g_i}{(\lambda - \sigma^2 g_r)^2 + \sigma^4 g_i^2} \right\rangle_\sigma,$$

(S76)

Since $p(\lambda, \rho)$ is very small for large $\lambda$, a more appropriate measure is to examine

$$\frac{\partial \log p}{\partial \rho} \equiv \frac{1}{p} \frac{\partial p}{\partial \rho} \approx \frac{\partial g_i}{\partial \rho} \frac{1}{g_i} + 2\frac{\partial g_r}{\partial \rho} \frac{\left\langle \frac{\sigma^4}{(\lambda - \sigma^2 g_r)^3} \right\rangle_\sigma}{\left\langle \frac{\sigma^2}{(\lambda - \sigma^2 g_r)^2} \right\rangle_\sigma}$$

(S77)

Considering the large eigenvalue case $\lambda \gg \sigma^2 g_r$ (the numerical value of $g_r$ is on the order of 1), we perform Taylor expansion and arrive at

$$\left\langle \frac{\sigma^2}{(\lambda - \sigma^2 g_r)^2} \right\rangle_\sigma \approx \left\langle \frac{\sigma^2}{\lambda^2} + \frac{2\sigma^4 g_r}{\lambda^3} + \frac{3\sigma^6 g_r^2}{\lambda^4} \right\rangle_\sigma \tag{S78}$$

$$\left\langle \frac{\sigma^4}{(\lambda - \sigma^2 g_r)^3} \right\rangle_\sigma \approx \left\langle \frac{\sigma^4}{\lambda^3} + \frac{3\sigma^6 g_r}{\lambda^4} \right\rangle_\sigma \tag{S79}$$

Note $\langle \sigma^2 \rangle_\sigma \equiv \mathrm{E}(\sigma^2)$ is normalized to 1.

$$\frac{\partial \log p}{\partial \rho} \approx \frac{\partial g_i}{\partial \rho} \frac{1}{g_i} + 2\frac{\partial g_r}{\partial \rho} \frac{\left\langle \dfrac{\sigma^4}{(\lambda - \sigma^2 g_r)^3} \right\rangle_\sigma}{\left\langle \dfrac{\sigma^2}{(\lambda - \sigma^2 g_r)^2} \right\rangle_\sigma}$$

$$\approx \frac{\partial g_i}{\partial \rho} \frac{1}{g_i} + 2\frac{\partial g_r}{\partial \rho} \frac{\left\langle \sigma^4 \right\rangle_\sigma + \dfrac{3g_r}{\lambda} \left\langle \sigma^6 \right\rangle_\sigma}{\lambda + 2g_r \left\langle \sigma^4 \right\rangle_\sigma + \dfrac{3g_r^2}{\lambda} \left\langle \sigma^6 \right\rangle_\sigma} \tag{S80}$$

By examining **Equations S71 and S73**, we find that when $\lambda \gg g_r$, $a \gg b$, $\theta \approx \pi$, $g_r$ decays weakly with $\rho$ while $g_i$ increases weakly with $\rho$ (also confirmed by numerical calculation, **Figure 1B, C**)

$$\frac{\partial g_r}{\partial \rho} < 0, \ \frac{\partial g_i}{\partial \rho} > 0.$$

It is therefore straightforward to see from 80 that the higher-order moment (e.g., $\mathrm{E}(\sigma^4)$) in the activity variance contributes to reducing the $\rho$-dependence in the eigendensity function.

## The relationship between CI and eigendensity

In this section, we show how the CI introduced in Methods is related to **Equation S80**, namely how the eigendensity changes with the neuronal density in the functional space. Recall the definition of CI in **Equation S13**:

$$\mathrm{CI} := \frac{1}{\log(q_0/q_1)} \int_{\log q_1}^{\log q_0} \left| \frac{\partial \log \lambda(q)}{\partial \log \rho} \right| \mathrm{d} \log q$$

where

$$q(\lambda) = \int_\lambda^\infty p(\lambda)\mathrm{d}\lambda$$

we used implicit differentiation to compute $\frac{\partial \log \lambda(q)}{\partial \log \rho}$. For clarity, we write the function $q(\lambda, \rho)$ explicitly involving $\lambda$ and $\rho$ as $Q(\lambda, \rho)$ in **Equations S81–S83**.

$$F(\lambda(q, \rho), q, \rho) = Q(\lambda(q, \rho), \rho) - q \equiv 0 \tag{S81}$$

$$\frac{\mathrm{d}F(\lambda(q, \rho), q, \rho)}{\mathrm{d}\rho} = \frac{\partial F(\lambda(q, \rho), q, \rho)}{\partial \rho} + \frac{\partial F(\lambda(q, \rho), q, \rho)}{\partial \lambda} \frac{\partial \lambda(q, \rho)}{\partial \rho} = 0 \tag{S82}$$

$$\frac{\partial \lambda(q, \rho)}{\partial \rho} = -\frac{\dfrac{\partial F(\lambda(q, \rho), q, \rho)}{\partial \rho}}{\dfrac{\partial F(\lambda(q, \rho), q, \rho)}{\partial \lambda}} = -\frac{\dfrac{\partial Q(\lambda(q, \rho), \rho)}{\partial \rho}}{\dfrac{\partial Q(\lambda(q, \rho), \rho)}{\partial \lambda}} \tag{S83}$$

Now we can write CI as

$$\frac{\partial \log \lambda(q,\rho)}{\partial \log \rho} = \frac{\rho}{\lambda(q,\rho)}\frac{\partial \lambda(q,\rho)}{\partial \rho} = -\frac{\rho}{\lambda(q,\rho)}\frac{\dfrac{\partial q(\rho,\lambda)}{\partial \rho}}{\dfrac{\partial q(\rho,\lambda)}{\partial \lambda}} \tag{S84}$$

from which we arrive at *Equation S15* in Methods:

$$\mathrm{CI} = \frac{1}{\log(q_0/q_1)}\int_{\log q_1}^{\log q_0}\left|\frac{\partial \log \lambda(q,\rho)}{\partial \log \rho}\right|\mathrm{d}\log q = \frac{1}{\log(q_0/q_1)}\int_{q_1}^{q_0}\left|-\frac{\rho}{q\lambda}\frac{\dfrac{\partial q}{\partial \rho}}{\dfrac{\partial q}{\partial \lambda}}\right|\mathrm{d}q$$

$$= \frac{1}{\log(q_0/q_1)}\int_{\lambda(q_1)}^{\lambda(q_0)}\left|-\frac{\rho}{q\lambda}\frac{\dfrac{\partial q}{\partial \rho}}{\dfrac{\partial q}{\partial \lambda}}\right|\frac{\partial q}{\partial \lambda}\mathrm{d}\lambda = \frac{1}{\log(q_0/q_1)}\int_{\lambda(q_0)}^{\lambda(q_1)}\left|\frac{\rho}{q\lambda}\frac{\partial q}{\partial \rho}\right|\mathrm{d}\lambda \tag{S85}$$

Finally, we can rewrite CI as a function of $\frac{\partial p}{\partial \rho}$ using a double integral:

$$\mathrm{CI} = \frac{1}{\log(q_0/q_1)}\int_{\lambda(q_0)}^{\lambda(q_1)}\left|\frac{\rho}{q\lambda}\frac{\partial q}{\partial \rho}\right|\mathrm{d}\lambda = \frac{1}{\log(q_0/q_1)}\int_{\lambda(q_0)}^{\lambda(q_1)}\mathrm{d}\lambda_1\left|\frac{\rho}{q\lambda_1}\int_{\lambda_1}^{\infty}\mathrm{d}\lambda_2\frac{\partial p(\lambda_2)}{\partial \rho}\right|$$

$$= \frac{1}{\log(q_0/q_1)}\int_{\lambda(q_0)}^{\lambda(q_1)}\frac{1}{\lambda_1}\mathrm{d}\lambda_1\left|\frac{\int_{\lambda_1}^{\infty}\mathrm{d}\lambda_2 p(\lambda_2)\dfrac{\partial \ln p(\lambda_2)}{\partial \ln \rho}}{\int_{\lambda_1}^{\infty}\mathrm{d}\lambda_2 p(\lambda_2)}\right| \tag{S86}$$

## Compare high-density theory and Gaussian variational method

This section aims to determine the conditions under which the high-density approximation aligns with the simulation results. To this end, we begin by comparing the kernel operator $\tilde{G}_h(\vec{k})$ in the high-density quadratic action and $\tilde{G}_v(\vec{k})$ in the variational approximation. We identify the condition when high-density theory would agree with the variational method as well as the numerical simulation, namely $z \gg \int \frac{\mathrm{d}^d\vec{k}}{(2\pi)^d}\tilde{G}_v(\vec{k})$. Secondly, we give a precise re-derivation of the high-density result by incorporating this condition into the variational approximation. Finally, we substitute $\int \frac{\mathrm{d}^d\vec{k}}{(2\pi)^d}\tilde{G}_v(\vec{k})$ with $\int \frac{\mathrm{d}^d\vec{k}}{(2\pi)^d}\tilde{G}_h(\vec{k})$ and estimate the parameter regime where the high-density theory would agree with numerical simulation. This analysis yields a deeper understanding of the relationship between high-density theory and variational method, and how they relate to simulation results.

### A simple comparison of the two methods

For the sake of simplicity, we consider the correlation matrix with $p(\sigma) = \delta(\sigma - 1)$ in this section. Returning to the explicit result *Equation S29–S31* ,

$$\langle \Xi^n(z)\rangle = (\det f)^{-n/2}(z)^{-\frac{Nn}{2}}\int_{-\infty}^{+\infty}D[\hat{\psi}^{\alpha}]e^{S_1} \tag{S87}$$

In the high-density approximation (*Equation S35*)

$$S_h = \int_{-L}^{L}\frac{\mathrm{d}^d\vec{x}}{V}\frac{N}{2z}\sum_{\alpha=1}^{n}\hat{\psi}^{\alpha}(\vec{x})^2 - \frac{1}{2}\sum_{\alpha=1}^{n}\int_{-\infty}^{+\infty}\mathrm{d}\vec{x}\mathrm{d}\vec{x}'f^{-1}(\vec{x}-\vec{x}')\hat{\psi}^{\alpha}(\vec{x})\hat{\psi}^{\alpha}(\vec{x}')$$

$$= -\frac{1}{2}\sum_{\alpha=1}^{n}\int_{-\infty}^{+\infty}\mathrm{d}\vec{x}\mathrm{d}\vec{x}'G_h^{-1}(\vec{x}-\vec{x}')\hat{\psi}^{\alpha}(\vec{x})\hat{\psi}^{\alpha}(\vec{x}') \tag{S88}$$

Here, we introduce $G_h$ as the kernel operator in the high-density quadratic action.

$$G_h^{-1}(\vec{x}-\vec{y}) = f^{-1}(\vec{x}-\vec{y}) - \frac{N}{Vz}\delta(\vec{x}-\vec{y}) \tag{S89}$$

Fourier transform of $G_h$ leads to

$$\tilde{G}_h(\vec{k}) = \frac{\tilde{f}(\vec{k})}{1 - \frac{\rho}{z}\tilde{f}(\vec{k})} \tag{S90}$$

In the variational method *Equation S63*, we have

$$\tilde{G}_v(\vec{k}) = \frac{\tilde{f}(\vec{k})}{1 - C\tilde{f}(\vec{k})} \quad , \quad C = \frac{\rho}{z - \int \frac{d^d\vec{k}}{(2\pi)^d}\tilde{G}_v(\vec{k})}, \tag{S91}$$

where we introduce $G_v$ as the kernel operator in the variational quadratic action. Clearly, the condition that $\tilde{G}_v(\vec{k})$ approaches $\tilde{G}_h(\vec{k})$ is given by

$$C \to \frac{\rho}{z} \quad , \quad z \gg \int \frac{d^d\vec{k}}{(2\pi)^d}\tilde{G}_v(\vec{k})$$

The function $ratio_v(z)$ is defined as:

$$ratio_v(z) = \frac{1}{z}\int \frac{d^d\vec{k}}{(2\pi)^d}\tilde{G}_v(\vec{k})$$

As $ratio_v(z)$ approaches 0, $\tilde{G}_v(\vec{k})$ becomes identical to $\tilde{G}_h(\vec{k})$. Note that $\tilde{G}_v(\vec{k})$ is difficult to compute; instead, we can compute and analyze $\int \frac{d^d\vec{k}}{(2\pi)^d}\tilde{G}_h(\vec{k})$ (see Appendix 2)

$$ratio_h(z) = \frac{1}{z}\int \frac{d^d\vec{k}}{(2\pi)^d}\tilde{G}_h(\vec{k}) \tag{S92}$$

## A re-derivation of the high-density result using the grand canonical ensemble

In this section, we re-derive the high-density result from the grand canonical ensemble and the variational method. The derivation also allows us to reproduce the approximation condition discussed in the previous section.

Let us recall the calculation of the free energy $F_v$ (*Equation S60*) in the variational approximation with $p(\sigma) = \delta(\sigma - 1)$

$$F_v = \frac{V}{2}\text{Tr}_n \int \frac{d^d\vec{k}}{(2\pi)^d}\frac{\tilde{G}(\vec{k})}{\tilde{f}(\vec{k})} - N(z)^{-\frac{n}{2}}\exp(-\frac{1}{2}\text{Tr}_n \log(1 - \frac{1}{z}\int \frac{d^d\vec{k}}{(2\pi)^d}\tilde{G}(\vec{k})))$$

$$- \frac{V}{2}\int \frac{d^d\vec{k}}{(2\pi)^d}\sum_{\alpha,\beta}\log(\tilde{G}_{\alpha\beta}(\vec{k})) \tag{S93}$$

$$= \frac{Vn}{2}\int \frac{d^d\vec{k}}{(2\pi)^d}\frac{\tilde{G}(\vec{k})}{\tilde{f}(\vec{k})} - N(z)^{-\frac{n}{2}}\left[1 - \frac{1}{z}\int \frac{d^d\vec{k}}{(2\pi)^d}\tilde{G}(\vec{k})\right]^{-\frac{n}{2}} - \frac{Vn}{2}\int \frac{d^d\vec{k}}{(2\pi)^d}\log\tilde{G}(\vec{k})$$

$$\lim_{n\to 0}F_v = \frac{Vn}{2}\int \frac{d^d\vec{k}}{(2\pi)^d}\frac{\tilde{G}(\vec{k})}{\tilde{f}(\vec{k})} + \frac{Nn}{2}\log\left[z - \int \frac{d^d\vec{k}}{(2\pi)^d}\tilde{G}(\vec{k})\right] - \frac{Vn}{2}\int \frac{d^d\vec{k}}{(2\pi)^d}\log\tilde{G}(\vec{k}) + N \tag{S94}$$

Following *Equations S61 and S63*:

$$\frac{\delta F_v}{\delta \tilde{G}} = 0 \tag{S95}$$

$$\frac{1}{\tilde{f}(\vec{k})} - \frac{\rho}{z - \int \frac{d^d\vec{k}}{(2\pi)^d}\tilde{G}(\vec{k})} - \frac{1}{\tilde{G}(\vec{k})} = 0 \tag{S96}$$

$$g(z) = \lim_{n \to 0} \frac{2}{nN} \frac{d}{dz} F_1 \approx \lim_{n \to 0} \frac{2}{nN} \frac{d}{dz} F_v = \lim_{n \to 0} \frac{2}{nN} \left( \frac{\partial}{\partial z} F_v + \int \frac{d^d \vec{k}}{(2\pi)^d} \frac{\partial \tilde{G}(\vec{k})}{\partial z} \frac{\partial}{\partial \tilde{G}(\vec{k})} F_v \right)$$

$$= \lim_{n \to 0} \frac{2}{nN} \frac{\partial}{\partial z} F_v = \frac{1}{z - \int \frac{d^d \vec{k}}{(2\pi)^d} \tilde{G}(\vec{k})} \tag{S97}$$

We can perform the same calculation in the high-density theory by considering the limit $ratio_v(z) = \frac{1}{z} \int \frac{d^d \vec{k}}{(2\pi)^d} \tilde{G}_v(\vec{k}) \to 0$:

$$\lim_{n \to 0} \lim_{ratio_v(z) \to 0} F_v = \frac{Vn}{2} \int \frac{d^d \vec{k}}{(2\pi)^d} \frac{\tilde{G}(\vec{k})}{\tilde{f}(\vec{k})} + \frac{Nn}{2} \log \left[ z - \int \frac{d^d \vec{k}}{(2\pi)^d} \tilde{G}(\vec{k}) \right]$$

$$- \frac{Vn}{2} \int \frac{d^d \vec{k}}{(2\pi)^d} \log \tilde{G}(\vec{k}) + N$$

$$= \frac{Vn}{2} \int \frac{d^d \vec{k}}{(2\pi)^d} \frac{\tilde{G}(\vec{k})}{\tilde{f}(\vec{k})} + \frac{Nn}{2} \log(z) - \frac{Nn}{2} \frac{1}{z} \int \frac{d^d \vec{k}}{(2\pi)^d} \tilde{G}(\vec{k})$$

$$- \frac{Vn}{2} \int \frac{d^d \vec{k}}{(2\pi)^d} \log \tilde{G}(\vec{k}) + N \tag{S98}$$

Therefore, we can define the free energy $F_h$ in the high-density theory as

$$F_h = \frac{Vn}{2} \int \frac{d^d \vec{k}}{(2\pi)^d} \frac{\tilde{G}(\vec{k})}{\tilde{f}(\vec{k})} + \frac{Nn}{2} \log(z) - \frac{Nn}{2} \frac{1}{z} \int \frac{d^d \vec{k}}{(2\pi)^d} \tilde{G}(\vec{k}) - \frac{Vn}{2} \int \frac{d^d \vec{k}}{(2\pi)^d} \log \tilde{G}(\vec{k}) + N \tag{S99}$$

then

$$\frac{\delta F_h}{\delta \tilde{G}} = 0 \tag{S100}$$

$$\frac{1}{\tilde{f}(\vec{k})} - \frac{\rho}{z} - \frac{1}{\tilde{G}(\vec{k})} = 0 \tag{S101}$$

This is precisely *Equation S90* derived in the previous section.

$$g(z) \approx \lim_{n \to 0} \frac{2}{nN} \frac{\partial}{\partial z} F_h = \frac{1}{z} + \frac{1}{z^2} \int \frac{d^d \vec{k}}{(2\pi)^d} \tilde{G}(\vec{k})$$

$$= \frac{1}{z} + \frac{1}{z^2} \int \frac{d^d \vec{k}}{(2\pi)^d} \frac{\tilde{f}(\vec{k})}{1 - \frac{\rho}{z} \tilde{f}(\vec{k})}$$

$$= \frac{1}{z} \left[ 1 + \int \frac{d^d \vec{k}}{(2\pi)^d} \frac{\tilde{f}(\vec{k})}{z - \rho \tilde{f}(\vec{k})} \right] \tag{S102}$$

$$= \frac{1}{z} \left[ \frac{1}{\rho} \int \frac{d^d \vec{k}}{(2\pi)^d} \frac{z - \rho \tilde{f}(\vec{k})}{z - \rho \tilde{f}(\vec{k})} + \int \frac{d^d \vec{k}}{(2\pi)^d} \frac{\tilde{f}(\vec{k})}{z - \rho \tilde{f}(\vec{k})} \right]$$

$$= \frac{1}{\rho} \int \frac{d^d \vec{k}}{(2\pi)^d} \frac{1}{z - \rho \tilde{f}(\vec{k})}$$

which is the resolvent of high-density approximation (*Equation S37*).

## Explicit expression for the integral

In this section, we provide an explicit expression for the integral $\int \frac{d^d \vec{k}}{(2\pi)^d} \tilde{G}_h(\vec{k})$ instead of $\int \frac{d^d \vec{k}}{(2\pi)^d} \tilde{G}_v(\vec{k})$, which is implicit and can not be calculated analytically. Like the derivation in *Appendix 1—figure 9*, we consider the lower and upper limits of integration for $\int \frac{d^d \vec{k}}{(2\pi)^d} \tilde{G}_h(\vec{k})$ as $[0, \frac{\pi}{\epsilon}]$. We then approximate

the Fourier transform $\tilde{f}(\vec{k})$ as a power-law function. To ensure that the singularity $\tilde{f}(\vec{k}_s) = \frac{z}{\rho}$ of $\tilde{G}_h(\vec{k})$ falls within the integration range of $[0, \frac{\pi}{\epsilon}]$, we introduce a simple correction $x_\epsilon = C(\frac{\pi}{\epsilon})^{\mu-d}$ to $\tilde{f}(\vec{k})$:

$$\tilde{f}(\vec{k}) = C\|\vec{k}\|^{\mu-d} - x_\epsilon \tag{103}$$

where $C = C_0\epsilon^\mu$, $C_0 = 2^{d-\mu}\pi^{\frac{d}{2}}\frac{\Gamma(\frac{d-\mu}{2})}{\Gamma(\frac{\mu}{2})}$ are all constants depending on the parameters $d$, $\mu$, and $\epsilon$. Then we compute the contour integral (**Appendix 2—figure 1A**) by Taylor expansion. As a result, we have

$$\int \frac{d^d\vec{k}}{(2\pi)^d}\tilde{G}_h(\vec{k}) = \int_0^{\frac{\pi}{\epsilon}} \frac{d^d\vec{k}}{(2\pi)^d}\frac{\tilde{f}(\vec{k})}{1 - \frac{\rho}{z}\tilde{f}(\vec{k})}$$

$$= \frac{1}{2\pi(\mu-d)}C^P\frac{z}{\rho}\left(\sum_{j=0}^{\infty}\frac{x_\epsilon^{1-P+j}(\frac{z}{\rho}+x_\epsilon)^{-1-j}}{1-P+j} - \pi\cot(\pi(1-P))(\frac{z}{\rho}+x_\epsilon)^{-P}\right)$$

$$- \frac{1}{2\pi(\mu-d)}C^P\frac{z}{\rho}\left(\sum_{j=0}^{\infty}\frac{x_\epsilon^{1-P+j}(\frac{z}{\rho}+x_\epsilon)^{-1-j}}{-P+j} - \pi\cot(\pi(-P))(\frac{z}{\rho}+x_\epsilon)^{-P-1}x_\epsilon\right) \tag{S104}$$

$$= \frac{1}{2\pi(d-\mu)}C^P\frac{z}{\rho}\left(\sum_{j=0}^{\infty}\frac{x_\epsilon^{1-P+j}(\frac{z}{\rho}+x_\epsilon)^{-1-j}}{(P-1-j)(P-j)} - \pi\cot(\pi P)(\frac{z}{\rho}+x_\epsilon)^{-P}\frac{z}{z+\rho x_\epsilon}\right)$$

$$= \frac{\pi^{d-1}z}{2(d-\mu)\rho\epsilon^d}\left(\sum_{j=0}^{\infty}\frac{(\frac{z}{\rho x_\epsilon}+1)^{-1-j}}{(P-1-j)(P-j)} - \pi\cot(\pi P)(\frac{z}{\rho x_\epsilon}+1)^{-P}\frac{z}{z+\rho x_\epsilon}\right)$$

where $P = \frac{d}{d-\mu} > 1$.

Now let us take a close look at the behavior of the function $ratio_h(z)$ **Equation S92**, plotted in **Appendix 2—figure 2A,B** . For small $z$, this function is negative. It then crosses zero and has a peak. As $z \to \infty$, the $ratio_h$ approaches zero. This is because **Equation S104** approaches a positive constant, which is given by

$$\lim_{z\to\infty}\int \frac{d^d\vec{k}}{(2\pi)^d}\tilde{G}_h(\vec{k}) = \frac{\pi^{d-1}C_2}{2(d-\mu)(P-1)P},$$

where $C_2 = C(\frac{\pi}{\epsilon})^\mu$. For $z \geq 1$, we find the leading order expansion at $j = 1$ already gives an accurate approximation (**Appendix 2—figure 2A,B**).

$$\int \frac{d^d\vec{k}}{(2\pi)^d}\tilde{G}_h(\vec{k}) \approx$$

$$\frac{1}{2\pi(d-\mu)}C^P\frac{z}{\rho}\left[\frac{x_\epsilon^{1-P}(\frac{z}{\rho}+x_\epsilon)^{-1}}{(P-1)(P)} + \frac{x_\epsilon^{2-P}(\frac{z}{\rho}+x_\epsilon)^{-2}}{(P-2)(P-1)} - \pi\cot(\pi P)(\frac{z}{\rho}+x_\epsilon)^{-\frac{d}{d-\mu}}\frac{z}{z+\rho x_\epsilon}\right] \tag{S105}$$

## Estimate the parameter condition when the high-density theory best agrees with numerical simulation

By analyzing the properties of the function $\int \frac{d^d\vec{k}}{(2\pi)^d}\tilde{G}_h(\vec{k})$, we think the high-density theory provides an accurate approximation when the zero-crossing of $\int \frac{d^d\vec{k}}{(2\pi)^d}\tilde{G}_h(\vec{k})$ is near $z = 1$ (the peak of low-density result **Mézard et al., 1999**) The root $z_0$ of the integral $\int \frac{d^d\vec{k}}{(2\pi)^d}\tilde{G}_h(\vec{k})$ is given by

$$\frac{\rho x_\epsilon}{z_0} = g_1(d, \mu) \tag{S106}$$

It is easy to see that $g_1(d, \mu)$ is a function of $P$ (or $\frac{d}{\mu}$) from **Equation S104**. We can rewrite **Equation S106** as

$$\frac{\rho x_\epsilon}{z_0} = g_2(\frac{d}{\mu}) \tag{S107}$$

Here, we can also see that $z_0$ can be expressed as:

$$z_0 = \frac{c_0 \pi^{\mu-d} \rho \epsilon^d}{g_1(d, \mu)}$$

Using this expression for $z_0$ and letting $z_0 = 1$, we can derive the following equation for $\rho \epsilon^d$, a *dimensionless parameter* that determines the condition when the high-density theory is an accurate approximation of our ERM model:

$$\rho \epsilon^d = \frac{z_0 g_2(\frac{d}{\mu})\Gamma(\frac{\mu}{2})}{2^{d-\mu} \pi^{-\frac{d}{2}} \Gamma(\frac{d-\mu}{2})} \tag{S108}$$

*Appendix 2—figure 2C* shows how $\rho \epsilon^d$ changes as a function $d$ for a small and fixed $\mu/d$. For example, when $d = 2$, $\mu = 0.5$, $\epsilon = 0.03125$, we find

$$\rho \epsilon^d = 0.83, \text{or } \rho = 850$$

This estimate is also consistent with our numerical simulation (*Figure 4—figure supplement 1*).

## Wick rotation

To ensure mathematical rigor in Appendix 2, we should make sure that the action $S_1$ in *Equation S46* is a real number. Here, we use Wick rotation to transform *Equations S28–S30*. The Gaussian integral *Equation S29* can be divergent when $G^{-1}(\vec{x} - \vec{y})$ is not positive definite, To address this issue, we can always write the partition function $\langle \Xi^n(z) \rangle$ as a Gaussian integral by choosing the appropriate axes with Wick rotation.

$$\langle \Xi^n(z) \rangle = (2\pi i)^{\frac{Nn}{2}} (\det f)^{-n/2} \int_{-\infty}^{+\infty} D[\hat{\psi}^\alpha] e^{S_1} \tag{S109}$$

$$S_1 = N \ln A - \frac{i}{2} \sum_{\alpha=1}^{n} \int_{-\infty}^{+\infty} d\vec{x} d\vec{x}' f^{-1}(\vec{x} - \vec{x}') \hat{\psi}^\alpha(\vec{x}) \hat{\psi}^\alpha(\vec{x}')$$

We can now change the integration variables by diagonalizing $\hat{\psi}^\alpha$ to $\tilde{\psi}^\alpha$ via $\hat{\psi}^\alpha = Q\tilde{\psi}^\alpha$, where $Q$ is Fourier base

$$\langle \Xi^n(z) \rangle = (2\pi i)^{\frac{Nn}{2}} (\det f)^{-n/2} \int_{-\infty}^{+\infty} D[\tilde{\psi}^\alpha] e^{S_1} \tag{S110}$$

$$S_1 = N \ln \tilde{A} - \frac{i}{2} \sum_{\alpha=1}^{n} \int_{-\infty}^{+\infty} d^d \vec{k} f^{-1}(\vec{k}) \tilde{\psi}^\alpha(\vec{k})^2$$

$$\tilde{A} = \int_{-\infty}^{\infty} \frac{d^d \vec{k}}{V} (z)^{\frac{n}{2}} \exp\left[\frac{i}{2z} \sum_{\alpha=1}^{n} \tilde{\psi}^\alpha(\vec{k})^2\right] \tag{S111}$$

by letting $L \to \infty$. Note that $e^{S_1}$ is analytic. Thus if

$$\lim_{\tilde{\psi}^\alpha \to (1-i)\infty} e^{S_1} = 0$$

and the convergence rate is faster than $1/\tilde{\psi}^2$, we can apply the Wick rotation $\psi^\alpha(\vec{x}) \to \psi^\alpha(\vec{x}) e^{-i\frac{\pi}{4}}$: instead of computing the integral on the real axis $C_1$, we now rotate the integral line 45 degree clockwise to $C_3$ in the complex plane:

$$\int_{C_1} D[\hat{\psi}^\alpha] e^{S_1} = \int_{C_3} D[\hat{\psi}^\alpha] e^{S_1} \tag{S112}$$

On the other hand, if

$$\lim_{\tilde{\psi}^\alpha \to (1+i)\infty} e^{S_1} = 0$$

and the convergence rate is faster than $1/\tilde{\psi}^2$, we can apply the Wick rotation $\psi^\alpha(\vec{x}) \to \psi^\alpha(\vec{x}) e^{i\frac{\pi}{4}}$, namely to rotate the integral line 45 degree counterclockwise to $C_2$:

$$\int_{C_1} D[\hat{\psi}^\alpha] e^{S_1} = \int_{C_2} D[\hat{\psi}^\alpha] e^{S_1} \tag{S113}$$

As a simple example, consider a one-dimensional Gaussian integral

$$\int_{-\infty}^{\infty} dx\, e^{-ikx^2}$$

When $k > 0$, we can use the Wick rotation $x \to x e^{-i\frac{\pi}{4}}$

$$\int_{-\infty}^{\infty} dx\, e^{-ikx^2} = e^{-i\frac{\pi}{4}} \int_{-\infty}^{\infty} dx\, e^{-kx^2} = e^{-i\frac{\pi}{4}} \sqrt{\frac{2\pi}{k}} = \sqrt{\frac{2\pi}{ik}}$$

When $k < 0$, we can use the Wick rotation $x \to x e^{i\frac{\pi}{4}}$

$$\int_{-\infty}^{\infty} dx\, e^{-ikx^2} = e^{i\frac{\pi}{4}} \int_{-\infty}^{\infty} dx\, e^{kx^2} = e^{i\frac{\pi}{4}} \sqrt{\frac{2\pi}{-k}} = \sqrt{\frac{2\pi}{ik}}$$

Without loss of generality, we rotate $\psi^\alpha(\vec{x}) \to \psi^\alpha(\vec{x}) e^{-i\frac{\pi}{4}}$ in Appendix 2 for subsequent calculations.

## Grand canonical ensemble

When using the Gaussian variational Approximation, we consider a critical extension from the *canonical ensemble* to the *grand canonical ensemble* when computing the partition function (*Equation S9*). We would like to justify this approximation in this section. Recall that the resolvent is given by

$$g(z) = -\frac{2}{N} \partial_z \langle \ln \Xi(z) \rangle$$

where $\Xi(z)$ can be viewed as the canonical partition function, the $\langle ... \rangle$ is the average over all random matrices $C$ for a given $N$. Let us now generalize (*Equation S9*) into grand canonical ensemble, namely

$$g(z) = \left\langle -\frac{2}{N} \partial_z \langle \ln \Xi(z) \rangle \right\rangle_N \tag{S114}$$

where $\langle ... \rangle_N$ indicates that we need to average over all possible random matrices and across all possible $N$, with the probability to have a matrix size $N$ given by the Poisson distribution $P(N) = e^{-a} \frac{a^N}{N!}$, where $a = \langle N \rangle$. When $\langle N \rangle$ is large, $P(N)$ has a very sharp peak at $\langle N \rangle$, and *Equation S114* can be approximated as

$$g(z) \approx -\frac{2}{\langle N \rangle} \partial_z \langle \ln \Xi(z) \rangle_{\langle N \rangle} \tag{S115}$$

Using the replica trick, we recall *Equation S12*

$$g(z) = \lim_{n \to 0} -\frac{2}{Nn} \partial_z \ln \langle \Xi^n(z) \rangle$$

Let us now define the grand canonical partition function as

$$\mathcal{Z} = \sum_{N=0}^{\infty} \langle \Xi_N^n(z) \rangle \frac{a^N}{N!}, \tag{S116}$$

Likewise, the resolvent in *Equation S12* is generalized to

$$g(z) = \lim_{n \to 0} -\frac{2}{\langle N \rangle n} \partial_z \ln \mathcal{Z}. \tag{S117}$$

To see whether this definition makes sense, we write

$$
\begin{aligned}
g(z) &= \lim_{n \to 0} -\frac{2}{\langle N \rangle n} \frac{\sum_{N=0}^{\infty} \partial_z \langle \Xi_N^n(z) \rangle a^N / N!}{\mathcal{Z}} \\
&= \lim_{n \to 0} -\frac{2}{\langle N \rangle n} \frac{\sum_{N=0}^{\infty} \partial_z [1 + n \langle \ln \Xi_N(z) \rangle] a^N / N!}{\sum_{N=0}^{\infty} \langle \Xi_N^n(z) \rangle \frac{a^N}{N!}} \\
&= -\frac{2}{\langle N \rangle} \frac{\sum_{N=0}^{\infty} \partial_z \langle \ln \Xi_N(z) \rangle a^N / N!}{\sum_{N=0}^{\infty} \frac{a^N}{N!}} \\
&= -\frac{2}{\langle N \rangle} \partial_z \Big\langle \ln \Xi(z) \Big\rangle_N,
\end{aligned}
\tag{S118}
$$

where the second equality uses the identity

$$\ln \Xi = \lim_{n \to 0} \frac{\Xi^n - 1}{n},$$

and the last equality is indeed *Equation S115* discussed earlier. Returning back to the explicit form of the grand canonical partition function in our ERM model (*Equations S29–S31*), we have

$$\mathcal{Z} = \int_{-\infty}^{+\infty} D[\hat{\psi}^\alpha] e^{S_0 + aA} = \int_{-\infty}^{+\infty} D[\hat{\psi}^\alpha] e^{S_0 + \langle N \rangle A}. \tag{S119}$$

Here, $\psi$ is the auxiliary fields (*Equation S15*), $S_0 = -\frac{1}{2} \sum_{\alpha=1}^{n} \int_{-\infty}^{+\infty} \mathrm{d}\vec{x} \mathrm{d}\vec{x}' f^{-1}(\vec{x} - \vec{x}') \hat{\psi}^\alpha(\vec{x}) \hat{\psi}^\alpha(\vec{x}')$ and $A$ are terms defined in *Equation S119*, *Equations S29–S31* is used in Appendix 2 to compute the free energy.

## E-I balanced asynchronized model summary

In this section, we discuss the E-I balanced asynchronized model (*Renart et al., 2010*), which predicts a different scaling D N under random sampling, since the variance $E_{i \neq j}^k(c_{ij}^2)$ scales as $1/N$ and diminishes as $N$ approaches large limit.

### Model

The simulation of binary networks involves updating neuron states within a network architecture identical to analytical studies. The update rule is probabilistic, with neuron activities set based on synaptic currents and a firing threshold. The dynamics resolution improves with network size, with neuron time constants effectively representing changes in firing activity. Parameters for simulations include connection probabilities, mean rates, thresholds, and synaptic strengths, scaled appropriately for network size.

Update rule: $\sigma_i^\alpha(t+1) = \Theta\left(\sum_j J_{ij}^{\alpha\beta} \sigma_j^\beta(t) - \theta_i^\alpha\right)$

Dynamics resolution: $dt = \frac{\tau}{3N}$

In the simulation of binary networks, the model's dynamics are governed by a set of parameters, each with a specific role:

$\sigma_i^\alpha(t+1)$: This represents the state of neuron $i$ in population $\alpha$ at the next time step $t+1$. The state is binary, where 1 indicates the neuron is active (firing) and 0 indicates it is inactive.

$\Theta(\cdot)$: The Heaviside step function used in the update rule. It determines the neuron's next state by comparing the net input to the neuron against its firing threshold. If the net input exceeds the threshold, the neuron's state is set to active; otherwise, it remains or becomes inactive.

$\sum_j J_{ij}^{\alpha\beta} \sigma_j^{\beta}(t)$: This sum represents the total synaptic input to neuron $i$ from all neurons $j$ in population $\beta$ at time $t$. $J_{ij}^{\alpha\beta}$ is the synaptic weight from neuron $j$ in population $\beta$ to neuron $i$ in population $\alpha$, and $\sigma_j^{\beta}(t)$ is the state of neuron $j$ at time $t$.

$\theta_i^{\alpha}$: The firing threshold of neuron $i$ in population $\alpha$. It is the value against which the net synaptic input is compared to determine whether neuron $i$ will fire (transition to state 1) or not (remain in state 0).

$\alpha = \{E, I\}$, $\beta = \{E, I, X\}$: Represents a specific population of neurons within the network. E: excitatory neurons, I: inhibitory neurons, or X: external source of neurons that provide inputs to the network but are not influenced by the network's internal dynamics.

## Firing Rate Correlation $r$

The mean firing rate correlation $E(r)$ scales inversely with the network size $N$, specifically, $E(r) \sim 1/N$. The standard deviation $\sigma_r$ of $r$ decays only as $1/\sqrt{N}$ (**Renart et al., 2010**).

Given that the variance of $r$, denoted as $\mathrm{Var}(r)$, is $\frac{b}{N}$, and the expected value of $r$, denoted as $E(r)$, is $\frac{a}{N}$, where $N$ is the sample size, and $a$ and $b$ are constants, we aim to calculate $E(r^2)$, the expected value of the square of the correlation coefficient $r$.

The term $E_{i \neq j}^k(c_{ij}^2)$ in PR dimension is given by:

$$\mathrm{Var}(r) = E(r^2) - [E(r)]^2 \tag{S120}$$

Substituting $\mathrm{Var}(r) = \frac{b}{N}$ and $E(r) = \frac{a}{N}$ into the equation, we get:

$$E_{i \neq j}^k(c_{ij}^2) = E(r^2) = \frac{b}{N} + \left(\frac{a}{N}\right)^2 \sim \frac{1}{N} \tag{S121}$$

Thus in PR dimension $D_{\mathrm{PR}}(C) = \frac{N^2 (E[\sigma^2])^2}{NE[\sigma^4] + N(N-1)E_{i \neq j}[c_{ij}^2]}$, the term $NE[\sigma^4]$ and $N(N-1)E_{i \neq j}[c_{ij}^2]$ are of the same order, and the PR dimension will not reach the upper bound $\frac{(E[\sigma^2])^2}{E_{i \neq j}[c_{ij}^2]}$.

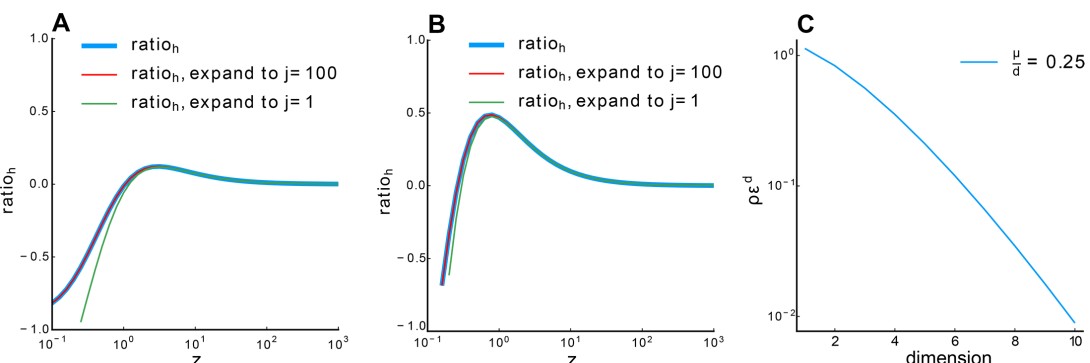

**Appendix 2—figure 2.** Relationship between $ratio_h$ and $z$. (**A**) $\rho = 1024$, (**B**) $\rho = 256$. Blue line: $ratio_h$ calculated numerically. Red line: 100-order expansion of **Equation S104**, which perfectly overlaps with the blue line. Green line: expansion to the first order. Other parameter: $\mu = 0.5$, $d = 2$, $\epsilon = 0.03125$. (**C**) Relationship between $\rho \epsilon^d$ and dimension $d$ with fixed $\frac{\mu}{d}$ **Equation S108**.

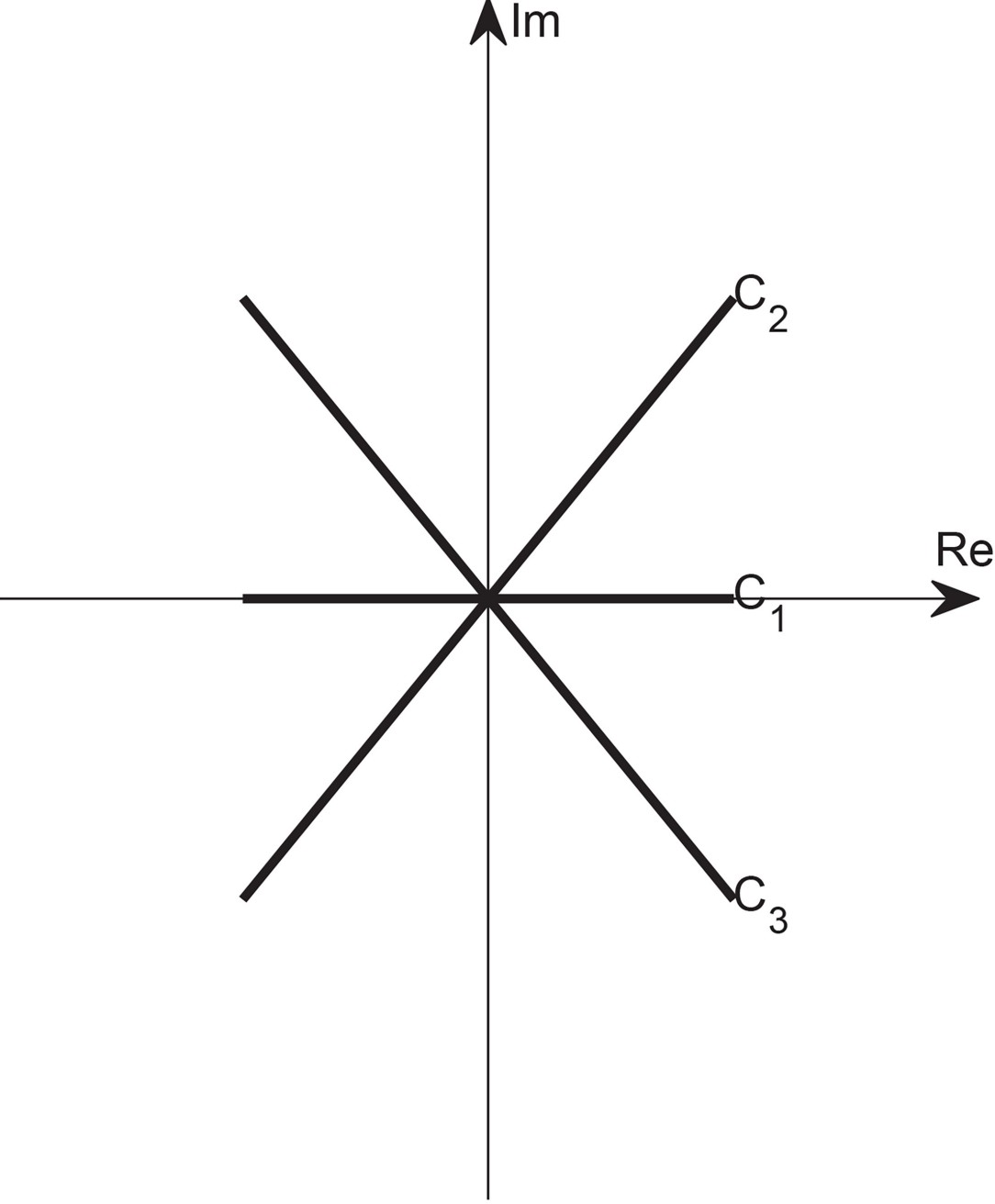

**Appendix 2—figure 3.** Wick rotation in complex plane.

**Appendix 2—table 1.** Table of notations.

| Notation | Description |
| --- | --- |
| $g(z)$ | Resolvent **Equation S5** |
| $\langle \dots \rangle$ | The average across realizations of $C$ (i.e., random $\vec{x}_i's$ and $\sigma_i^2's$), **Equation S4** |
| $\Xi(z)$ | Canonical partition function, Gaussian integral representation of the determinant $[\det(z - C)]^{1/2}$ , **Equation S8** |
| $\phi$ | Intermediate variable for Gaussian integral representation $\Xi(z)$, **Equation S8** |

*Appendix 2—table 1 Continued on next page*

*Appendix 2—table 1 Continued*

| Notation | Description |
| --- | --- |
| $\Psi$ | Density field of $\phi$ |
| $\hat{\Psi}$ | Respective Lagrange multiplier fields of $\Psi$ |
| $S_1$ | The action in $\Xi(z)$ (by analogy with the path integral formulation of quantum mechanics) |
| $S_h$ | The action in the high-density approximation of $\Xi(z)$ |
| $S_v$ | The action in the variational approximation of $\Xi(z)$ |
| $A$ | Term in $S_1$ |
| $f^{-1}$ | The operator inverse of $f$, **Equation S26** |
| $G$ | Quadratic kernel in the Gaussian integral approximation of $\Xi(z)$ |
| $G^{-1}$ | The operator inverse of $G$, same definition as $f^{-1}$ |
| $\tilde{G}$ | The Fourier transform of $G$ |

