## [Editor Report · eLife Assessment]

This **important** study shows a surprising scale-invariance of the covariance spectrum of large-scale recordings in the zebrafish brain in vivo. A **convincing** analysis demonstrates that a Euclidean random matrix model of the covariance matrix recapitulates these properties. The results provide several new and insightful approaches for probing large-scale neural recordings.

---

## [Referee Report · Joint public review]

Summary:

The authors examine the eigenvalue spectrum of the covariance matrix of neural recordings in the whole-brain larval zebrafish during hunting and spontaneous behavior. They find that the spectrum is approximately power law, and, more importantly, exhibits scale-invariance under random subsampling of neurons. This property is not exhibited by conventional models of covariance spectra, motivating the introduction of the Euclidean random matrix model. The authors show that this tractable model captures the scale invariance they observe. They also examine the effects of subsampling based on anatomical location or functional relationships. Finally, they discuss the benefit of neural codes which can be subsampled without significant loss of information.

Strengths:

With large-scale neural recordings becoming increasingly common, neuroscientists are faced with the question: how should we analyze them? To address that question, this paper proposes the Euclidean random matrix model, which embeds neurons randomly in an abstract feature space. This model is analytically tractable and matches two nontrivial features of the covariance matrix: approximate power law scaling, and invariance under subsampling. It thus introduces an important conceptual and technical advance for understanding large-scale simultaneously recorded neural activity.

Comment:

Are there quantitative comparisons of the collapse indices for the null models in Figure 2 and the data covariance in 2F? If so, this could be potentially useful to report.

---

## [Author Response]

The following is the authors’ response to the original reviews.

**Public Reviews:**
Summary:The authors examine the eigenvalue spectrum of the covariance matrix of neural recordings in the whole-brain larval zebrafish during hunting and spontaneous behavior. They find that the spectrum is approximately power law, and, more importantly, exhibits scale-invariance under random subsampling of neurons. This property is not exhibited by conventional models of covariance spectra, motivating the introduction of the Euclidean random matrix model. The authors show that this tractable model captures the scale invariance they observe. They also examine the effects of subsampling based on anatomical location or functional relationships. Finally, they briefly discuss the benefit of neural codes which can be subsampled without significant loss of information.Strengths:With large-scale neural recordings becoming increasingly common, neuroscientists are faced with the question: how should we analyze them? To address that question, this paper proposes the Euclidean random matrix model, which embeds neurons randomly in an abstract feature space. This model is analytically tractable and matches two nontrivial features of the covariance matrix: approximate power law scaling, and invariance under subsampling. It thus introduces an important conceptual and technical advance for understanding large-scale simultaneously recorded neural activity.Weaknesses:The downside of using summary statistics is that they can be hard to interpret. Often the finding of scale invariance, and approximate power law behavior, points to something interesting. But here caution is in order: for instance, most critical phenomena in neural activity have been explained by relatively simple models that have very little to do with computation (Aitchison et al., PLoS CB 12:e1005110, 2016; Morrell et al., eLife 12, RP89337, 2024). Whether the same holds for the properties found here remains an open question.

We are grateful for the thorough and constructive feedback provided on our manuscript. We have addressed each point raised by you.

Regarding the main concern about power law behavior and scale invariance, we would like to clarify that our study does not aim to establish criticality. Instead, we focus on describing and understanding a specific scale-invariant property in terms of collapsed eigenspectra in neural activity. We tested Morrell et al.’s latent-variable model (eLife 12, RP89337, 2024, [1]), where a slowly varying latent factor drives population activity. Although it produces a seemingly power-law-like spectrum, random sampling does not replicate the strict spectral collapse observed in our data (second row in Fig. S23). This highlights that simply adding latent factors does not fully recapitulate the scale invariance we measure, suggesting richer or more intricate processes may be involved in real neural recordings.

Specifically, we have incorporated five key revisions.

• As mentioned, we evaluated the latent variable model proposed by Morrell et al., and found that they fail to reproduce the scale-invariant eigenspectra observed in our data; these results are now presented in the Discussion section and supported by a new Supplementary Figure (Fig. S23).

• We included a comparison with the findings of Manley et al. (2024 [2]) regarding the issue of saturating dimension in the Discussion section, highlighting the methodological differences and their implications.

• We added a new mathematical derivation in the Methods section, elucidating the bounded dimensionality using the spectral properties of our model. • We have added a sentence in the Discussion section to further emphasize the robustness of our findings by demonstrating their consistency across diverse datasets and experimental techniques.

• We have incorporated a brief discussion on the implications for neural coding (lines 330-332). In particular, Fisher information can become unbounded when the slope of the power-law rank plot is less than one, as highlighted in the recent work by Moosavi et al. (bioRxiv 2024.08.23.608710, Aug, 2024 [3]).

We believe these revisions address the concerns raised during the review process and collectively strengthen our manuscript to provides a more comprehensive and robust understanding of the geometry and dimensionality of brain-wide activity. We appreciate your consideration of our revised manuscript and look forward to your feedback.

**Recommendations for the authors:**
In particular, in our experience replies to the reviewers are getting longer than the paper, and we (and I’m sure you!) want to avoid that. Maybe just reply explicitly to the ones you disagree with? We’re pretty flexible on our end.(1) The main weakness, from our point of view, is whether the finding of scale invariance means something interesting, or should be expected from a null model. We can suggest such model; if it is inconsistent with the data, that would make the results far more interesting.Morrell et al. (eLife 12, RP89337,2024 [1]) suggest a very simple model in which the whole population is driven by a slowly time-varying quantity. It would be nice to determine whether it matched this data. If it couldn’t, that would add some evidence that there is something interesting going on.

We appreciate your insightful suggestion to consider the model proposed by Morrell et al. (eLife 12, RP89337, 2024 [1]), where a slowly time-varying quantity drives the entire neural population. We conducted simulations using parameters from Morrell et al. [4, 1], as detailed below.

Our simulations show that Morrell’s model can replicate a degree of scaleinvariance when using *functional sampling* or RG as referred to in Morrell et al, 2021, PRL [4] (FSap, Fig.S23A-D, Author response image 1). However, it fails to fully capture the scale-invariance of collapsing spectra we observed in data under *random sampling* (RSap, Fig.S23E-H). This discrepancy suggests that additional dynamics or structures in the neural activity are not captured by this simple model, indicating the presence of potentially novel and interesting features in the data that merit further investigation.

Unlike random sampling, the collapse of eigenspectra under functional sampling does not require a stringent condition on the kernel function *f*(*x*) in our ERM theory (see Discussion line 269-275), potentially explaining the differing results between Fig.S23A-D and Fig.S23E-H.

We have incorporated these findings into the Result section 2.1 (lines 100-101) and Discussion section (lines 277-282, quoted below):

“Morrell et al. [4, 1] suggested a simple model in which a slow time-varying factor influences the entire neural population. To explore the effects of latent variables, we assessed if this model explains the scale invariance in our data. The model posits that neural activity is primarily driven by a few shared latent factors. Simulations showed that the resulting eigenspectra differed considerably from our findings (Fig. S23). Although the Morrell model demonstrated a degree of scale invariance under functional sampling, it did not align with the scale-invariant features under random sampling observed in our data, suggesting that this simple model might not capture all crucial features in our observations.”

**Author response image 1. sa2fig1:** Morrell’s latent model. A: We reproduce the results as presented in Morrell et al., PRL 126(11), 118302 (2021) [4]. Parameters are same as Fig. S23A. Sampled 16 to 256 neurons. Unlike in our study, the mean eigenvalues are not normalized to one. Dashed line: eigenvalues fitted to a power law. See also Morrell et al. [4] Fig.1C. Parameters are same as Author response image 1. *µ* is the power law exponent (black) of the fit, which is different from the *µ* parameter used to characterize the slow decay of the spatial correlation function, but corresponds to the parameter *α* in our study.

(2) The quantification of the degree of scale invariance is done using a ”collapse index” (CI), which could be better explained/motivated. The fact that the measure is computed only for the non-leading eigenvalues makes sense but it is not clear when originally introduced. How does this measure compare to other measures of the distance between distributions?

We thank you for raising this important point regarding the explanation and motivation for our Collapse Index (CI). We defined the Collapse Index (CI) instead of other measures of distance between distributions for two main reasons. First, the CI provides an intuitive quantification of the shift of the eigenspectrum motivated by our high-density theory for the ERM model (Eq. 3, Fig. 4A). This high-density theory is only valid for large eigenvalues excluding the leading ones, and hence we compute the CI measure with a similar restriction of the range of area integration. Second, when using distribution to assess the collapse (e.g., we can use kernel density method to estimate the distribution of eigenvalues and then calculate the KL divergence between the two distributions), it is necessary to first estimate the distributions. This estimation step introduces errors, such as inaccuracies in estimating the probability of large eigenvalues.

We agree that a clearer explanation would enhance the manuscript and thus have made modifications accordingly. The CI is now introduced more clearly in the Results section (lines 145-148) and further detailed in the Methods section (lines 630-636). We have also revised the CI diagram in Fig. 4A to better illustrate the shift concept using a more intuitive cartoon representation.

(3) The paper focuses on the case in which the dimensionality saturates to a finite value as the number of recorded neurons is increased. It would be useful to contrast with a case in which this does not occur. The paper would be strengthened by a comparison with Manley et al. 2024, which argued that, unlike this study, dimensionality of activity in spontaneously behaving head-fixed mice did not saturate.

Thank you for highlighting this comparison. We have included a discussion (lines 303-309) comparing our approach with Manley et al. (2024) [2]. While Manley et al. [2] primarily used shared variance component analysis (SVCA) to estimate neural dimensionality, they observed that using PCA led to dimensionality saturation (see Figure S4D, Manley et al. [2]), consistent with our findings (Fig. 2D). We acknowledge the value of SVCA as an alternative approach and agree that it is an interesting avenue for future research. In our study, we chose to use PCA for several reasons. PCA is a well-established and widely trusted method in the neuroscience community, with a proven track record of revealing meaningful patterns in neural data. Its mathematical properties are well understood, making it particularly suitable for our theoretical analysis. While we appreciate the insights that newer methods like SVCA can provide, we believe PCA remains the most appropriate tool for addressing our specific research questions.

(4) More importantly, we don’t understand why dimensionality saturates. For the rank plot given in Eq. 3,\begin{document}$$\displaystyle \lambda_{k} \sim 1 / k^{-1+\mu / d}$$\end{document}where k is rank. Using this, one can estimate sums over eigenvalues by integrals. Focusing on the N-dependence, we have\begin{document}$$\displaystyle \begin{aligned} &\displaystyle\sum_{k} \lambda_{k} \sim \int_{1}^{N} d k / k^{-1+\mu / d} \sim N^{\mu / d} \\ &\displaystyle\sum_{k} \lambda_{k}^{2} \sim \int_{1}^{N} d k / k^{-2+2 \mu / d} \sim N^{-1+2 \mu / d}-1\end{aligned}$$\end{document}This gives\begin{document}$$\displaystyle D_{P R} \sim N^{\min \{1,2 \mu / d\}}$$\end{document}We don’t think you ever told us what mu/d was (see point 13 below), but in the discussion you implied that it was around 1/2 (line 249). In that case, *DPR* should be approximately linear in N. Could you explain why it isn’t?

Thank you for your careful derivation. Along this line of calculations you suggested, we have now added derivations on using the ERM spectrum to estimate the upper bound of the dimension in the Methods (section 4.14.4). To deduce *DPR* from the spectrum, we focus on the high-density region, where an analytical expression for large eigenvalues *λ* is given by:\begin{document}$$\displaystyle \begin{aligned} \lambda_{r} &= \displaystyle \gamma\left(\frac{r}{N}\right)^{-1+\frac{\mu}{d}} \cdot \rho^{\frac{\mu}{d}}=\gamma r^{-1+\frac{\pi}{d}} L^{-\mu} N \quad \text { for } \quad r \leq \beta(N), \\ \lambda_{r} &= \displaystyle \eta(r, N, L) \text { for } \quad r\gt\beta(N) .\end{aligned}$$\end{document}

Here, *d* is dimension of functional space, *L* is the linear size of functional space, *ρ* is the neuron density and *γ* is the coefficient in Eq. (3), which only depends on *d*, *µ* and E(σ^2^). The primary difference between your derivation and ours is that the eigenvalue *λr* decays rapidly after the threshold *r* = *β*(*N*), which significantly affects the summations \begin{document}$\sum_{r=1}^{N} \lambda_{r}$\end{document} and \begin{document}$\sum_{r=1}^{N} \lambda_{r}^{2}$\end{document}. Since we did not discuss the small eigenvalues in the article, we represent them here as an unknown function *η*(*r,N,L*).

The sum \begin{document}$\sum_{r=1}^{N} \lambda_{r}$\end{document} is the trace of the covariance matrix *C*. As emphasized in the Methods section, without changing the properties the covariance spectrum, we always consider a normalized covariance matrix such that the mean neural activity variance E(σ^2^) = 1. Thus\begin{document}$$\displaystyle \sum_{r=1}^{N} \lambda_{r}=\operatorname{Tr}(C)=N$$\end{document}

rather than\begin{document}$$\displaystyle \sum_{r=1}^{N} \lambda_{\tau} \sim \gamma L^{-\mu} N \int_{1}^{N} d k / k^{-1+\mu / d} \sim\left(\gamma L^{-\mu} N\right) N^{\mu / d}$$\end{document}

The issue stems from overlooking that Eq. (3) is valid only for large eigenvalues (*λ >* 1).

Using the Cauchy–Schwarz inequality, we have a upper bound of \begin{document}$\sum_{r=1}^{N} \lambda_{r}^{2}$\end{document}\begin{document}$$\displaystyle \sum_{r=1}^{N} \lambda_{r}^{2} \leq\left(\sum_{r} \lambda_{r}\right)^{2}=N^{2}$$\end{document}

Conversely, \begin{document}$\lambda_{1}^{2}$\end{document} provides a lower bound of \begin{document}$\sum_{r=1}^{N} \lambda_{r}^{2}$\end{document}:\begin{document}$$\displaystyle \sum_{r=1}^{N} \lambda_{r}^{2}\gt\lambda_{1}^{2}=L^{-2 \mu} N^{2} \gamma^{2}$$\end{document}

As a result, we must have\begin{document}$$\displaystyle 1 \leq D_{P R}=\frac{\left(\sum_{r-1}^{N} \lambda_{r}\right)^{2}}{\sum_{r-1}^{N} \lambda_{r}^{2}}< L^{2 \mu} \gamma^{-2}$$\end{document}

In random sampling (RSap), *L* is fixed. We thus must have a bounded dimensionality that is independent of *N* for our ERM model. In functional sampling (FSap), *L* varies while the neuronal density *ρ* is fixed, leading to a different scaling relationship of the upper bound, see Methods (section 4.14.4) for further discussion.

(5) The authors work directly with ROIs rather than attempting to separate the signals from each neuron in an ROI. It would be worth discussing whether this has a significant effect on the results.

We appreciate your thoughtful question on the potential impact of using ROIs. The use of ROIs likely does not impact our key findings since they are validated across multiple datasets with various recording techniques and animal models, from zebrafish calcium imaging to mouse brain multi-electrode recordings (see Figure S2, S24). The consistency of the scale-invariant covariance spectrum in diverse datasets suggests that ROIs in zebrafish data do not significantly alter the conclusions, and they together enhance the generalizability of our results. We highlight this in the Discussion section (lines 319-323).

(6) Does the Euclidean random matrix model allow the authors to infer the value of *D* or *µ*? Since the measured observables only depend on *µ/D* it seems that one cannot infer the latent dimension where distances between neurons are computed. Are there any experiments that one could, in principle, perform to measure D or mu? Currently the conclusion from the model and data is that *D/µ* is a large number so that the spectrum is independent of neuron density rho. What about the heterogeneity of the scales *σi*, can this be constrained by data?

Measuring *d* and *µ* in the ERM Model

We agree with you that the individual values of *d* and *µ* cannot be determined separately from our analysis. In our analysis using the Euclidean Random Matrix (ERM) model, we fit the ratio *µ/d*, rather than the individual values of *d* (dimension of the functional space) or *µ* (exponent of the distance-dependent kernel function). This limitation is inherent because the model’s predictions for observable quantities, such as the distribution of pairwise correlation, are dependent solely on this ratio.

Currently there are no directly targeted experiments to measure *d*. The dimensions of the functional space is largely a theoretical construct: it could serve to represent latent variables encoding cognitive factors that are distributed throughout the brain or specific sensory or motor feature maps within a particular brain region. It may also be viewed as the embedding space to describe functional connectivity between neurons. Thus, a direct experimental measurement of the dimensions of the functional space could be challenging. Although there are variations in the biological interpretation of the functional space, the consistent scale invariance observed across various brain regions indicates that the neuronal relationships within the functional space can be described by a uniform slowly decaying kernel function.

Regarding the Heterogeneity *of σi*

The heterogeneity of neuronal activity variances (*σi*) is a critical factor in our analysis. Our findings indicate that this heterogeneity:

(1) Enhances scale invariance: The covariance matrix spectrum, which incorporates the heterogeneity of \begin{document}$\sigma_{i}^{2}$\end{document}, exhibits stronger scale invariance compared to the correlation matrix spectrum, which imposes \begin{document}$\sigma_{i}^{2} \equiv 1$\end{document} for all neurons. This observation is supported by both experimental data and theoretical predictions from the ERM model, particularly in the intermediate density regime.

(2) Can be constrained by data: We fit a log-normal distribution to the experimentally observed *σ2* values to capture the heterogeneity in our model which leads to excellent agreement with data (section 4.8.1). Figure S10 provides evidence for this by directly comparing the eigenspectra obtained from experimental data (Fig S10A-F) with those generated by the fitted ERM model (Fig S10M-R). These results suggest that the data provides valuable information about the distribution of neuronal activity variances.

In conclusion, the ERM model and our analysis cannot separately determine *d* and *µ*. We also highlight that the neuronal activity variance heterogeneity, constrained by experimental data, plays a crucial role in improving the scale invariance.

(7) Does the fitting procedure for the positions x in the latent space recover a ground truth in your statistical regime (for the number of recorded neurons)? Suppose you sampled some neurons from a Euclidean random matrix theory. Does the MDS technique the authors use recover the correct distances?

While sampling neurons from a Euclidean random matrix model, we demonstrated numerically that the MDS technique can accurately recover the true distances, provided that the true parameter *f*(*x*) is known. To quantify the precision of recovery, we applied the CCA analysis (Section 4.9) and compared the true coordinates \begin{document}$\vec{x}_{i}$\end{document} from the original Euclidean random matrix with the fitted coordinates \begin{document}$\vec{x}_{i}^{*}$\end{document} obtained through our MDS procedure. The CCA correlation between the true and fitted coordinates in each spatial dimension is nearly 1 (the difference from 1 is less than 10^−7^). When fitting with experimental data, one source of error arises from parameter estimation. To evaluate this, we assess the estimation error of the fitted parameters. When we choose µ = 0_.5 in our ERM model and then fit the distribution of the pairwise correlation (Eq. 21), the estimated parameter is \begin{document}$\bar{\mu}$\end{document} = 0.503 ± 0._007 (standard deviation). Then, we use the MDS-recovered distances to fit the coordinates with the fitted kernel function \begin{document}$\tilde{f}(x)$\end{document} , which is determined by the fitted parameter ^−5^).\begin{document}$\bar{\mu}$\end{document}. The CCA correlation between the true and fitted coordinates in each direction remains nearly 1 (the difference from 1 is less than 10

(8) l. 49: ”... both the dimensionality and covariance spectrum remain invariant ...”. Just to be clear, if the spectrum is invariant, then the dimensionality automatically is too. Correct?

Thanks for the question. In fact, there is no direct causal relationship between eigenvalue spectrum invariance and dimensionality invariance as we elaborate below and added discussions in lines 311-317. For eigenvalue spectrum invariance, we focus on the large eigenvalues, whereas dimensionality invariance considers the second order statistics of all eigenvalues. Consequently, the invariance results for these two concepts may differ. And dimensional and spectral invariance have different requirements:

(1) The condition for dimensional saturation is finite mean square covariance

The participation ratio *DPR* for random sampling (RSap) is given by Eq. 5:\begin{document}$$\displaystyle D_{P R}(C)=\frac{N^{2} \mathrm{E}\left(\sigma^{2}\right)^{2}}{N \mathrm{E}\left(\sigma^{4}\right)+N(N-1) \mathrm{E}_{i \neq j}\left(C_{i j}^{2}\right)}$$\end{document}

This expression becomes invariant as *N* → ∞ if the mean square covariance is finite. In contrast, neural dynamics models, such as the balanced excitatory-inhibitory (E-I) neural network [5], exhibit a different behavior, where \begin{document}$\mathrm{E}_{i \neq j}\left(C_{i j}^{2}\right) \sim 1 / N$\end{document}, leading to unbounded dimensionality (see discussion lines 291-295, section 6.9 in SI).

(2) The requirements for spectral invariance involving the kernel function

In our Euclidean Random Matrix (ERM) model, the eigenvalue distribution follows:\begin{document}$$\displaystyle \lambda \sim\left(\frac{r}{N}\right)^{-1+\mu / d} \cdot \rho^{\mu / d}$$\end{document}

For spectral invariance to emerge: (1) The eigenvalue distribution must remain unchanged after sampling. (2) Since sampling reduces the neuronal density *ρ*. (3) The ratio *µ/d* must approach 0 to maintain invariance.

We can also demonstrate that *DPR* is independent of density *ρ* in the large *N* limit (see the answer of question 4).

In conclusion, there is no causal relationship between spectral invariance and dimensionality invariance. This is also the reason why we need to consider both properties separately in our analysis.

(9) In Eq. 1, the exact expression, which includes i=j, isn’t a lot harder than the one with i=j excluded. So why *i≠j*?

The choice is for illustration purposes. In Eq. 1, we wanted to demonstrate that the dimension saturates to a value independent of *N*. When dividing the numerator and denominator of this expression by *N2*, the term \begin{document}$E_{i \neq j}\left(C_{i j}^{2}\right)$\end{document} is independent of the neuron number *N*, but the term associated with the diagonal entries \begin{document}$\frac{1}{N} E\left(C_{i i}^{2}\right)$\end{document} is of order *O*(1_/N_) and can be ignored for large *N*.

(10) Fig. 2D: Could you explain where the theory line comes from?

We first estimate \begin{document}$E_{i \neq j}\left(C_{i j}^{2}\right), E\left(\sigma^{2}\right), E\left(\sigma^{4}\right)$\end{document} from all neurons, and then compute *DPR* for different neuron numbers *N* using Eq.5 (\begin{document}$D_{P R}(C)=\frac{N^{2} \mathrm{E}\left(\sigma^{2}\right)^{2}}{N \mathrm{E}\left(\sigma^{4}\right)+N(N-1) \mathrm{E}_{i \neq j}\left(C_{i j}^{2}\right)}$\end{document}). This is further clarified in lines 511-512.

(11) l 94-5: ”It [scale invariance] is also absent when replacing the neural covariance matrix eigenvectors with random ones, keeping the eigenvalues identical (Fig. 2H).” If eigenvalues are identical, why does the spectrum change?

The eigenspectra of the covariance matrices in full size are the same by construction, but the eigenspectra of the sampled covariance matrices are different because the eigenvectors affect the sampling results. Please also refer to the construction process described in section 4.3 where this is also discussed: “The composite covariance matrix with substituted eigenvectors in (Fig. 2H) was created as described in the following steps. First, we generated a random orthogonal matrix *Ur<.sup>* (based on the Haar measure) for the new eigenvectors. This was achieved by QR decomposition *A=UrR* of a random matrix *A* with i.i.d. entries *Aij* ∼ N(0_,*1*/N_). The composite covariance matrix *Cr* was then defined as, where Λ is a diagonal matrix that contains the eigenvalues of *C*. Note that since all the eigenvalues are real and *Ur* is orthogonal, the resulting *Cr* is a real and symmetric matrix. By construction, *Cr* and C have the same eigenvalues, but their sampled eigenspectra can differ.”

(12) Eq 3: There’s no dependence on the distribution of sigma. Is that correct?

Indeed, this is true in the high-density regime when the neuron density *ρ* is large. The *p*(*λ*) depends only on *E*(*σ2*) rather than the distribution of *σ* (see Eq. 8). However, in the intermediate density regime, *p*(*λ*) depends on the distribution of *σ* (see Eq.9 and Eq.10). In our analysis, we consider *E*(*σ4*) as a measure of heterogeneity.

(13) Please tell us the best fit values of *µ/d*.

This information now is added in the figure caption of Fig S10: *µ/d* = [0_.*456*,*0*.*258*,*0*.*205*,*0*.*262*,*0*.*302*,*0*._308] in fish 1-6.

(14) l 133: ”The eigenspectrum is rho-independent whenever *µ/d* ≈ 0.”It looks to me like rho sets the scale but not the shape. Correct? If so, why do we care about the overall scale – isn’t it the shape that’s important?

Yes, our study focuses on the overall scale not only the shape, because many models, such as the ERM with other kernel functions, random RNNs, Morrell’s latent model [4, 1], can exhibit a power-law spectrum. However, these models do not exhibit scale-invariance in terms of spectrum curve collapsing. Therefore, considering the overall scale reveal additional non-trivial phenomenon.

(15) Figs. 3 and 4: Are the grey dots the same as in previous figures? Either way, please specify what they are in the figure caption.

Yes, they are the same, and thank you for pointing it out. It has been specified in the figure caption now.

(16) Fig. 4B: Top is correlation matrix, bottom is covariance matrix, correct? If so, that should be explicit. If not, it should be clear what the plots are.

That is correct. Both matrices (correlation - top, covariance - bottom) are labeled in the figure caption and plot (text in the lower left corner).

(17) l 158: ”First, the shape of the kernel function f(x) over a small distance ...”. What does ”over a small distance” mean?

We thank you for seeking clarification on this point. We understand that the phrase ”over a small distance” could be made clearer. We made a revised explanation in lines 164-165 Here, “over a small distance” refers to modifications of the particular kernel function *f(x)* we use Eq. 11 near *x* = 0 in the functional space, while preserving the overall power-law decay at larger distances. The t-distribution based *f(x)* (Eq. 11) has a natural parameter *ϵ* that describes the transition to near 0. So we modified *f(x)* in different ways, all within this interval of |*x*| ≤ *ϵ*, and considered different values of *ϵ*. Table S3 and Figure S7 provide a summary of these modifications. Figure S7 visually compares these modifications to the standard power-law kernel function, highlighting the differences in shape near *x* = 0.

Our findings indicate that these alterations to the kernel function at small distances do not significantly affect the distribution of large eigenvalues in the covariance spectrum. This supports our conclusion that the large eigenvalues are primarily determined by the slow decay of the kernel function at larger distances in the functional space, as this characteristic governs the overall correlations in neural activity.

(18) l390 \begin{document}$C_{i j}=1 /(T-1) \sum_{t=1}^{T}\left(x_{i}(t)-\bar{x}_{i}\right)\left(x_{j}(t)-\bar{x}_{j}\right)$\end{document}. This *xi* is, we believe, different from the *xi* which is position in feature space. Given the difficulty of this paper, it doesn’t help to use the same symbol to mean two different things. But maybe we’re wrong?

Thank you for your careful reading and suggestion. Indeed here *xi* was representing activity rather than feature space position. We have thus revised the notation (Line 390 has been updated to line 439 as well.):\begin{document}$$\displaystyle C_{i j}=\frac{1}{T-1} \sum_{t=1}^{T}\left(a_{i}(t)-\bar{a}_{i}\right)\left(a_{j}(t)-\bar{a}_{j}\right)$$\end{document}

In this revised notation: *ai*(*t*) represents the neural activity of neuron *i* at time *t* (typically the firing rate we infer from calcium imaging). \begin{document}$\bar{a}_{i}$\end{document} is simply the mean activity of neuron *i* across time. Meanwhile, we’ll keep *xi* exclusively for denoting positions in the functional space.

This change should make it much easier to distinguish between neural activity measurements and spatial coordinates in the functional space.

(19) Eq. 19: is it correct that *g*(*u*) is not normalized to 1? If so, does that matter?

It is correct that the approximation of *g*(*u*) is not normalized to 1, as Eq. 19 provides an approximation suitable only for small pairwise distances (i.e., large correlation). Therefore, we believe this does not pose an issue. We have newly added this note in lines 691-693.

(20) I get a different answer in Eq. 20:\begin{document}$$\begin{aligned}g(u) \sim S_{d-1} \sim u^{d-1} &= \displaystyle \epsilon^{d-1}(u / \epsilon)^{d-1}=\epsilon^{d-1} / R^{d-1} \\ u &=\epsilon R^{-1 / \mu} \\ |d u / d R| &=\displaystyle (\epsilon / \mu) / R^{1+1 / \mu} \\ g(u)|d u / d R| &\sim\displaystyle \epsilon^{d-1}(\epsilon / \mu) / R^{d-1+1+1 / \mu}=\left(\epsilon^{d} / \mu\right) / R^{d+1 / \mu}\end{aligned}$$\end{document}Whereas in Eq. 20,\begin{document}$$ \displaystyle g(u)|d u / d R| \sim \epsilon^{d-1}(\epsilon / \mu) / R^{d-1+1+1 / \mu}=\left(\epsilon^{d} / \mu\right) / R^{1+d / \mu}$$\end{document}Which is correct?

Thank you for your careful derivation. We believe the difference arises in the calculation of *g*(*u*).In our calculations:\begin{document}$$\displaystyle g(u) \sim u^{d-1} \sim\left(\epsilon R^{-1 / \mu}\right)^{d-1} \sim \epsilon^{d-1} / R^{\frac{d-1}{\mu}}$$\end{document}

(Your first equation seems to missed an 1_/µ_ in *R*’s exponent.)\begin{document}$$\displaystyle g(u)|d u / d R| \sim \epsilon^{d-1} R^{\frac{1-d}{\mu}}(\epsilon / \mu) R^{-1-1 / \mu}=\left(\epsilon^{d} / \mu\right) / R^{1+d / \mu}$$\end{document}

That is, Eq. 20 is correct. From these, we obtain\begin{document}$$\displaystyle g(u) \sim \epsilon^{d-1} / R^{\frac{d-1}{\mu}}$$\end{document}

rather than\begin{document}$$\displaystyle g(u) \sim \epsilon^{d-1} / R^{d-1}$$\end{document}

We hope this clarifies the question.

(21) I’m not sure we fully understand the CCA analysis. First, our guess as to what you did: After sampling (either Asap or Fsap), you used ERM to embed the neurons in a 2-D space, and then applied canonical correlation analysis (CCA). Is that correct? If so, it would be nice if that were more clear.

We first used ERM to embed all the neurons in a 2-D functional space, before any sampling. Once we have the embedding, we can quantify how similar the functional coordinates are with the anatomical coordinates using *RCCA* (section 2.4). We can then use the anatomical and functional coordinates to perform ASap and FSap, respectively. Our theory in section 2.4 predicts the effect on dimension under these samplings given the value of *RCCA* estimated earlier (Fig. 5D). The detailed description of the CCA analysis is in section 4.9, where we explain how CCA is used to find the axes in both anatomical and functional spaces that maximize the correlation between projections of neuron coordinates.

As to how you sampled under Fsap, I could not figure that out – even after reading supplementary information. A clearer explanation would be very helpful.

Thank you for your feedback. Functional sampling (FSap) entails the expansion of regions of interest (ROIs) within the functional space, as illustrated in Figure 5A, concurrently with the calculation of the covariance matrix for all neurons contained within the ROI. Technically, we implemented the sampling using the RG approach [6], which is further elaborated in Section 4.12 (lines 852-899), quoted below.

Stage (i): Iterative Clustering We begin with *N*0 neurons, where *N*0 is assumed to be a power of 2. In the first iteration, we compute Pearson’s correlation coefficients for all neuron pairs. We then search greedily for the most correlated pairs and group the half pairs with the highest correlation into the first cluster; the remaining neurons form the second cluster. For each pair (*a,b*), we define a coarse-grained variable according to:\begin{document}$$\displaystyle x_{i}^{k}=Z_{a b}^{k-1}\left(x_{a}^{k-1}+x_{b}^{k-1}\right)$$\end{document}

where \begin{document}$Z_{a b}^{k-1}$\end{document} normalizes the average to ensure unit nonzero activity. This process reduces the number of neurons to *N*_1_ = *N*_0_/2. In subsequent iterations, we continue grouping the most correlated pairs of the coarse-grained neurons, iteratively reducing the number of neurons by half at each step. This process continues until the desired level of coarse-graining is achieved.

When applying the RG approach to ERM, instead of combining neural activity, we merge correlation matrices to traverse different scales. During the _k_th iteration, we compute the coarse-grained covariance as:\begin{document}$$\displaystyle c_{i j}^{k}=c_{a b}^{k-1}+c_{a c}^{k-1}+c_{b c}^{k-1}+c_{b d}^{k-1}$$\end{document}

and the variance as:\begin{document}$$\displaystyle c_{i i}^{k}=c_{a a}^{k-1}+c_{b b}^{k-1}+2 c_{a b}^{k-1}$$\end{document}

Following these calculations, we normalize the coarse-grained covariance matrix to ensure that all variances are equal to one. Note that these coarse-grained covariances are only used in stage (i) and not used to calculate the spectrum.

Stage (ii): Eigenspectrum Calculation The calculation of eigenspectra at different scales proceeds through three sequential steps. First, for each cluster identified in Stage (i), we compute the covariance matrix using the original firing rates of neurons within that cluster (not the coarse-grained activities). Second, we calculate the eigenspectrum for each cluster. Finally, we average these eigenspectra across all clusters at a given iteration level to obtain the representative eigenspectrum for that scale.

In stage (ii), we calculate the eigenspectra of the sub-covariance matrices across different cluster sizes as described in [6]. Let *N*_0_ = 2*n* be the original number of neurons. To reduce it to size *N* = *N*_0_/2*k* = 2*n-k*, where *k* is the kth reduction step, consider the coarse-grained neurons in step *n* − *k* in stage (i). Each coarse-grained neuron is a cluster of 2*n-k* neurons. We then calculate spectrum of the block of the original covariance matrix corresponding to neurons of each cluster (there are 2*k* such blocks). Lastly, an average of these 2*k* spectra is computed.

For example, when reducing from *N*_0_ = 2^3^ = 8 to *N* = 2^3−1^ = 4 neurons (*k* = 1), we would have two clusters of 4 neurons each. We calculate the eigenspectrum for each 4x4 block of the original covariance matrix, then average these two spectra together. To better understand this process through a concrete example, consider a hypothetical scenario where a set of eight neurons, labeled 1,2,3,...,7,8, are subjected to a two-step clustering procedure. In the first step, neurons are grouped based on their maximum correlation pairs, for example, resulting in the formation of four pairs: {1,2},{3,4},{5,6}, and {7,8} (see Fig. S22). Subsequently, the neurons are further grouped into two clusters based on the results of the RG step mentioned above. Specifically, if the correlation between the coarse-grained variables of the pair {1,2} and the pair {3,4} is found to be the largest among all other pairs of coarse-grained variables, the first group consists of neurons {1,2,3,4}, while the second group contains neurons {5,6,7,8}. Next, take the size of the cluster *N* = 4 for example. The eigenspectra of the covariance matrices of the four neurons within each cluster are computed. This results in two eigenspectra, one for each cluster. The correlation matrices used to compute the eigenspectra of different sizes do not involve coarse-grained neurons. It is the real neurons 1,2,3,...,7,8, but with expanding cluster sizes. Finally, the average of the eigenspectra of the two clusters is calculated.

(22) Line 37: ”even if two cell assemblies have the same *DPR*, they can have different shapes.” What is meant by shape here isn’t clear.

Thank you for pointing out this potential ambiguity. The “shape” here refers to the geometric configuration of the neural activity space characterized as a highdimensional ellipsoid by the covariance. Specifically, if we denote the eigenvalues of the covariance matrix as *λ1,λ2,...,λN*, then \begin{document}$\sqrt{\lambda_{i}}$\end{document} corresponds to the length of the *i*-th semi-axis of this ellipsoid (Figure 1B). As shown in Figure 1C, two neural populations with the same dimensionality (*DPR* = 25/11 ≈ 2.27) exhibit different eigenvalue spectra, leading to differently shaped ellipsoids. This clarification is now included in lines 39-40.

(23) Please discuss if any information about the latent dimension or kernel function can be inferred from the measurements.

Same as comment(6): we would like to clarify that in our analysis using the Euclidean Random Matrix (ERM) model, we fit the ratio *µ/d*, rather than the individual values of *d* (dimension of the functional space) or *µ* (exponent of the distancedependent kernel function). This limitation is inherent because the model’s predictions for observable quantities, such as the eigenvalue spectrum of the covariance matrix, are dependent solely on this ratio.

For the kernel function, once the *d* is chosen, we can infer the general shape of the kernel function from data (Figs S12 and S13), up to a certain extent (see also lines 164-166). In particular, we can compare the eigenspectrum of the simulation results for different kernel functions with the eigenspectrum of our data. This allows us to qualitatively exclude certain kernel functions, such as the exponential and Gaussian kernels (Fig. S4), which show clear differences from our data.

References

(1) M. C. Morrell, I. Nemenman, A. Sederberg, Neural criticality from effective latent variables. eLife 12, RP89337 (2024).

(2) J. Manley, S. Lu, K. Barber, J. Demas, H. Kim, D. Meyer, F. M. Traub, A. Vaziri, Simultaneous, cortex-wide dynamics of up to 1 million neurons reveal unbounded scaling of dimensionality with neuron number. Neuron (2024).

(3) S. A. Moosavi, S. S. R. Hindupur, H. Shimazaki, Population coding under the scale-invariance of high-dimensional noise (2024).

(4) M. C. Morrell, A. J. Sederberg, I. Nemenman, Latent dynamical variables produce signatures of spatiotemporal criticality in large biological systems. Physical Review Letters 126, 118302 (2021).

(5) A. Renart, J. De La Rocha, P. Bartho, L. Hollender, N. Parga, A. Reyes, K. D. Harris, The asynchronous state in cortical circuits. science 327, 587–590 (2010).

(6) L. Meshulam, J. L. Gauthier, C. D. Brody, D. W. Tank, W. Bialek, Coarse graining, fixed points, and scaling in a large population of neurons. Physical Review Letters 123, 178103 (2019).